# Online Conformal Prediction via Online Optimization

**Felipe Areces** [* 1]   **Christopher Mohri** [* 2]   **Tatsunori Hashimoto** [2]   **John Duchi** [1 3]

## Abstract

We introduce a family of algorithms for online conformal prediction with coverage guarantees for both adversarial and stochastic data. In the adversarial setting, we establish the standard guarantee: over time, a pre-specified target fraction of confidence sets cover the ground truth. For stochastic data, we provide a guarantee at every time instead of just on average over time: the probability that a confidence set covers the ground truth—conditioned on past observations—converges to a pre-specified target when the conditional quantiles of the errors are a linear function of past data. Complementary to our theory, our experiments spanning over 15 datasets suggest that the performance improvement of our methods over baselines grows with the magnitude of the data's dependence, even when baselines are tuned on the test set. We put these findings to the test by pre-registering an experiment for electricity demand forecasting in Texas, where our algorithms achieve over a 10% reduction in confidence set sizes, a more than a 30% improvement in quantile and absolute losses with respect to the observed errors, and significant outcomes on all 78 out of 78 pre-registered hypotheses. We provide documentation for the pypi package implementing our algorithms here: https://conformalopt.readthedocs.io/.

## 1. Introduction

It would be unreasonable to presume that a predictive model would maintain its accuracy eternally—populations change, new scenarios arise, individuals modify behavior—so seeking a predictor that can always guarantee its accuracy is no more than a snipe hunt. Instead, however, we might seek an appropriate (and dynamic) level of confidence in our predictions, even in the face of changing data processes. Even more, we ought to develop procedures that can provide such guarantees in arbitrary settings, so that we do not rely on (likely invalid) modeling assumptions, but which can in fact do the "right thing," adapting to underlying structure when it exists. To address these desiderata, this work adopts and adapts online conformal prediction, providing predictive confidence sets for which, over time, a pre-specified fraction covers the ground truth, even if data is adversarial.

Current algorithms for online conformal prediction resemble bang-bang controllers from control theory, which switch between two extreme states to achieve a desired behavior. The first algorithm to propose this approach, Gibbs & Candès's Adaptive Conformal Inference (ACI) 2021, constructs confidence sets using a single parameter that governs their level of conservativeness. The update rule for this parameter is simple: on an error, increase the parameter by a constant, and otherwise, decrease it by (another) constant, inducing bang-bang behavior. Existing theoretical analyses for ACI suggest making these switches as aggressive as possible to obtain the tightest coverage. Several works have extended ACI by adaptively tuning the magnitude of these updates (Gibbs & Candès, 2024; Zaffran et al., 2022; Bhatnagar et al., 2023) or with more sophisticated update rules that, that for example, incorporate other aspects from control (Angelopoulos et al., 2023; Yang et al., 2024), but all fundamentally share the same bang-bang backbone.

This purely adversarial approach overlooks the possibility of achieving stronger guarantees under more predictable data. In this work, we show that directly modeling the distribution of errors can provide both new theoretical guarantees and empirical benefits. Our key observation is that we can develop algorithms that not only satisfy the standard adversarial guarantees, but also achieve time-conditional coverage guarantees for stochastic data by solving a stochastic optimization problem. These time-conditional coverage guarantees ensure that we eventually achieve the target coverage level at all times, instead of only when averaged across time. While we are not the first to depart from purely adversarial guarantees—past work considers the i.i.d. setting (Angelopoulos et al., 2024)—we provide guarantees for a

[*]Equal contribution   [1]Department of Electrical Engineering, Stanford University, Stanford, USA [2]Department of Computer Science, Stanford University, Stanford, USA [3]Department of Statistics, Stanford University, Stanford, USA. Correspondence to: Felipe Areces <fareces@stanford.edu>, Christopher Mohri <xmohri@stanford.edu>.

*Proceedings of the 42^{nd} International Conference on Machine Learning*, Vancouver, Canada. PMLR 267, 2025. Copyright 2025 by the author(s).

general class of standard stochastic processes that exhibit dependence across time, such as autoregressive processes.

Our theoretical results show the benefits of this approach across three main settings. In the first, we show the standard adversarial guarantee that over time, a pre-specified fraction of our prediction sets contain the true label. That is, in the worst-case setting where an adversary hand-picks the data, we still satisfy the same general guarantees as in past work. In the other two, we consider stochastic data. We first prove that when our model for the underlying errors is well-specified, we get a time-conditional coverage guarantee. The byproducts of these guarantees are new theoretical results for strongly convex stochastic optimization, where we develop convergence guarantees for stochastic gradient methods that operate with dependent data. In the second setting, where our model is mis-specified but an ergodic stochastic process generates the data, we show the convergence of our algorithms to the minimizer of an expected loss with respect to the stationary data distribution. Here, we extend past results on convex optimization with ergodic data (Duchi et al., 2012; Agarwal & Duchi, 2013) to last-iterate strongly convex settings.

In addition to our theory, we empirically evaluate our proposed algorithms alongside existing ones. We first use several standard datasets in the online conformal literature, where we control the marginal coverage of all algorithms to make fair comparisons. In the spirit of truly testing our proposed algorithms, we also conduct a pre-registered experiment on a period of electricity consumption data in Texas. There, we record our hyperparameter choices and experiment design before the data is available. Across all 18 datasets, our algorithms never perform discernibly worse than baselines (even after tuning baseline hyperparameters on the test set), and consistently outperform them whenever the data exhibits strong linear dependence (which we measure in terms of autocorrelation of the first lag).

We summarize our contributions and the organization of the paper below.

- Section 3 presents a new algorithm for online conformal prediction, allowing for arbitrary parametric online conformal predictors beyond scalars.
- Section 4 develops adversarial guarantees, showing it always achieves long-run coverage (1).
- Sections 5 and 6 develop stochastic guarantees, showing it attains conditional coverage when well-specified and relaxed conditional coverage when mis-specified.
- Section 7 contains experiments on 17 datasets with a pre-registration to compare our method with existing alternatives and provide evidence of its empirical benefits.

We defer all proofs to Appendix C.

## 2. Preliminaries

The goal of online conformal prediction is to generate confidence sets in an online fashion. In this setting, we sequentially observe covariates $X_t \in \mathcal{X}$ and wish to construct confidence sets $C_t \colon \mathcal{X} \rightrightarrows \mathcal{Y}$ for the labels $Y_t \in \mathcal{Y}$.

We construct these confidence sets from a score function $s_t \colon \mathcal{X} \times \mathcal{Y} \to \mathbb{R}$, which may be learned and updated over time. The score function defines $C_t(X_t)$ via the level sets

$$C_t(X_t) := \{y \in \mathcal{Y} \colon s_t(X_t, y) \leq q_t\},$$

for some threshold $q_t \in \mathbb{R}$. The conformal confidence set makes an error at time $t$ if the *conformal score*

$$S_t := s_t(X_t, Y_t),$$

exceeds the threshold $q_t$, which we formally define as $\mathrm{err}_t := \mathbb{1}_{q_t < S_t}$. The task of the online conformal learner is then to choose the thresholds $q_t$ adaptively.

The validity guarantee typically sought in online conformal prediction is that of *long-run coverage*, which requires that for any pre-specified $\alpha \in (0, 1)$, the algorithm makes an error roughly an $\alpha$ fraction of the time over $T \in \mathbb{N}$ steps:

$$\left| \frac{1}{T} \sum_{t=1}^{T} \mathrm{err}_t - \alpha \right| = o(1), \tag{1}$$

even when the data is adversarial. This is, however, a weak notion of validity, as the individual confidence sets $C_t$ need not satisfy any non-trivial coverage guarantee. In fact, one could disregard the data and satisfy the guarantee (1) by alternating between $q_t = \infty$ and $q_t = -\infty$, in which case the sets are either $C_t(X_t) = \emptyset$ with no coverage or $C_t(X_t) = \mathcal{Y}$ with trivial full coverage. A stronger guarantee, which would give a real sense of uncertainty at time $t$, would require that *conditioned on the past*, $C_t(X_t)$ contains $Y_t$ with probability $1 - \alpha$. This, of course, requires stochastic data.

Our main goal is therefore to develop algorithms that satisfy both this stronger conditional guarantee for stochastic processes as well as long-run coverage for adversarial data. More formally, when the covariate and label pairs $\{(X_t, Y_t)\}_t$ are stochastic with canonical filtration $\{\mathcal{F}_t\}_t$ defined by $\mathcal{F}_t = \sigma(\{(X_i, Y_i)\}_{i=1}^{t})$, we seek algorithms that satisfy the following consistency guarantee:

$$\left| \mathbb{P}(Y_t \notin C_t(X_t) \mid \mathcal{F}_{t-1}) - \alpha \right| = o(1), \tag{2}$$

for a general class of stochastic processes.

To preface algorithms seeking this stronger coverage condition, we briefly provide an equivalent characterization of (2) in terms of the conformal scores. Since $Y_t \in C_t(X_t) \Leftrightarrow$

$q_t \geq S_t$, achieving (2) is equivalent to tracking the $(1 - \alpha)$ conditional quantile of $S_t \mid \mathcal{F}_{t-1}$ defined as

$$q_t^\star := \inf\{q \in \mathbb{R} \colon \mathbb{P}(q \geq S_t \mid \mathcal{F}_{t-1}) \geq 1 - \alpha\},$$

as long as these quantiles are unique. This property brings about a natural loss function for our problem: the $(1 - \alpha)$ quantile loss, which is defined as

$$\ell_{\text{quantile}}(r) := (1 - \alpha) \max\{r, 0\} + \alpha \max\{-r, 0\}.$$

As first shown in Koenker & Bassett Jr. (1978), the quantiles $q_t^\star$ are the minimizers of the expected quantile loss:

$$q_t^\star = \operatorname*{argmin}_{q \in \mathbb{R}} \mathbb{E}\left[\ell_{\text{quantile}}(S_t - q) \mid \mathcal{F}_{t-1}\right].$$

We can thus achieve our goal (2) by minimizing this conditional loss, and we will use this equivalent characterization to develop online conformal algorithms.

## 2.1. Definitions

We define our *online conformal predictor* at time $t$ as a function $q_t \colon \mathcal{X}^{t-1} \times \mathcal{Y}^{t-1} \times \Theta \to \mathbb{R}$. The predictor uses the previous covariates $X_1^{t-1} := \{X_i\}_{i=1}^{t-1}$ and labels $Y_1^{t-1} := \{Y_i\}_{i=1}^{t-1}$ to compute the threshold $q_t = q_t(X_1^{t-1}, Y_1^{t-1}, \theta)$. We can update the parameter $\theta \in \Theta$ over time, and often use the simplified notation $q_t(\theta) := q_t(X_1^{t-1}, Y_1^{t-1}, \theta)$.

The closed convex parameter set $\Theta$ defining conformal predictors $\{q_t\}_t$ is *well-specified* for a stochastic process $\{(X_t, Y_t)\}_t$ if a single parameter $\theta^\star$ represents all the conditional quantiles:

$$q_t^\star = q_t(\theta^\star),$$

and *asymptotically well-specified* if some parameters only satisfy this equality asymptotically, so that for $\theta^\star \in \Theta^\star$

$$\limsup_t |q_t^\star - q_t(\theta^\star)| = 0.$$

In the asymptotic regime, our conformal predictors are also *regular* if for any $\theta \in \Theta \setminus \Theta^\star$, $\liminf_{t \to \infty} |q_t^\star - q_t(\theta)| > 0$ almost surely, and $\eta$-*bounded* if for a learning rate schedule $\eta = \{\eta_t\}_t$ and any $\theta^\star \in \Theta^\star$, $\sum_{t=1}^\infty \eta_t |q_t^\star - q_t(\theta^\star)| < \infty$ almost surely.

Many of our results apply to conformal predictors $q_t(\cdot)$ *with a bias term*. In this case, for $\tilde{\theta} \in \tilde{\Theta} \subseteq \mathbb{R}^{d-1}$ and $c \in \mathbb{R}$ such that $\theta^\top = [\tilde{\theta}^\top, c]$ and all $t \in \mathbb{N}$, there exists $h_t \colon \mathcal{X}^{t-1} \times \mathcal{Y}^{t-1} \times \tilde{\Theta} \to \mathbb{R}$ satisfying

$$q_t(\theta) = h_t(\tilde{\theta}) + c.$$

We often also require these predictors to be *bounded* so that $\tilde{\Theta} \subseteq \{\tilde{\theta} \in \mathbb{R}^{d-1} \colon \forall t \in \mathbb{N}, |h_t(\tilde{\theta})| \leq K_q\}$ for $K_q \in (0, \infty)$. In our experiments, we define conformal predictors using *feature maps* $\Phi_t \colon \mathcal{X}^{t-1} \times \mathcal{Y}^{t-1} \to \mathbb{R}^d$ producing conformal covariates $Z_t = \Phi_t(X_1^{t-1}, Y_1^{t-1})$, so that $q_t(\theta) = Z_t^\top \theta$; these are bounded if $Z_t$ and $\tilde{\theta}$ have bounded Euclidean norm $\|\cdot\|$.

---

**Algorithm 1** Batched projected online gradient descent

---

**Input:** $\theta_1 \in \mathbb{R}^d$, batch size $m$, convex feasible set $\Theta$, learning rate schedule $\{\eta_b\}$
**for** $b = 1$ **to** $B$ **do**
  **for** $i = 1$ **to** $m$ **do**
    Predict threshold $q_{b,i}(\theta_b)$
    Observe score $S_{b,i}$
    Compute gradient $\hat{\mathbf{g}}_{b,i} \leftarrow \partial \ell_{b,i}(\theta_b)$
  **end for**
  Update $\theta_{b+1} \leftarrow \Pi_\Theta\left(\theta_b - \eta_b \frac{1}{m} \sum_{i=1}^m \hat{\mathbf{g}}_{b,i}\right)$
**end for**

---

## 3. Algorithms

Our online conformal predictors serve as estimators for the conditional quantiles $q_t^\star$, so the core idea of our method is to optimize their parameters via online gradient descent on the quantile loss $\ell_t(\theta) := \ell_{\text{quantile}}(S_t - q_t(\theta))$ so as to approach the the minimizers of its conditional expectation $L_t(\theta) := \mathbb{E}[\ell_t(\theta) \mid \mathcal{F}_{t-1}]$. This procedure follows similar algorithmic principles as ACI but allows for arbitrary parametric online conformal predictors beyond scalars.

We present the framework of our proposed method in Algorithm 1, which performs a gradient descent step on the conformal predictors' parameter $\theta \in \Theta$ after observing each batch of data. To simplify notation regarding batches, we use $t$ to represent time $t$, and the tuple $(b, i)$ to denote the $i$'th sample in the $b$'th batch. Our batch size is $m$, so time $(b, i)$ is equivalent to time $(b - 1)m + i$. When subscript $b$ appears on its own, it refers to a quantity that is uniform across batch $b$.

In the following examples, we instantiate some special cases of conformal predictors that can be used with Algorithm 1.

**Example 3.1.** *Scalar quantile tracking (SQT)* (Angelopoulos et al., 2023): Choose $\Theta = \mathbb{R}$ and

$$q_t(\theta) = \theta.$$

This model is well-specified when $\{S_t\}_t$ are i.i.d. or all conditional quantiles $q_t^\star$ are equal.

**Example 3.2.** *Scalar quantile tracking (SQT) + scorecaster* $\hat{q}_t$: Choose $\Theta = \mathbb{R}$ and

$$q_t(\theta) = \theta + \hat{q}_t.$$

This model is asymptotically well-specified when $(\hat{q}_t - q_t^\star)$ converges to a constant and fully well-specified when $(\hat{q}_t - q_t^\star)$ is constant across all time. Since the conditional quantiles $q_t^\star$ are not necessarily the same for all $t$ as in Example 3.1, the scorecaster aims to predict the time dependence in the conditional quantile so that the differences $(\hat{q}_t - q_t^\star)$ are constant. In our experiment section, the model SQT+AR is of this form and fits $\hat{q}_t$ as a quantile estimator for autoregressive (AR) models.

**Example 3.3.** *Linear quantile tracking (LQT) of order $p$:* Choose $\Theta = \mathbb{R}^{p+1}$, feature map $\Phi_t(X_1^{t-1}, Y_1^{t-1}) = (S_{t-1}, \cdots, S_{t-p}, 1)$, and

$$q_t(\theta) = \theta^\top \Phi_t(X_1^{t-1}, Y_1^{t-1}).$$

This model is well-specified for AR processes of order at most $p$, and can be extended to richer feature vectors.

Our experiments focus on the models described above, but our framework is designed to be flexible and allow for even more general conformal predictors. For example, given access to $p \in \mathbb{N}$ scorecasters $\{\hat{q}_{t,i}\}_{i=1}^p$, we could extend Example 3.2 by choosing $\Theta = [0, 1]^p \times \mathbb{R}$ and

$$q_t(\theta) = \theta^\top [\hat{q}_{t,1}, \cdots, \hat{q}_{t,p}, 1].$$

Similarly to the SQT + scorecaster method, this variant would be well-specified if any of the scorecasters satisfy the conditions in Example 3.2 and enjoy regret bounds against the best predictor in hindsight with respect to the quantile loss immediately from classical online convex optimization results (Hazan, 2016). In particular, by choosing SQT predictors with learning rates in a fixed grid as scorecasters, we could obtain a conformal predictor that preserves long-run coverage and attains quantile loss at least as good as the best SQT predictor in that grid without the need for tuning.

## 4. Guarantees under adversarial data

We first focus on the setting where covariate and label pairs $(X_t, Y_t)$ are adversarial. In this context, we show that by arguments similar to those in Gibbs & Candès (2021); Angelopoulos et al. (2023; 2024), our algorithms satisfy the long-run coverage guarantee (1). We provide the proof in Section C.1 of Appendix C.

**Theorem 4.1.** *Let the scores $S_t$ be almost surely bounded for any time $t$ with $\sup_t |S_t| \leq K_s < \infty$ and let $\{\eta_b\}_b$ be a non-increasing sequence of learning rates. Then, for any bounded conformal predictor with a bias term initialized so that $|c_1| \leq K_s + K_q$, Algorithm 1 with batch size $m$ and $B = \lfloor \frac{T}{m} \rfloor$ almost surely satisfies*

$$\left| \frac{1}{T} \sum_{t=1}^T \mathrm{err}_t - \alpha \right| \leq \frac{2m(K_s + K_q + \eta_1)}{T\eta_B} + \frac{m-1}{T}.$$

This long-run coverage bound generalizes those in Angelopoulos et al. (2023) and Angelopoulos et al. (2024) for SQT, as it allows batch sizes larger than 1 and more complex underlying models.

## 5. Guarantees under well-specification

We now show that in addition to always satisfying the long-run coverage guarantee (1) for adversarial data, our algorithms achieve the conditional coverage guarantee (2) when

the parameter set $\Theta$ is well-specified. We begin by showing this property asymptotically, and then provide finite sample rates characterizing the speed of convergence when suitably large batch sizes are chosen.

For the remaining sections with stochastic data we always assume that our conformal predictor $q_t$ is linear in its parameter $\theta \in \Theta \subseteq \mathbb{R}^d$, so that $q_t(\theta) = \theta^\top Z_t$ for covariates $Z_t = \Phi_t(X_1^{t-1}, Y_1^{t-1}) \in \mathbb{R}^d$. This implies that $\ell_t$ is convex for all $t \in \mathbb{N}$. We also assume that these covariates have almost surely bounded norm $\sup_t \|Z_t\| \leq G$ so that our conformal predictor is $G$-Lipschitz.

For this section in particular, we make an assumption on the distribution of $\{S_t\}_t$: the conditional score distributions $S_t \mid \mathcal{F}_{t-1}$ have uniformly lower and upper bounded continuous densities $f_t$ in some $\varepsilon$-neighborhood around their unique quantiles $q_t^\star$. More concretely, for all $t$: $p \leq f_t(s) \leq u$ if $|s - q_t^\star| \leq \varepsilon$.

While all of our stochastic guarantees apply to the existing SQT algorithm, they are stronger for the LQT algorithm; its larger parameter set $\Theta$ is well-specified more often and has minimizers $\theta^\star \in \Theta$ at least as good, according to any loss.

### 5.1. Asymptotic Consistency

The following result shows the consistency of Algorithm 1 under the weakest assumptions, where we require only asymptotic well-specification and non-increasing learning rates $\eta_t$ satisfying $\sum_{t=1}^\infty \eta_t = \infty$ and $\sum_{t=1}^\infty \eta_t^2 < \infty$. We provide the proof in Section C.2 of Appendix C.

**Theorem 5.1.** *Let the assumptions in the previous paragraph hold, and assume that the conformal predictors have a bias term, are asymptotically well-specified, regular, and $\eta$-bounded. Then the iterates $\theta_b$ produced by Algorithm 1 satisfy*

$$\theta_b \xrightarrow{a.s.} \theta_\infty \in \Theta^\star,$$

*and attain the conditional coverage guarantee*

$$\mathbb{P}\left(Y_t \notin C_t(X_t) \mid \mathcal{F}_{t-1}\right) \xrightarrow{a.s.} \alpha.$$

This consistency result also holds more broadly for conformal predictors that ensure that $\ell_t$ is convex for all $t$, with linear models being a special case.

### 5.2. Finite-sample Convergence

We will now provide finite-sample convergence rates for Algorithm 1. The two main challenges are the changing expected conditional loss $L_t$ (with expected subgradients that depends on the time $t$) and the lack of strong convexity or smoothness in the quantile loss.

A key result for our convergence analysis is the following strong convexity lemma, which shows that the batched con-

ditional losses are large if our iterate $\theta$ is far from $\theta^\star$. This is non-trivial, as each conditional loss $L_t$ can have an infinite number of minimizes, for example when $Z_t$ represents the last $p \geq 1$ scores and has a bias term. We provide a proof of this Lemma in Section C.3 of Appendix C.

**Lemma 5.2.** *Let $\Theta$ be well-specified with finite diameter $D$. For batch $b$, let $\lambda_b$ be the minimum eigenvalue of the sample covariance matrix $\frac{1}{m} \sum_{i=1}^{m} Z_{b,i} Z_{b,i}^\top$ and let $\mu_b := \frac{p\lambda_b}{2} \min\left\{\frac{\varepsilon}{GD}, 1\right\}$. Then, the conditional loss gap in batch $b$ is lower bounded for any $\theta \in \Theta$:*

$$\frac{1}{m} \sum_{i=1}^{m} L_{b,i}(\theta) - L_{b,i}(\theta^\star) \geq \mu_b \|\theta - \theta^\star\|^2.$$

The bound above is data-dependent and for our main convergence result, we will need a positive lower bound on $\min_{b \in [B]} \mathbb{E}[\lambda_b \mid \mathcal{F}_{b-1,m}]$, which ensures that the progress with each gradient step is lower-bounded. This requires an analysis of the minimum eigenvalues $\lambda_b$. We provide an example lower bound for the weakly-stationary AR(1) model $S_t = \phi S_{t-1} + \varepsilon_t$ with $|\phi| < 1$ and zero mean $\varepsilon_t \overset{\text{iid}}{\sim} P_\varepsilon$, which we follow for most of our experiments, and prove it in Section C.4 of Appendix C.

**Lemma 5.3.** *Let $\{S_t\}_t$ follow a weakly-stationary AR(1) stochastic process where $\mathbb{E}\left[\varepsilon_t^4\right] \leq O(1)\mathbb{E}\left[\varepsilon_t^2\right]^2$, and there is a constant $\sigma > 0$ such that $\mathbb{P}([\varepsilon]_+ \geq \sigma) \geq \frac{1}{4}$ and $\mathbb{P}([-\varepsilon]_+ \geq \sigma) \geq \frac{1}{4}$. Let $Z_{b,i} = (S_{b,i}, 1)$ and $\Sigma_b = \frac{1}{m} \sum_{i=1}^{m} Z_{b,i} Z_{b,i}^\top$. Then for $m \geq \frac{1}{1-\phi}$ there are numerical constants $p, c > 0$ such that*

$$\mathbb{P}(\lambda_{\min}(\Sigma_b) \geq c \cdot (\sigma \wedge 1) \mid \mathcal{F}_{b-1,m}) \geq p.$$

*In particular, $\mathbb{E}[\lambda_{\min}(\Sigma_b) \mid \mathcal{F}_{b-1,m}] \geq pc(\sigma \wedge 1)$.*

We can now state our finite-sample convergence result, which we prove in Section C.5 of Appendix C, provided that a result analogous to Lemma 5.3 holds almost surely.

**Theorem 5.4.** *Let the assumptions of Lemma 5.2 hold, and let $\mu_{\min} \leq \min_{b \in [B]} \mathbb{E}[\mu_b \mid \mathcal{F}_{b-1,m}]$, $B \geq 4$, and $c \in [0, 1]$. Then, for any $b \leq B$ and a universal constant $C$, the iterates of Algorithm 1 with learning rates $\eta_b = \frac{2^{c-1}}{\mu_{\min} b^c}$ satisfy the following bound with probability at least $1 - \delta$:*

$$\|\theta_b - \theta^*\|^2 \leq C \frac{(\log(B \log(B)/\delta) + 1)G^2}{\mu_{\min}^2 b^c}.$$

*This implies that if $u'$ is a uniform upper bound for the conditional densities in a neighborhood of size $GD$ around $q_t^\star$, then for some (other) constant $C'$ depending on $m, u', G, \mu_{\min}$, and any $t \leq T$,*

$$\left|\mathbb{P}(Y_t \notin C_t(X_t) \mid \mathcal{F}_{t-1}) - \alpha\right| \leq C' \sqrt{\frac{(\log(T \log(T)/\delta))}{t^c}}.$$

Our result provides a data-dependent, last-iterate high-probability bound. To the best of our knowledge, it is the first finite-sample convergence analysis for SGD that does not rely on stationarity or mixing, but instead assumes only the existence of a unique minimizer consistent across time.

This convergence resuls shows a trade-off in terms of learning rate when compared to the long-run coverage bound in Theorem 4.1. As the learning rates decay quickly and approach $\Theta(1/t)$, this bound tightens to a fast $O(1/t)$, while the long-run coverage bound turns vacuous. On the other hand, as the learning rates approach $\Theta(1)$, this convergence rate becomes vacuous while the long-run coverage bound strengthens to $O(1/t)$. This suggests learning rate schedules with faster decay to achieve better convergence rates when the well-specified assumption holds. We will observe the same learning rate tradeoff in the following section, where we trade well-specification for a mixing assumption on the data.

# 6. Guarantees under mis-specification

Under mis-specification, the full conditional guarantee (2) is impossible. Thus, we weaken this goal to consider achieving a type of best-in-class approximation to conditional coverage, where for a sequence of coverage values $\alpha_t$ "optimal" relative to the data and parameters $\Theta$, we obtain

$$\left|\mathbb{P}(Y_t \notin C_t(X_t) \mid \mathcal{F}_{t-1}) - \alpha_t\right| = o(1).$$

If $\alpha_t = \alpha$ for all $t$, which holds under well-specification, we recover conditional coverage (2).

We thus consider a setting where there is some stationary distribution $\Pi$ governing the long run behavior of $\{(S_t, Z_t)\}$. For this long-run distribution $\Pi$, we define the $\alpha_t$ implicitly via the best quantile predictor

$$\theta^\star \in \underset{\theta \in \Theta}{\arg\min}\left\{L_\Pi(\theta) := \mathbb{E}_\Pi\left[\ell_{\text{quantile}}(S - Z^\top \theta)\right]\right\},$$

(Typically, we consider an ergodic stochastic process converging to a stationary distribution $\Pi$.) When the minimizer $\theta^\star$ is unique, it defines conformal sets

$$C_t^\star(X_t) = \left\{y \in \mathcal{Y} \mid s(X_t, y) \leq Z_t^\top \theta^\star\right\}$$

and "relatively optimal" confidence values

$$\alpha_t = \mathbb{P}(Y_t \notin C_t^\star(X_t) \mid \mathcal{F}_{t-1}). \qquad (3)$$

These $\alpha_t$ are often close to $\alpha$, as all other $\theta \in \Theta$ attain worse expected coverage relative to our chosen features $Z_i$. We formalize and prove this, as well as providing synthetic experiments showing the coverage properties of $\alpha_t$, in Section D.1 of Appendix D.

Aside from the good coverage provided by $\theta^\star$, it is not immediately clear that the quantile loss is the correct objective to minimize. The iterates of Algorithm 1 perform well with respect to the quantile loss since they achieve sublinear regret by the argument in Theorem 3.1 of Hazan (2016), but one may instead be concerned with alternative loss functions, such as conformal set sizes or the square loss. However, in Appendix D.2 we argue as in Mannor et al. (2009) that in the online adversarial setting, minimizing a loss function subject to long-run average constraints is impossible even in the case of simple linear functions, so it is unlikely that this is possible in the case of minimizing some arbitrary loss subject to the long-run coverage guarantee (1).

To characterize the speed of convergence to the $\alpha_t$ in (3), we will assume the stochastic process generating $\{(S_t, Z_t)\}_t$ is $\beta$-mixing. If for $0 < r \le t$, we let $P_{[r]}^t$ be the distribution of $(Z_t, S_t)$ given $\mathcal{F}_r$ with density $p_{[r]}^t$, we can define the $\beta$-mixing coefficient

$$\beta(\tau) := \sup_{t \in \mathbb{N}} \left\{ 2\mathbb{E}\left[ \left\| P_{[t]}^{t+\tau} - \Pi \right\|_{\mathrm{TV}} \right] \right\},$$

with mixing time $\tau_\beta(P, \epsilon) = \inf\{\tau \in \mathbb{N} \mid \beta(\tau) \le \epsilon\}$.

Under these assumptions, Algorithm 1 with $m = 1$ reduces to a special instance of ergodic mirror descent (Duchi et al., 2012; Agarwal & Duchi, 2013) applied to the convex function $L_\Pi(\theta)$. When the scores $S$ have densities, the expected quantile loss behaves like a strongly convex function, allowing us to extend prior analyses to the last-iterate strongly convex setting. We provide a proof of this extension in Section C.6 of Appendix C. (Our proofs formally require that under $\Pi$, the conditional score distribution $S \mid Z$ has continuous density $\pi_Z$ with the uniformly upper and lower-bounds $p_\Pi \le \pi_Z(s) \le u_\Pi$ in a $\varepsilon$-neighborhood of $Z^T\theta^\star$.)

**Theorem 6.1.** *Let the assumptions stated above hold, and let $\Theta$ have finite diameter $D$. For some $\lambda > 0$, let $\mathbb{E}_\Pi\left[ZZ^\top\right] \succeq \lambda I$, $t \in [T+1]$, $c \in [0, 1]$, and $\mu = p_\Pi \lambda \min\left\{\frac{\varepsilon}{GD}, 1\right\}$. Then the minimizer $\theta^\star$ is unique and the iterates $\theta_t$ produced by Algorithm 1 with $m = 1$ and $\eta_t = \frac{2^c}{\mu t^c}$ satisfy*

$$\mathbb{E}\left[ \|\theta_t - \theta^\star\|^2 \right] \le C \left( \frac{G^2}{\mu^2} + \frac{GD}{\mu} \right) \frac{\tau}{t^c},$$

*for $\tau = \tau_\beta(P, t^{-c})$ and a universal constant $C$. This implies that if $u$ is a uniform upper bound for the conditional densities in a $\varepsilon$-neighborhood around $Z_t^\top\theta^\star$, for some other constant $C'$ now depending on $\mu, G, D, u, \varepsilon$,*

$$\mathbb{E}\left[|\mathbb{P}\left(Y_t \notin C_t(X_t) \mid \mathcal{F}_{t-1}\right) - \alpha_t|\right] \le C'\sqrt{\frac{\tau}{t^c}}.$$

*In both cases, the expectation is taken over the samples $\{(S_i, Z_i)\}_{i=1}^t$.*

Here we also observe the same tradeoff as in Theorem 5.4 in terms of the learning rate when compared to the long-run coverage guarantee of Theorem 4.1.

# 7. Experiments

In this section, we empirically evaluate Algorithm 1 alongside baselines. We focus on SQT+AR as in Example 3.2 and LQT of order $p$ as in Example 3.3, and use batch size $m = 1$ since larger batch sizes provide comparable results.

## 7.1. Baseline algorithms

We now describe the baselines we compare against.

**Adaptive conformal inference (ACI)** (Gibbs & Candès, 2021). In offline conformal prediction, the threshold $q$ to construct confidence sets $C(\cdot)$ is chosen (up to a minor correction) as the $(1 - \alpha)$ empirical quantile of a heldout set of validation scores. ACI instead chooses the $(1 - \alpha_t)$ empirical quantile of past scores for time-varying $\alpha_t$ updated via gradient descent on the quantile loss.

Our remaining baselines fall under the conformal PID control framework (Angelopoulos et al., 2023).

**Scalar quantile tracking (SQT)**. This method (also known as conformal P control) directly updates the threshold $q_t$ via gradient descent on the quantile loss. It predicts $q_t = \sum_{i=1}^{t-1} \eta_i(\mathrm{err}_i - \alpha)$ when initialized at $q_1 = 0$.

**Conformal PI control (PI)**. This method incorporates an *error integrator* $r_t: \mathbb{R} \to \mathbb{R}$, adding $r_t(\sum_{i=1}^{t-1} \mathrm{err}_i - \alpha)$ to the SQT update. As in Angelopoulos et al. (2023), we use the *tangent integrator* $r_t(x) := K_I \tan(x \log(t)/tC_{\mathrm{sat}})$, where $K_I, C_{\mathrm{sat}} > 0$ are hyperparameters and $\tan(x) = \mathrm{sign}(x) \cdot \infty$ for $x \notin [-\pi/2, \pi/2]$.

**Conformal PID control (PID)**. This method is the same as PI, but also predicts future scores via a *scorecaster* $\hat{q}_t$, whose predictions are added to the PI update. As in Angelopoulos et al. (2023), we use a Theta (Assimakopoulos & Nikolopoulos, 2000) scorecaster, which we call the PID(T) method, and an AR scorecaster, which we call the PID(A) method.

We use the code in Angelopoulos et al. (2023) to implement the ACI, PI, and PID methods, where we replace predictions of $\infty$ with the maximum score in each test set. We test all methods with both fixed and decaying step sizes. The decaying step sizes are of the form $c \cdot t^{-0.6}$ as in Angelopoulos et al. (2024), which satisfies the conditions of our stochastic consistency guarantees. For the PI and PID methods, we additionally adopt their learning rate trick of multiplying the learning rate by the highest score over a trailing window, even though this does not satisfy the conditions of Theorem 4.1. We describe the hyperparameter grids for each

method in Section B.1 of Appendix B.

## 7.2. Existing data

In this section, we run our algorithms and baselines on datasets appearing in the online conformal literature. We describe our datasets below.

### 7.2.1. DATASETS

**Stock data (AMZN, GOOGL, MSFT).** Using stock data is common in online conformal work. Here we consider the returns of Amazon, Google, and Microsoft stock, which are datasets used in Angelopoulos et al. (2023) and contain roughly 3,000 observations each.

**Daily climate.** This dataset has 1,575 daily temperature measurements in Delhi, India from 2013 to 2017, and is also used in Angelopoulos et al. (2023).

For the stock and daily climate datasets, we use the following four base forecasters: autoregressive (AR) model of order 3, Theta model, Prophet model, and Transformer model. These are retrained at every time-step and give predictions $\{\hat{Y}_t\}_t$ of the time series $\{Y_t\}_t$ defining conformal scores $S_t := |\hat{Y}_t - Y_t|$. We use the code in Angelopoulos et al. (2023) to generate the predictions and subsequent scores. This gives a total of 16 (small) datasets.

**Elec2** (Harries, 1999). This dataset consists of 45,312 hourly measurements of electricity demand in New South Wales, Austrailia from May 7, 1996 to December 5, 1998. As in Angelopoulos et al. (2024), we use a one-day delayed moving average as base forecaster, that is $\hat{Y}_t := \frac{1}{24} \sum_{i=1}^{24} Y_{t-24-i}$ and conformal scores $S_t := |\hat{Y}_t - Y_t|$.

### 7.2.2. EXPERIMENTAL SET-UP

In all experiments, we set the confidence level to $1 - \alpha = 0.9$. We always reserve the first scores as a validation set, and set the rest as the test set. We tune the hyperparameters for our algorithms on the validation set, and *for the baselines, we directly tune the hyperparameters on the test set*. We justify this approach below.

We choose our experimental design following the recommendation in Gulrajani & Lopez-Paz (2020) to specify strategies for model selection under distribution shifts. Since this is not explicitly done in all prior online conformal work, we tune baseline hyperparameters on the test set to identify a rough upper bound on their performance. We want to neither rely on heuristics (which can only be worse) nor subject baselines to our (potentially suboptimal) tuning strategies, so our setup ensures that the best possible choices, subject to our extensive grid, are always made. Several follow-up works on ACI have considered adaptively setting hyperparameters; these are orthogonal to this work, and can be

adapted to our algorithms (Gibbs & Candès, 2024; Zaffran et al., 2022; Bhatnagar et al., 2023).

For our methods, we recommend tuning hyperparameters by performing a grid search on recent data. One almost always has access to a validation set of past conformal scores, which can give a sense of general properties of the data. We provide a starting grid in Section B.1 of Appendix B, which is implemented in our code and used in all our experiments. If the optimal values are found at the edge of the grid, the grid should be made larger.

### 7.2.3. EXPERIMENTAL RESULTS

Our experimental results come in two parts, as we separate datasets into ones with stronger and weaker linear dependence. Since we fit linear autoregressive models to the scores, we expect our algorithms to do well when the data exhibits linear dependence, and otherwise be on par with existing algorithms.

To measure this property in our setting, we consider the autocorrelation of the first lag, which quantifies the degree of linear dependence between a time series and its one-step lagged values. We foresee increasing linear dependence when the base forecaster gets worse. For example, in a regression setting, if the labels $Y_t \in \mathbb{R}$ are decomposed into their conditional mean $\mu_t$ and uncorrelated zero-mean noise $w_t$ as $Y_t = \mu_t + w_t$, then the scores $S_t = |w_t|$ have no linear dependence when the base forecaster predicts the conditional mean $\mu_t$ perfectly. Otherwise, the error in conditional mean (a function of past data) introduces correlations between the scores. This is supported by the stock data where we can directly compare base predictors: we show in Table 4 of Appendix B that larger loss (average score) corresponds to larger autocorrelation of the first lag.

In the first four blocks of Table 1, we present results for the Microsoft stock dataset. Due to space constraints, we defer to Appendix B the remaining results for Amazon stock, Google stock, and the daily climate dataset, although they are similar. In the subsequent block, we report results for the larger Elec2 dataset. We only present separate results for both fixed and decaying step sizes for our main algorithm (LQT)—for all other algorithms we report the best result. We focus on both the quantile loss to measure the closeness of the threshold to the true scores, as well as the set sizes to show how conservative the conformal sets are. In Appendix B, we also measure the average *win rate* against LQT(fixed), which we define as $\frac{1}{T} \sum_{t=1}^{T} \mathbb{1}\{\ell_{t,\text{baseline}} \leq \ell_{t,\text{LQT (fixed)}}\}$ when $\ell$ corresponds to quantile loss or set size, to get a sense of instantaneous performance.

The key takeaway is that when there is linear dependence in the conformal scores, our algorithms fit the data better

| | Dataset, metric | LQT (fixed) | LQT (decay) | SQT+AR | SQT | ACI | PI | PID(T) | PID(A) |
|---|---|---|---|---|---|---|---|---|---|
| Smaller Autocorr. | **MSFT ar**, q. loss (avg) | 0.091 | 0.09 | 0.09 | 0.088 | 0.09 | 0.088 | 0.088 | 0.091 |
| | **MSFT ar**, set size (avg) | 0.853 | 0.818 | 0.869 | 0.83 | 0.85 | 0.822 | 0.812 | 0.838 |
| | **MSFT theta**, q. loss (avg) | 0.139 | 0.133 | 0.136 | 0.14 | 0.154 | 0.14 | 0.128 | 0.138 |
| | **MSFT theta**, set size (avg) | 2.048 | 1.956 | 1.997 | 2.113 | 2.266 | 2.092 | 1.922 | 2.068 |
| Larger Autocorrelation | **MSFT prophet**, q. loss (avg) | **0.104** | **0.1** | **0.101** | 0.151 | 0.213 | 0.15 | 0.134 | 0.134 |
| | **MSFT prophet**, set size (avg) | **2.718** | **2.705** | **2.738** | 3.117 | 3.57 | 3.143 | 2.938 | 2.934 |
| | **MSFT transformer**, q. loss (avg) | **0.115** | **0.118** | **0.107** | 0.289 | 0.352 | 0.196 | 0.149 | 0.148 |
| | **MSFT transformer**, set size (avg) | **5.947** | **5.942** | **5.852** | 7.629 | 7.938 | 6.661 | 6.119 | 6.104 |
| | **Elec2**, q. loss (avg) | **0.005** | **0.005** | **0.005** | 0.013 | 0.015 | 0.013 | 0.011 | 0.011 |
| | **Elec2**, set size (avg) | **0.16** | **0.159** | **0.157** | 0.229 | 0.244 | 0.227 | 0.201 | 0.201 |
| | **Elec2**, runtime (sec) | 0.18 | - | 129.82 | 0.05 | 17.57 | 0.58 | 234.76 | 3.94 |
| | **ERCOT**, q. loss (avg) | **29.095** | **29.158** | **36.918** | 59.118 | 76.629 | 53.233 | 42.344 | 42.349 |
| | **ERCOT**, set size (avg) | **691.496** | **691.414** | 746.546 | 977.629 | 1117.69 | 912.214 | 778.808 | 778.99 |

*Table 1.* Performance of conformal predictors across datasets, separated by datasets with smaller autocorrelation (top 2) and larger autocorrelation (bottom 4) of the first lag. Our algorithms are the three on the left. Numbers are in bold if they represent at least a 5% improvement over *all* methods in the other category. All algorithms achieved at least 0.88 long-run coverage. We do not report runtime for LQT(decay) because it is the same as LQT(fixed).

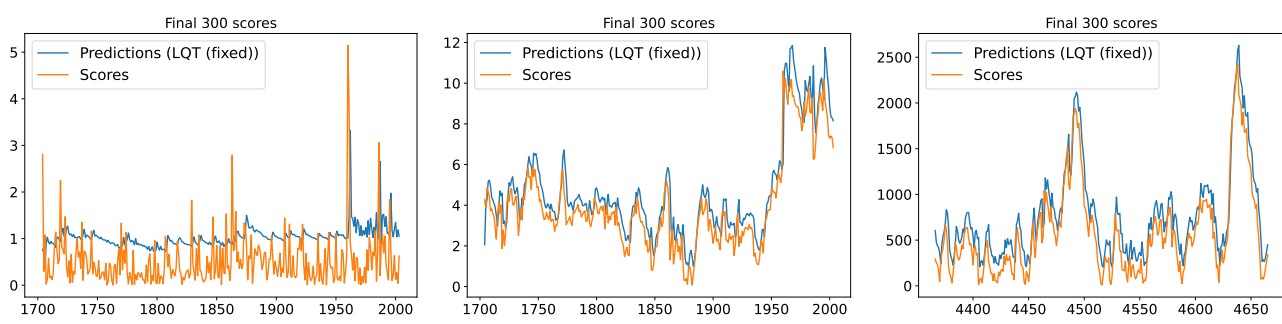

*Figure 1.* Linear quantile tracking (fixed) predictions versus scores for MSFT AR (left) and MSFT Prophet (middle) and ERCOT (right).

than the baselines. Otherwise, all algorithms perform similarly. For stock data, this dependence typically exists for the Prophet and Transformer base predictors, but not as much with the Theta and AR base predictors. This can be seen in Figure 1, where we show an ending window of scores and predictions for the LQT algorithm on both the MSFT AR and Prophet datasets. In the former, the scores look more like pure noise with a first lag having autocorrelation 0.19, while there are clearer waves in the latter with the first lag having autocorrelation 0.95. LQT is roughly constant in the former and the scores appear unpredictable; in the latter, LQT tracks the scores.

Even when results are similar, it is advantageous to use the LQT algorithm due to its lightweight nature and simple update. In Table 1, we also show the running times of each algorithm with a fixed step size for the largest Elec2 dataset. Here the $p$ order for LQT is 1, which is a very common choice in our experiments. This is only slightly slower than SQT, and much faster than any other method like PID that attempts to model the score distribution.

### 7.3. Pre-registered experiment

Our main experiment is pre-registered on Open Science Framework (OSF). The pre-registration document is included in the supplementary material and we provide the OSF link here: `https://osf.io/64jbp/`.

We gather data from the Electric Reliability Council of Texas (ERCOT), an organization that operates Texas's electrical grid. This data is accessible through the Grid Status API, which provides the true electricity load and a forecast for the load every 5 minutes. Our conformal scores are their absolute difference, and our test data is from 12/18/2024 to 1/04/2025, which provides roughly 5,000 scores. We tuned the hyperparameters for our algorithms on the data from 2 weeks prior to the date of preregistration, and *the hyperparameters for all baselines are tuned on the test data*. Summarized results are at the bottom of Table 1, and we also show an ending window of scores in the rightmost plot of Figure 1, as well as long-run coverage rates in Figure 2. Our methods show significant improvements, and the

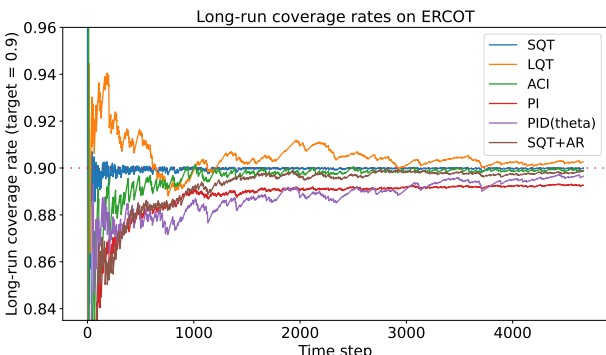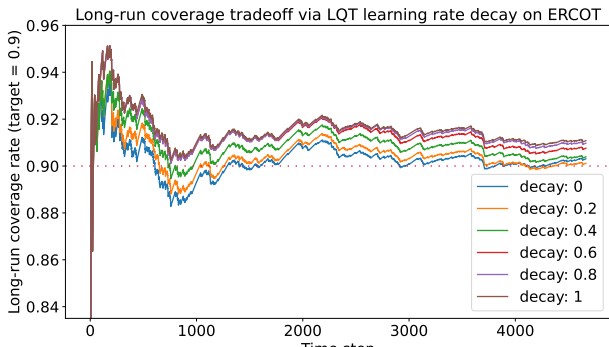

*Figure 2.* Long-run coverage of online conformal algorithms on pre-registered ERCOT data. The left plot shows the long-run coverage as a function of time for several algorithms; all converge to the target $(1 - \alpha = 0.9)$ rather quickly. On the right, we vary the learning rate decay (the learning rate is of the form $ct^{-x}$, we vary $x$ here) for LQT to exhibit the tradeoff from our theory. We observe a slower convergence for more quickly decaying learning rates.

autocorrelation of the first lag was $0.94$.

In addition to reporting our standard metrics, we also committed to hypothesis tests in our preregistration. We performed 78 tests with several loss functions comparing each baseline to LQT(fixed), and we describe these in Section B.2 of Appendix B. We report here that all 78 hypothesis tests can be rejected at a $0.05$ significance level, and that almost all p-values are significantly smaller than $0.05$.

## 8. Conclusion

In this work, we developed a new framework for online conformal prediction that not only satisfies standard marginal adversarial guarantees but also achieves time-conditional coverage at every time for more predictable stochastic data. In our experiments, we observed that the linear dependence of the data, which we measure in terms of autocorrelation, is a strong indicator of the performance improvements that our algorithms offer over the baselines.

While our approach focused on simple autoregressive models, we expect that more complex conformal predictors that, for example, use richer feature vectors or incorporate transformations of the data, can offer further empirical improvements. We expect that techniques from offline conformal prediction that approach covariate-conditional coverage can be incorporated into our framework to provide even stronger conditional guarantees. We also believe that similar consistency guarantees can hold when $\ell_t$ is weakly convex, as stochastic subgradient iterates converge to stationary points in this case (Duchi & Ruan, 2017; Davis & Drusvyatskiy, 2019).

## Acknowledgements

Partially supported by the Office of Naval Research Grant N00014-22-1-2669 and N00014-24-1-2609, as well as a gift from Schmidt Sciences. Felipe Areces acknowledges support from the Mr. and Mrs. Chun Chiu Stanford Graduate Fellowship.

## Impact Statement

This paper presents work whose goal is to advance the field of Machine Learning. There are many potential societal consequences of our work, none which we feel must be specifically highlighted here.

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

# A. Related work

Online conformal prediction builds on standard offline conformal prediction for exchangeable data (Vovk et al., 2005), a framework that has recently grown in popularity, with its tools also finding applications in areas such as risk control (Angelopoulos et al., 2022) and language modeling (Mohri & Hashimoto, 2024). It guarantees that $\mathbb{P}(Y_{t+1} \in C(X_{t+1})) \geq 1 - \alpha$, and $\mathbb{P}(Y_{t+1} \in C(X_{t+1})) \leq 1 - \alpha + \frac{1}{t+1}$, when the conformal scores $\{S_t\}_t$ are almost surely distinct (Lei et al., 2018). A key property of these results is that the probability statements are *marginal* over both the dataset used to fit the predictive sets $C$ and the new covariate $X_{t+1}$.

Real-world data can however fail to satisfy the exchangeability assumption. Similar to how more traditional machine learning theory has been adapted to develop guarantees under weaker assumptions (Mansour et al., 2009; Awasthi et al., 2023), adaptive conformal inference (ACI) (Gibbs & Candès, 2021) introduced the first algorithm to present an adversarial approach to conformal prediction. Subsequent work follows two main directions. The first explores how to adaptively tune the magnitude of ACI updates using online learning techniques, resulting in algorithms such as AgACI (Zaffran et al., 2022), SAOCP (Bhatnagar et al., 2023), and DtACI (Gibbs & Candès, 2024). These methods primarily address the adversarial setting, offering improved regret guarantees while preserving similar long-run coverage properties as ACI. Similarly to our work, Zaffran et al. (2022) discuss the idea of making ACI adaptive to time series with general dependency, but AgACI is not guaranteed to provide adversarial long-run coverage and is thus not comparable with our algorithms.

The second line of work proposes modifications to ACI's update rule inspired by ideas from control theory, leading to new algorithms like PID-conformal (Angelopoulos et al., 2023) and BCI (Yang et al., 2024). Our work is more closely aligned with a recent line of research developing online conformal algorithms that achieve good performance in stochastic settings while preserving long-run coverage when data is adversarial, such as decaying step-size ACI (Angelopoulos et al., 2024), which focuses on the i.i.d. setting.

Our work focuses on developing *time-conditional* conformal guarantees that average over the covariate distribution while conditioning on the set $C$, as in Angelopoulos et al. (2024). Such guarantees are fundamental for ensuring that our algorithms remain valid without the need for retraining, as is common in standard offline conformal algorithms such as CQR (Romano et al., 2019). Recent work also addresses the remaining challenge of providing *covariate-conditional*—or simply *conditional*—coverage guarantees, both in the offline setting (Barber et al., 2022; Gibbs et al., 2023; Jung et al., 2023; Areces et al., 2024) and in the online adversarial setting (Gupta et al., 2022; Bastani et al., 2022). In particular, Kiyani et al. (2024) introduce mean squared conditional error as a metric to evaluate algorithms seeking covariate-conditional guarantees in the stochastic i.i.d. setting, which is similar to the notion of time-conditional coverage we discuss in Section 6.

Our results are also related to the convergence theory of Stochastic Gradient Descent (SGD) (Robbins & Monro, 1951), which provides rates of convergence for the gap (or error) between the loss attained by the algorithm's outputs and the true minimizer. In the case where the algorithm receives i.i.d. samples from the target distribution and the loss is convex and smooth, Nemirovski et al. (2009) show that after $T$ steps of SGD, the expected error of the last iterate is $O(T^{-1/2})$ and $O(T^{-1})$ if the loss is additionally strongly convex.

The quantile regression framework, as presented in Koenker & Bassett Jr. (1978), has a non-smooth objective, so our results build upon the convergence theory for non-smooth functions. In this setting, many guarantees focus on the loss attained by the average iterate rather than the final iterate, achieving optimal expected rates of $O\left(T^{-1/2}\right)$ (Nemirovski & Yudin, 1983) and $O\left(T^{-1}\right)$ (Rakhlin et al., 2012) when the loss is strongly convex. Shamir & Zhang (2013) show that these rates are indeed better than those for the final iterates, which only achieve expected errors of $O\left(\log(T)T^{-1/2}\right)$ and $O\left(\log(T)T^{-1}\right)$ in the convex and strongly convex settings, respectively. Harvey et al. (2019) extends both of these to the high probability regime.

In our setting, we analyze the performance of SGD when the data is not sampled i.i.d. from the target distribution, so our results are also related to the convergence theory of SGD with dependent data. The class of problems we focus on is part of the family of stochastic problems with exogenous correlated noise (Kushner & Yin, 2003), which Duchi et al. (2012) analyzes in the context of $\beta$-mixing and $\phi$-mixing processes. Duchi et al. (2012) show that for convex losses under such mixing assumptions, the error rate of the average SGD iterate is governed by the $\beta$ and $\phi$ mixing times of our process, respectively. Agarwal & Duchi (2013) extend these results to the strongly convex setting but still focus on average iterate guarantees.

| Dataset, metric | LQT (fixed) | LQT (decay) | QT+AR | SQT | ACI | PI | PID(T) | PID(A) |
|---|---|---|---|---|---|---|---|---|
| **Elec2**, q. loss (avg) | **0.005** | **0.005** | **0.005** | 0.013 | 0.015 | 0.013 | 0.011 | 0.011 |
| **Elec2**, set size (avg) | **0.16** | **0.159** | **0.157** | 0.229 | 0.244 | 0.227 | 0.201 | 0.201 |
| **Elec2**, q. loss (win %) | 1.0 | 0.563 | 0.603 | 0.158 | 0.183 | 0.189 | 0.241 | 0.243 |
| **Elec2**, set size (win %) | 1.0 | 0.567 | 0.612 | 0.183 | 0.206 | 0.209 | 0.255 | 0.258 |
| **Elec2**, q. loss (avg) p-val | 1.0 | 0.939 | 0.899 | 0.0 | 0.0 | 0.0 | 0.0 | 0.0 |
| **Elec2**, set size (avg) p-val | 1.0 | 0.948 | 0.914 | 0.0 | 0.0 | 0.0 | 0.0 | 0.0 |
| **Elec2**, a. loss (avg) p-val | 1.0 | 0.928 | 0.886 | 0.0 | 0.0 | 0.0 | 0.0 | 0.0 |
| **Elec2**, q. loss (win %) p-val | 1.0 | 0.881 | 0.977 | 0.0 | 0.0 | 0.0 | 0.0 | 0.0 |
| **Elec2**, set size (win %) p-val | 1.0 | 0.884 | 0.988 | 0.0 | 0.0 | 0.0 | 0.0 | 0.0 |
| **Elec2**, a. loss (win %) p-val | 1.0 | 0.885 | 0.973 | 0.0 | 0.0 | 0.0 | 0.0 | 0.0 |

*Table 2.* Elec2 dataset. Our algorithms are the three on the left. Numbers are bolded if they represent at least a $5\%$ improvement over all methods in the other category. All algorithms achieved at least $0.89$ coverage.

| Dataset, metric | LQT (fixed) | LQT (decay) | QT+AR | SQT | ACI | PI | PID(T) | PID(A) |
|---|---|---|---|---|---|---|---|---|
| **ERCOT**, q. loss (avg) | **29.095** | **29.158** | **36.918** | 59.118 | 76.629 | 53.233 | 42.344 | 42.349 |
| **ERCOT**, set size (avg) | **691.496** | **691.414** | 746.546 | 977.629 | 1117.69 | 912.214 | 778.808 | 778.99 |
| **ERCOT**, q. loss (win %) | 1.0 | 0.371 | 0.399 | 0.227 | 0.288 | 0.296 | 0.286 | 0.283 |
| **ERCOT**, set size (win %) | 1.0 | 0.34 | 0.407 | 0.243 | 0.31 | 0.317 | 0.301 | 0.296 |
| **ERCOT**, q. loss (avg) p-val | 1.0 | 0.5 | 0.0 | 0.003 | 0.016 | 0.002 | 0.0 | 0.0 |
| **ERCOT**, set size (avg) p-val | 1.0 | 0.507 | 0.0 | 0.002 | 0.016 | 0.001 | 0.0 | 0.001 |
| **ERCOT**, a. loss (avg) p-val | 1.0 | 0.494 | 0.0 | 0.004 | 0.015 | 0.003 | 0.0 | 0.0 |
| **ERCOT**, q. loss (win %) p-val | 1.0 | 0.183 | 0.001 | 0.0 | 0.0 | 0.0 | 0.0 | 0.0 |
| **ERCOT**, set size (win %) p-val | 1.0 | 0.175 | 0.003 | 0.0 | 0.0 | 0.0 | 0.0 | 0.0 |
| **ERCOT**, a. loss (win %) p-val | 1.0 | 0.18 | 0.0 | 0.0 | 0.0 | 0.0 | 0.0 | 0.0 |

*Table 3.* Pre-registered ERCOT dataset. Our algorithms are the three on the left. Numbers are bolded if they represent at least a $5\%$ improvement over all methods in the other category. All algorithms achieved at least $0.89$ coverage.

## B. Additional Experiments

In this section, we report more detailed results for our experiments. In Table 2 and Table 3, we present full results for the Elec2 dataset and the pre-registered ERCOT dataset, respectively. The hyperparameter grids are in B.1, with ours tuned on validation data and the baseline hyperparameters tuned on the test sets. These additionally contain win rates versus LQT (fixed), defined as $\frac{1}{T}\sum_{t=1}^{T}\mathbb{1}\{\ell_{t,\text{baseline}} \le \ell_{t,\text{LQT (fixed)}}\}$ when $\ell$ corresponds to quantile loss or set size, as well as p-values to determine significance. For each baseline algorithm and loss function $\ell$ corresponding to quantile loss, absolute loss, and set size, we construct a time series of the form

$$\{\ell_{t,\text{baseline}} - \ell_{t,\text{LQT (fixed)}}\}_t,$$

and one with corresponding indicator form

$$\{\mathbb{1}\{\ell_{t,\text{baseline}} \ge \ell_{t,\text{LQT (fixed)}}\} - 0.5\}_t.$$

This gives 6 time series per method. With the assumption that the data is (weakly) stationary, our null hypothesis is that the baseline algorithm is at least as good as LQT(fixed). That is, the mean $\mu$ of each stochastic process is non-positive: $\mu \le 0$, with alternative hypothesis $\mu > 0$. Since the data is not independent, we must resort to a hypothesis test for dependent data. For that reason, we use the test in Lobato (2001, Section 2.1), which we describe in detail in Section B.2.

Lastly, we show full results for the smaller datasets in Table 6. We reserve the first $1/3$ of the datasets as validation data and tune our hyperparamters with the hyperparameter grid in Appendix B.1, while still tuning baseline hyperparameters on the test set.

We also show the relationship between loss of base predictor and autocorrelation of the first lag for the stock data in Table 4.

| Dataset | Base predictor | Loss (average score) | Autocorrelation of first lag |
|---------|----------------|----------------------|------------------------------|
| GOOGL | AR | 4.68 | 0.23 |
| | Prophet | 24.47 | 0.95 |
| | Theta | 11.38 | 0.76 |
| | Transformer | 151.40 | 0.99 |
| MSFT | AR | 0.38 | 0.19 |
| | Prophet | 1.78 | 0.95 |
| | Theta | 0.93 | 0.75 |
| | Transformer | 5.03 | 0.99 |
| AMZN | AR | 3.85 | 0.31 |
| | Prophet | 25.28 | 0.98 |
| | Theta | 3.86 | 0.30 |
| | Transformer | 63.53 | 0.99 |
| Daily-Climate | AR | 1.25 | 0.19 |
| | Prophet | 2.16 | 0.67 |
| | Theta | 1.48 | 0.15 |
| | Transformer | 9.83 | 0.97 |

*Table 4.* Relationship between loss (average score) and dependence (measured via autocorrelation of first lag) for stock data.

### B.1. Hyperparameters

Here we describe the hyperparameters used in our experiments. All hyperparameters are tuned via grid search, with grids presented in Table 5. Again, we emphasize that all baseline hyperparameters are always tuned on the test sets. When tuning hyperparameters, we use the ones that lead to the best quantile loss, since this is implicitly what all algorithms minimize and is also a measure of the closeness of the predictions to the true scores, provided that they achieved $1 - \alpha - 0.01 = 0.89$ coverage. If no choice achieves at least $0.89$ coverage, we disregard that constraint.

| Hyperparameter | Applicable Algorithms | Grid Values |
|----------------|----------------------|-------------|
| Learning rate | All | 1e-5, 1e-4, 1e-3, 1e-2, 1e-1, 1e0, 1e1, 1e2, 1e3, 1e4, 1e5 |
| P order | LQT, SQT+AR | 0, 1, 2 |
| Bias covariate | LQT | 0.1, 1, 5, 10, 100, 200, 1000 |
| $K_I$ | PI, PID(A), PID(T) | 10, 100, 200, 1000, Angelopoulos et al. (2023, Appendix B) heuristic |
| $C_{\text{sat}}$ | PI, PID(A), PID(T) | 0.1, 1, 5, 20, Angelopoulos et al. (2023, Appendix B) heuristic |

*Table 5.* Hyperparameter grids.

We provide intuition for the hyperparameters of LQT here. The $p$ order corresponds to the AR($p$) model that we use to predict the quantiles of our scores. Therefore, it should correspond to the number of lags with large partial autocorrelation, which can be examined by plotting the partial autocorrelation function (PACF) plot. As a heuristic, we observe that choosing $p$ equal to the number of consecutive lags with a magnitude larger than $0.2$ leads to good performance. In some cases, values larger than those in Table 5 can be helpful, as in the case of Elec2 dataset. The other main hyperparameter (besides the learning rate) is the feature corresponding to the bias in the conformal covariate. Specifically, if we choose a $p$ order of 1, the conformal covariate will be of the form $Z_t = [S_{t-1}, w]$, where $w$ is the bias covariate. While any non-zero bias term satisfies our guarantees, we generally observe that it should be on the same order of magnitude as the scores. For that reason, a heuristic is the average validation score (or a slightly smaller value). We can likely achieve a similar effect by normalizing the scores and setting $w = 1$.

#### B.1.1. SCORECASTER DETAILS

For the PID baseline methods, we implement both the Theta and AR scorecaster using code from Angelopoulos et al. (2023). Angelopoulos et al. (2023) train the scorescaster at using all past data at every time step. We follow this procedure for the smaller datasets in Table 6. This becomes prohibitively slow for larger datasets, so there we only use the most recent 200

scores. In the ERCOT pre-registration, we had committed to only using the most recent 10 scores, but we show better results here with 200 scores since this is a baseline.

The algorithm SQT+AR also uses an AR scorecaster to predict the next conditional quantile. In all datasets besides ERCOT, the AR scorecaster is trained using the most recent 200 scores via quantile regression and the `cvxpy` python library. The runtime can likely be made much faster by using other libraries. At the time of preregistration, we had only committed to using the most recent 10 scores, so for ERCOT data we only use the most recent 10 scores since this is one of our proposed methods.

### B.2. Dependent hypothesis testing

To determine if our algorithm, represented by LQT (fixed), provides a significant improvement over a baseline for a specific loss function $\ell$, we look at the following two sequences of random variables

$$X_t = \ell_{t,\text{baseline}} - \ell_{t,\text{LQT (fixed)}} \qquad\qquad W_t = \mathbb{1}\{\ell_{t,\text{baseline}} > \ell_{t,\text{LQT (fixed)}}\} - 0.5.$$

In this case, if we assume that $\mathbb{E}[X_t] = \mu_x$ and $\mathbb{E}[W_t] = \mu_w$ are constant over time, testing for a significant improvement would be equivalent to testing the null hypotheses $(H_0, \tilde{H}_0)$ against the alternative hypotheses $(H_1, \tilde{H}_1)$ in the following hypothesis tests

$$H_0 : \mu_x \leq 0 \qquad\qquad\qquad \tilde{H}_0 : \mu_w \leq 0$$
$$H_1 : \mu_x > 0 \qquad\qquad\qquad \tilde{H}_1 : \mu_w > 0.$$

However, because $X_t$ and $W_t$ come from a time series and are not i.i.d. samples from an underlying distribution, standard hypothesis tests are not applicable here. For this reason, we use the hypothesis test in Lobato (2001, Section 2.1) which applies to an arbitrary dependent sequence $Y_t$ on a probability space $(\Omega, \mathcal{A}, P)$ satisfying the following properties:

1. $Y_t$ is wide-sense stationary so that $\mathbb{E}[Y_t] = \mu_y$ for all $t \in \mathbb{N}$ and $\mathbb{E}[Y_{t_1}Y_{t_2}] = \mathbb{E}[Y_{t_1+\tau}Y_{t_2+\tau}]$ for all $t_1, t_2, \tau \in \mathbb{N}$.

2. For some $\delta > 0$, $\mathbb{E}[Y_t^{2+\delta}] < \infty$.

3. For $S_t = \sum_{i=1}^{t} Y_i$, $\mathbb{E}[S_t^2] = \sigma_t^2$ satisfies $\sigma_t^2 \to \infty$.

4. $Y_t$ is $\rho$-mixing so that the maximum correlation coefficient

$$\rho(k) = \sup\{\text{corr}(Z_1, Z_2) : Z_1 \in L_2(\Omega, \sigma(Y_i : i \leq t)), Z_2 \in L_2(\Omega, \sigma(Y_i : i \geq t+k)), t \in \mathbb{N}\},$$

satisfies $\rho(k) \to 0$.

Under these assumptions, if $\mu_y = 0$, then Herrndorf (1984, Corollary) shows that for all $r \in [0, 1]$

$$\frac{1}{\sqrt{t}} \sum_{i=1}^{\lfloor rt \rfloor} Y_i \xrightarrow{d} \phi B(r),$$

where $\phi = \sqrt{2\pi f_y(0)}$, $f_y(0)$ is the spectral density of $Y_t$ at zero frequency, and $B(r)$ is a Brownian motion process. Lobato (2001) now uses this result to conclude that the test statistic

$$V_t = \frac{t^{1/2}\bar{Y}_t}{s_t^{1/2}} \quad \text{with} \quad s_t = \frac{1}{t^2}\sum_{i=1}^{t}\left(\sum_{j=1}^{i} Y_j - \bar{Y}_t\right)^2,$$

satisfies

$$V_t \xrightarrow{d} U_{1/2} \quad \text{with} \quad U_{1/2} = \frac{B(1)}{\sqrt{\int_0^1 (B(r) - rB(1))^2\, dr}}.$$

Assuming that the conditions outlined above hold for $X_t$ and $W_t$ under the null hypothesis, we can now perform one-sided hypothesis tests on our random sequences by computing the value of $V_t$ and comparing it to the critical values of the distribution of $U_{1/2}$. We provide estimated p-values with the same simulation technique in Lobato (2001) but using 10,000,000 sequences of length 10,000.

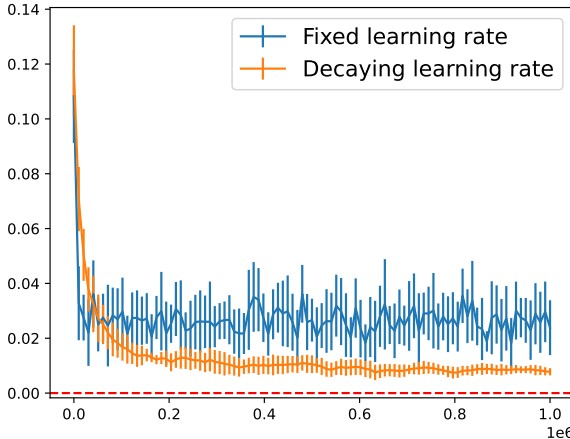

*Figure 3.* $\|\theta_t - \theta^\star\|$ vs. time $t$ for synthetic AR(2) data, averaged over 10 runs where the error bars represent standard deviation.

### B.3. Synthetic data

Here we experimentally verify our convergence theory in Section 5 by generating synthetic AR($p$) data where our well-specified assumption holds. In this case, an optimal parameter $\theta^\star \in \Theta$ exists, so we can track the distance between the iterates of Algorithm 1 and this optimal parameter.

Our synthetic data is sampled from the following AR(2) process:

$$S_t = 0.3S_{t-1} - 0.3S_{t-2} + \epsilon_t,$$

where $\epsilon_t \overset{\text{iid}}{\sim} N(0,1)$. If we now run Algorithm 1 with covariates $Z_t = [S_{t-1}, S_{t-2}, 1]^\top$, the optimal parameters are $\theta^\star = [0.3, -0.3, \phi^{-1}(1-\alpha)]^\top$, where $\phi^{-1}$ represents the inverse of the standard normal CDF.

In Figure 3, we plot the distance to $\theta^\star$ over time for fixed and decaying learning rates. The decaying learning rates are $\Theta(1/t)$, which Theorem 5.4 suggests gives the fastest convergence. The results are as expected: fixed learning rates appear to converge to a noise ball, and decaying learning rates appear to converge to $\theta^\star$, suggesting that the conditional coverage condition (2) holds.

| Dataset, metric | LQT (fixed) | LQT (decay) | SQT+AR | SQT | ACI | PI | PID(T) | PID(A) |
|---|---|---|---|---|---|---|---|---|
| **MSFT ar**, q. loss (avg) | 0.091 | 0.09 | 0.09 | 0.088 | 0.09 | 0.088 | 0.088 | 0.091 |
| **MSFT ar**, set size (avg) | 0.853 | 0.818 | 0.869 | 0.83 | 0.85 | 0.822 | 0.812 | 0.838 |
| **MSFT ar**, q. loss (win %) | 1.0 | 0.554 | 0.486 | 0.538 | 0.538 | 0.515 | 0.526 | 0.517 |
| **MSFT ar**, set size (win %) | 1.0 | 0.566 | 0.474 | 0.542 | 0.538 | 0.526 | 0.528 | 0.517 |
| **MSFT theta**, q. loss (avg) | 0.139 | 0.133 | 0.136 | 0.14 | 0.154 | 0.14 | 0.128 | 0.138 |
| **MSFT theta**, set size (avg) | 2.048 | 1.956 | 1.997 | 2.113 | 2.266 | 2.092 | 1.922 | 2.068 |
| **MSFT theta**, q. loss (win %) | 1.0 | 0.561 | 0.485 | 0.428 | 0.447 | 0.451 | 0.547 | 0.505 |
| **MSFT theta**, set size (win %) | 1.0 | 0.563 | 0.49 | 0.423 | 0.446 | 0.457 | 0.543 | 0.504 |
| **MSFT prophet**, q. loss (avg) | **0.104** | **0.1** | **0.101** | 0.151 | 0.213 | 0.15 | 0.134 | 0.134 |
| **MSFT prophet**, set size (avg) | **2.718** | **2.705** | **2.738** | 3.117 | 3.57 | 3.143 | 2.938 | 2.934 |
| **MSFT prophet**, q. loss (win %) | 1.0 | 0.519 | 0.466 | 0.302 | 0.281 | 0.301 | 0.257 | 0.251 |
| **MSFT prophet**, set size (win %) | 1.0 | 0.508 | 0.449 | 0.31 | 0.3 | 0.309 | 0.268 | 0.269 |
| **MSFT transformer**, q. loss (avg) | **0.115** | **0.118** | **0.107** | 0.289 | 0.352 | 0.196 | 0.149 | 0.148 |
| **MSFT transformer**, set size (avg) | 5.947 | 5.942 | 5.852 | 7.629 | 7.938 | 6.661 | 6.119 | 6.104 |
| **MSFT transformer**, q. loss (win %) | 1.0 | 0.443 | 0.543 | 0.29 | 0.243 | 0.345 | 0.38 | 0.373 |
| **MSFT transformer**, set size (win %) | 1.0 | 0.441 | 0.536 | 0.294 | 0.257 | 0.356 | 0.379 | 0.373 |
| **AMZN ar**, q. loss (avg) | 1.198 | 1.215 | 1.205 | 1.198 | 1.29 | 1.195 | 1.197 | 1.228 |
| **AMZN ar**, set size (avg) | 10.112 | 9.711 | 10.398 | 10.112 | 11.592 | 10.009 | 10.15 | 10.63 |
| **AMZN ar**, q. loss (win %) | 1.0 | 0.538 | 0.446 | 1.0 | 0.508 | 0.537 | 0.528 | 0.469 |
| **AMZN ar**, set size (win %) | 1.0 | 0.556 | 0.446 | 1.0 | 0.513 | 0.554 | 0.554 | 0.481 |
| **AMZN theta**, q. loss (avg) | 1.218 | 1.203 | 1.207 | 1.188 | 1.283 | 1.192 | 1.195 | 1.23 |
| **AMZN theta**, set size (avg) | 10.252 | 10.364 | 10.586 | 9.927 | 11.364 | 9.91 | 9.976 | 10.426 |
| **AMZN theta**, q. loss (win %) | 1.0 | 0.405 | 0.397 | 0.452 | 0.42 | 0.458 | 0.453 | 0.406 |
| **AMZN theta**, set size (win %) | 1.0 | 0.379 | 0.371 | 0.434 | 0.394 | 0.446 | 0.438 | 0.391 |
| **AMZN prophet**, q. loss (avg) | 2.208 | **1.419** | **1.341** | 2.432 | 2.998 | 2.264 | 1.855 | 1.84 |
| **AMZN prophet**, set size (avg) | 49.995 | **42.825** | **43.105** | 52.808 | 57.934 | 51.747 | 46.209 | 46.231 |
| **AMZN prophet**, q. loss (win %) | 1.0 | 0.529 | 0.511 | 0.341 | 0.293 | 0.364 | 0.384 | 0.378 |
| **AMZN prophet**, set size (win %) | 1.0 | 0.514 | 0.488 | 0.352 | 0.302 | 0.369 | 0.386 | 0.379 |
| **AMZN transformer**, q. loss (avg) | **1.713***  | **1.863** | **1.39** | 4.975 | 6.622 | 4.286 | 2.055 | 2.057 |
| **AMZN transformer**, set size (avg) | 93.963* | 97.903 | 92.614 | 127.9 | 143.724 | 120.728 | 97.04 | 97.04 |
| **AMZN transformer**, q. loss (win %) | 1.0 | 0.389 | 0.567 | 0.309 | 0.25 | 0.292 | 0.283 | 0.274 |
| **AMZN transformer**, set size (win %) | 1.0 | 0.367 | 0.545 | 0.314 | 0.261 | 0.307 | 0.277 | 0.269 |
| **GOOGL ar**, q. loss (avg) | 1.117 | 1.136 | 1.138 | 1.122 | 1.185 | 1.124 | 1.118 | 1.142 |
| **GOOGL ar**, set size (avg) | 10.436 | 9.828 | 10.681 | 10.335 | 11.167 | 10.334 | 9.932 | 10.827 |
| **GOOGL ar**, q. loss (win %) | 1.0 | 0.63 | 0.465 | 0.418 | 0.545 | 0.541 | 0.633 | 0.495 |
| **GOOGL ar**, set size (win %) | 1.0 | 0.675 | 0.477 | 0.413 | 0.557 | 0.567 | 0.666 | 0.494 |
| **GOOGL theta**, q. loss (avg) | 1.744 | 1.61 | 1.615 | 1.76 | 1.911 | 1.762 | 1.623 | 1.724 |
| **GOOGL theta**, set size (avg) | 24.999 | 23.433 | 23.027 | 25.503 | 27.248 | 24.773 | 23.235 | 24.545 |
| **GOOGL theta**, q. loss (win %) | 1.0 | 0.594 | 0.603 | 0.458 | 0.495 | 0.509 | 0.586 | 0.512 |
| **GOOGL theta**, set size (win %) | 1.0 | 0.598 | 0.616 | 0.455 | 0.5 | 0.523 | 0.598 | 0.519 |
| **GOOGL prophet**, q. loss (avg) | **1.335** | **1.33** | **1.317** | 2.086 | 2.459 | 2.072 | 1.812 | 1.803 |
| **GOOGL prophet**, set size (avg) | **34.916** | **35.136** | **35.297** | 43.597 | 47.222 | 43.553 | 39.424 | 39.154 |
| **GOOGL prophet**, q. loss (win %) | 1.0 | 0.38 | 0.406 | 0.27 | 0.258 | 0.272 | 0.245 | 0.251 |
| **GOOGL prophet**, set size (win %) | 1.0 | 0.333 | 0.371 | 0.279 | 0.27 | 0.284 | 0.248 | 0.25 |
| **GOOGL transformer**, q. loss (avg) | **1.711** | **1.585*** | **1.6** | 5.729 | 12.424 | 4.045 | 2.396 | 2.407 |
| **GOOGL transformer**, set size (avg) | 208.989 | 206.891* | 207.588 | 245.381 | 291.003 | 228.834 | 212.054 | 211.965 |
| **GOOGL transformer**, q. loss (win %) | 1.0 | 0.566 | 0.601 | 0.158 | 0.212 | 0.277 | 0.318 | 0.318 |
| **GOOGL transformer**, set size (win %) | 1.0 | 0.557 | 0.605 | 0.174 | 0.227 | 0.292 | 0.322 | 0.322 |
| **daily-climate ar**, q. loss (avg) | 0.243 | 0.246 | 0.249 | 0.243 | 0.251 | 0.239 | 0.237 | 0.254 |
| **daily-climate ar**, set size (avg) | 2.708 | 2.514 | 2.609 | 2.632 | 2.689 | 2.705 | 2.583 | 2.708 |
| **daily-climate ar**, q. loss (win %) | 1.0 | 0.537 | 0.525 | 1.0 | 0.5 | 0.549 | 0.556 | 0.47 |
| **daily-climate ar**, set size (win %) | 1.0 | 0.539 | 0.532 | 1.0 | 0.504 | 0.558 | 0.564 | 0.461 |
| **daily-climate theta**, q. loss (avg) | 0.274 | 0.268 | 0.269 | 0.264 | 0.275 | 0.265 | 0.263 | 0.288 |
| **daily-climate theta**, set size (avg) | 3.325 | 3.351 | 3.291 | 3.295 | 3.192 | 3.245 | 3.174 | 3.247 |
| **daily-climate theta**, q. loss (win %) | 1.0 | 0.47 | 0.476 | 0.539 | 0.527 | 0.552 | 0.542 | 0.483 |
| **daily-climate theta**, set size (win %) | 1.0 | 0.454 | 0.473 | 0.541 | 0.533 | 0.57 | 0.539 | 0.498 |
| **daily-climate prophet**, q. loss (avg) | **0.269** | **0.275** | **0.281** | 0.317 | 0.3 | 0.309 | 0.32 | 0.317 |
| **daily-climate prophet**, set size (avg) | **3.828** | **3.788** | **3.918** | 4.219 | 4.22 | 4.168 | 4.2 | 4.162 |
| **daily-climate prophet**, q. loss (win %) | 1.0 | 0.506 | 0.483 | 0.347 | 0.383 | 0.394 | 0.356 | 0.374 |
| **daily-climate prophet**, set size (win %) | 1.0 | 0.515 | 0.482 | 0.351 | 0.39 | 0.404 | 0.371 | 0.399 |
| **daily-climate transformer**, q. loss (avg) | **0.313** | **0.317** | **0.315** | 0.443 | 0.359 | 0.392 | 0.42 | 0.421 |
| **daily-climate transformer**, set size (avg) | **9.1** | **9.148** | **8.955** | 10.126 | 9.923 | 9.808 | 9.831 | 9.899 |
| **daily-climate transformer**, q. loss (win %) | 1.0 | 0.409 | 0.549 | 0.431 | 0.376 | 0.431 | 0.345 | 0.311 |
| **daily-climate transformer**, set size (win %) | 1.0 | 0.406 | 0.548 | 0.437 | 0.384 | 0.436 | 0.359 | 0.317 |

*Table 6.* Numbers are bolded if they represent at least a 5% improvement over all methods in the other category. Asterisk numbers achieved less than 0.87 coverage. All baselines achieved at least 0.89 coverage because we tuned them on the test set.

# C. Deferred proofs for Sections 4, 5, and 6

## C.1. Proof of Theorem 4.1

This proof follows Theorems 1 and 2 of (Angelopoulos et al., 2024). Similarly, we argue that adversarial coverage follows from the boundedness of the bias terms in our iterates. More concretely, using the decomposition $\theta_b = (\tilde{\theta}_b, c_b) \in \tilde{\Theta} \times \mathbb{R}$ we prove the following proposition.

**Proposition C.1.** *If $\sup_b |c_b| \leq C < \infty$ almost surely, then with probability 1, Algorithm 1 satifies*

$$\left| \frac{1}{T} \sum_{t=1}^{T} \text{err}_t - \alpha \right| \leq \frac{2mC}{T\eta_B} + \frac{m-1}{T}.$$

It now remains to show that the bias terms $c_b$ are indeed bounded, which we do via the following claim.

**Claim C.1.** *The iterates of Algorithm 1 satisfy $\sup_b |c_b| \leq K_s + K_q + \eta_1$ almost surely.*

We now provide proofs for both of these results.

**Demonstrating Proposition C.1.** The final time $T$ is not necessarily divisible by the batch size $m$, so by time $T$, Algorithm 1 will only perform updates based on the first $Bm \leq T$ scores. For this reason, we first point out that by the triangle inequality,

$$\left| \frac{1}{T} \sum_{t=1}^{T} (\text{err}_t - \alpha) \right| \leq \left| \frac{1}{T} \sum_{t=1}^{Bm} (\text{err}_t - \alpha) \right| + \frac{1}{T} \sum_{i=1}^{T-Bm} |\text{err}_{B+1,i} - \alpha|,$$

and since $|\text{err}_{B+1,i} - \alpha| \leq 1$ for all $i \in [T - Bm]$ and $T - Bm \leq m - 1$,

$$\left| \frac{1}{T} \sum_{t=1}^{T} (\text{err}_t - \alpha) \right| \leq \left| \frac{1}{T} \sum_{t=1}^{Bm} (\text{err}_t - \alpha) \right| + \frac{m-1}{T}.$$

The rightmost term $\frac{m-1}{T}$ matches the result in Proposition C.1, so it only remains to show that

$$\left| \frac{1}{T} \sum_{t=1}^{Bm} (\text{err}_t - \alpha) \right| \leq \frac{2mC}{T\eta_B}. \tag{4}$$

For this purpose, we define

$$\Delta_b := \begin{cases} m\eta_1^{-1} & \text{if } b = 1 \\ m(\eta_b^{-1} - \eta_{b-1}^{-1}) & \text{if } b > 1, \end{cases}$$

to argue that the left-hand side of inequality (4) can be written in terms of the bias terms $\{c_b\}_{b=1}^{B}$ as

$$\left| \frac{1}{T} \sum_{t=1}^{Bm} (\text{err}_t - \alpha) \right| = \left| \frac{1}{T} \sum_{b=1}^{B} \sum_{i=1}^{m} (\text{err}_{b,i} - \alpha) \right|$$

$$= \left| \frac{1}{T} \sum_{b=1}^{B} \sum_{i=1}^{m} \left( \sum_{r=1}^{b} \Delta_r \right) \frac{\eta_b}{m} (\text{err}_{b,i} - \alpha) \right|$$

$$= \left| \frac{1}{T} \sum_{r=1}^{B} \Delta_r \left( \sum_{b=r}^{B} \sum_{i=1}^{m} \frac{\eta_b}{m} (\text{err}_{b,i} - \alpha) \right) \right|$$

$$= \left| \frac{1}{T} \sum_{r=1}^{B} \Delta_r (c_{B+1} - c_r) \right|.$$

Since the bias terms are bounded $|c_{B+1} - c_r| \leq 2C$, by Hölder's inequality we have that

$$\left| \frac{1}{T} \sum_{t=1}^{Bm} \text{err}_t - \alpha \right| \leq \frac{2C}{T} \|\Delta_{1:B}\|_1 .$$

Since the learning rate schedule $\{\eta_b\}_{b=1}^B$ is non-increasing $\|\Delta_{1:B}\|_1$ is a telescoping sum so that

$$\|\Delta_{1:B}\|_1 = m\eta_B^{-1},$$

which shows inequality (4) and thus the desired result.

**Demonstrating Claim C.1.** For $C := (K_s + K_q + \eta_1)$, we will only prove that $c_b \geq -C$ with probability 1, since the other direction $c_b \leq C$ follows by an identical argument. We will proceed by contradiction, so we let $\tau := \inf\{b \in \mathbb{N} \mid c_b < -C\} < \infty$ be the first index where $c_b$ falls below $-C$, and note that $\tau > 1$ since $|c_1| \leq K_s + K_q$. We will now show that if a finite $\tau$ existed, then $c_{\tau-1} < c_\tau < -C$, contradicting our definition that $\tau$ is the first index where $c_b$ falls below $-C$.

Recall that $\Theta = \tilde{\Theta} \times \mathbb{R}$, so the projection step in Algorithm 1 leaves the last coordinate unchanged, and the update for the bias term has bounded magnitude

$$\sup_b |c_{b+1} - c_b| = \sup_b \frac{\eta_b}{m} \sum_{i=1}^m |\text{err}_{b,i} - \alpha| \leq \eta_1.$$

This implies that the previous bias term $c_{\tau-1} < -(K_s + K_q)$ and the predictions for batch $(\tau - 1)$ with $i \in [m]$ satisfy

$$q_{\tau-1,i}(\theta_{\tau-1}) = h_{\tau-1,i}(\tilde{\theta}_{\tau-1}) + c_{\tau-1} < -K_s.$$

But, by our assumption that all scores are at least $-K_s$ almost surely, these predictions all lead to errors as $\text{err}_{\tau-1,i} = \mathbb{1}_{q_{\tau-1,i} < S_{\tau-1,i}} = 1$ and thus the previous update must inrease the bias term:

$$c_\tau = c_{\tau-1} + \frac{\eta_\tau}{m} \sum_{i=1}^m (\text{err}_{\tau-1,i} - \alpha) > c_{\tau-1},$$

which shows the desired contradiction and demonstrates Claim C.1. This argument only requires that the bias term can take any value in the set $[-C, C]$ so it can hold if $\Theta$ has finite diameter, as is required by some of our other results.

### C.2. Proof of Theorem 5.1

In our search for consistent conformal predictors that satisfy the conditional coverage guarantee (2), we must first characterize what model parameters (if any) attain this property. Our first step is thus to establish the conditional coverage of any sequence which eventually represents all the conditional quantiles, matching our definition of asymptotic well-specification:

**Proposition C.2.** *For any sequence of iterates satisfying $\theta_b \xrightarrow{a.s.} \theta_\infty \in \Theta^\star$, then*

$$\mathbb{P}\left(Y_t \in C_t(X_t) \mid \mathcal{F}_{t-1}\right) \xrightarrow{a.s.} 1 - \alpha,$$

*and our conformal predictors satisfy the consistency guarantee (2).*

In light of this proposition, proving Theorem 5.1 is equivalent to showing that Algorithm 1 eventually converges to some $\theta_\infty \in \Theta^\star$:

**Proposition C.3.** *The sequence of iterates $\{\theta_b\}_b$ produced by Algorithm 1 satisfies*

$$\theta_b \xrightarrow{a.s.} \theta_\infty \in \Theta^\star.$$

We now provide proofs for both of these statements.

**Demonstrating Proposition C.2.** Recall that for $b = \lfloor \frac{t}{m} \rfloor$

$$\mathbb{P}\left(Y_t \in C_t(X_t) \mid \mathcal{F}_{t-1}\right) = \mathbb{P}\left(S_t \leq q_t(\theta_b) \mid \mathcal{F}_{t-1}\right)$$

$$= \mathbb{P}\left(S_t \leq q_t^\star \mid \mathcal{F}_{t-1}\right) + \int (\mathbf{1}_{S_t \in [q_t^\star, q_t(\theta_b)]} - \mathbf{1}_{S_t \in [q_t(\theta_b), q_t^\star]}) dP_{S_t \mid \mathcal{F}_{t-1}}.$$

For any $\varepsilon > 0$, if we let $\varepsilon$ be the neighborhood around the quantiles where the uniform density upper bound $p_1$ holds, with probability 1 we can take $t$ and $b$ large enough that

$$|q_t(\theta_\infty) - q_t^\star| \leq \frac{\min\{\epsilon/p_1, \varepsilon\}}{2}$$

$$|q_t(\theta_b) - q_t(\theta_\infty)| \leq \frac{\min\{\epsilon/p_1, \varepsilon\}}{2},$$

simultaneously. For such $t$, it must also be the case that

$$|\mathbb{P}\left(Y_t \in C_t(X_t) \mid \mathcal{F}_{t-1}\right) - (1 - \alpha)| \leq \epsilon,$$

and since $\epsilon$ was arbitrary

$$\mathbb{P}\left(Y_t \in C_t(X_t) \mid \mathcal{F}_{t-1}\right) \xrightarrow{a.s.} 1 - \alpha.$$

**Demonstrating Proposition C.3.** We start by defining $\hat{\mathbf{g}}_{b,i} \in \partial l_{b,i}(\theta_b)$ as our empirical subgradient for batch $b$ and index $i$ satisfying $\mathbf{g}_{b,i} = \mathbb{E}\left[\hat{\mathbf{g}}_{b,i} \mid \mathcal{F}_{b,i-1}\right] \in \partial L_{b,i}(\theta_b)$. For any $\theta^\star \in \Theta^\star$, we now use our assumption that the conformal predictor has a bias term to define

$$\Delta_{b,i} := \begin{bmatrix} 0 \\ \vdots \\ 0 \\ q_{b,i}^\star - q_{b,i}(\theta^\star) \end{bmatrix},$$

which can be nonzero as we only require asymptotic well-specification, so that

$$q_{b,i}(\theta^\star + \Delta_{b,i}) = q_{b,i}^\star,$$

and

$$0 \in \partial L_{b,i}(\theta^\star + \Delta_{b,i}).$$

Our proof is based on the proof of Theorem 4 in Chapter 7 of Ryu & Yin (2022), which hinges on the following three claims:

(i) Almost surely, $\lim_{b \to \infty} \|\theta_b - \theta^\star\|$ exists and is finite for all $\theta^\star \in \Theta^\star$.

(ii) The iterates $\{\theta_b\}_b$ are arbitrarily close to optimal infinitely often, since for any $\theta^\star \in \Theta^\star$

$$\liminf_{b \to \infty} \frac{1}{m} \sum_{i=1}^{m} \langle \mathbf{g}_{b,i}, \theta_b - (\theta^\star + \Delta_{b,i}) \rangle = 0.$$

(iii) Claims (i) and (ii) imply that $\theta_b \xrightarrow{a.s.} \theta_\infty \in \Theta^\star$.

**Demonstrating Claims (i) and (ii).** For this part, we rely on the Robbins and Siegmund quasimartingale convergence theorem (Robbins & Siegmund, 1971), which we state in Theorem E.1 in Appendix E without proof. To apply this result, we must first construct an appropriate quasimartingale.

**Claim C.2.** *For any $\theta^\star \in \Theta^\star$, define the $\mathcal{F}_{mb}$-measurable random variables*

$$V_b := \|\theta_{b+1} - \theta^\star\|^2$$

$$D_b := 2\frac{\eta_{b+1}}{m} \sum_{i=1}^m \mathbb{E}\left[\langle \hat{\mathbf{g}}_{b+1,i}, \theta_{b+1} - (\theta^\star + \Delta_{b+1,i})\rangle \mid \mathcal{F}_{mb}\right]$$

$$U_b := G^2\eta_{b+1}^2 + 2\frac{\eta_{b+1}}{m}\sum_{i=1}^m \mathbb{E}\left[|q_{b+1,i}^\star - q_{b+1,i}(\theta^\star)| \mid \mathcal{F}_{mb}\right].$$

*Then $V_b, D_b, U_b$ are non-negative with $\sum_{b=1}^\infty U_b < \infty$ almost surely and*

$$\mathbb{E}\left[V_b \mid \mathcal{F}_{m(b-1)}\right] \leq V_{b-1} - D_{b-1} + U_{b-1},$$

*so we can apply Theorem E.1.*

*Proof.* Since $\theta_{b+1} = \Pi_\Theta\left(\theta_b - \frac{\eta_b}{m}\sum_{i=1}^m \hat{\mathbf{g}}_{b,i}\right)$

$$\|\theta_{b+1} - \theta^\star\|^2 \leq \|\theta_b - \theta^\star\|^2 + \eta_b^2 \left\|\frac{1}{m}\sum_{i=1}^m \hat{\mathbf{g}}_{b,i}\right\|^2 - 2\frac{\eta_b}{m}\sum_{i=1}^m \langle \hat{\mathbf{g}}_{b,i}, \theta_b - \theta^\star\rangle$$

$$\leq \|\theta_b - \theta^\star\|^2 + G^2\eta_b^2 - 2\frac{\eta_b}{m}\sum_{i=1}^m \langle \hat{\mathbf{g}}_{b,i}, \theta_b - (\theta^\star + \Delta_{b,i})\rangle - 2\frac{\eta_b}{m}\sum_{i=1}^m \langle \hat{\mathbf{g}}_{b,i}, \Delta_{b,i}\rangle.$$

If we now let $\theta_b = (\tilde{\theta}_b, c_b) \in \tilde{\Theta} \times \mathbb{R}$, we can use the fact that our model has a bias term to argue that the last coordinate of our empirical gradient $[\hat{\mathbf{g}}_{b,i}]_d$ satisfies

$$[\hat{\mathbf{g}}_{b,i}]_d \in \partial_{c_b}\ell(S_{b,i} - h_{b,i}(\tilde{\theta}_b) - c_b),$$

so that $|[\hat{\mathbf{g}}_{b,i}]_d| \leq \max\{\alpha, 1 - \alpha\} < 1$ and

$$\left|\sum_{i=1}^m \langle \hat{\mathbf{g}}_{b,i}, \Delta_{b,i}\rangle\right| = \left|\sum_{i=1}^m [\hat{\mathbf{g}}_{b,i}]_d(q_{b,i}^\star - q_{b,i}(\theta^\star))\right| \leq \sum_{i=1}^m |q_{b,i}^\star - q_{b,i}(\theta^\star)|.$$

We can now use this fact and our previous upper bound to conclude that

$$\|\theta_{b+1} - \theta^\star\|^2 \leq \|\theta_b - \theta^\star\|^2 + G^2\eta_b^2 + 2\frac{\eta_b}{m}\sum_{i=1}^m |q_{b,i}^\star - q_{b,i}(\theta^\star)| - 2\frac{\eta_b}{m}\sum_{i=1}^m \langle \hat{\mathbf{g}}_{b,i}, \theta_b - (\theta^\star + \Delta_{b,i})\rangle.$$

This shows that the $\mathcal{F}_{mb}$-measurable random variables

$$D_b = 2\frac{\eta_{b+1}}{m}\sum_{i=1}^m \mathbb{E}\left[\langle \hat{\mathbf{g}}_{b+1,i}, \theta_{b+1} - (\theta^\star + \Delta_{b+1,i})\rangle \mid \mathcal{F}_{mb}\right]$$

$$U_b = G^2\eta_{b+1}^2 + 2\frac{\eta_{b+1}}{m}\sum_{i=1}^m \mathbb{E}\left[|q_{b+1,i}^\star - q_{b+1,i}(\theta^\star)| \mid \mathcal{F}_{mb}\right].$$

satisfy

$$\mathbb{E}\left[V_b \mid \mathcal{F}_{m(b-1)}\right] \leq V_{b-1} - D_{b-1} + U_{b-1}.$$

The sequence $\{D_b\}_b$ is also non-negative as

$$D_b = 2\frac{\eta_{b+1}}{m}\sum_{i=1}^m \mathbb{E}\left[\mathbb{E}\left[\langle \hat{\mathbf{g}}_{b+1,i}, \theta_{b+1} - (\theta^\star + \Delta_{b+1,i})\rangle \mid \mathcal{F}_{mb+i-1}\right] \mid \mathcal{F}_{mb}\right]$$

$$= 2\frac{\eta_{b+1}}{m}\sum_{i=1}^m \mathbb{E}\left[\langle \mathbf{g}_{b+1,i}, \theta_{b+1} - (\theta^\star + \Delta_{b+1,i})\rangle \mid \mathcal{F}_{mb}\right]$$

$$\geq 0,$$

since $L_{b,i}(\theta)$ is convex and $(\theta^\star + \Delta_{b,i})$ minimizes $L_{b,i}$ for any $(b,i)$. Additionally, $U_b$ is also non-negative and by $\eta$-summability and the fact that $\sum_{b=1}^\infty \eta_b^2 < \infty$, with probability 1

$$\sum_{b=1}^\infty U_b < \infty.$$

$\square$

We can now use Claim C.2 and Theorem E.1 to conclude that almost surely

(1) $\lim_{b\to\infty} V_b$ exists and is finite.

(2) $\sum_{b=2}^\infty \sum_{i=1}^m \frac{\eta_b}{m} \langle \mathbf{g}_{b,i}, \theta_b - (\theta^\star + \Delta_{b,i}) \rangle < \infty.$

Theorem E.1 states that $\sum_{b=1}^\infty D_b < \infty$ almost surely instead of (2), but this is immediate consequence of the former since $\sum_{b=1}^\infty D_b$ being almost surely finite implies that $\mathbb{E}\left[\sum_{b=1}^\infty D_b\right] < \infty$ and

$$\mathbb{E}\left[\sum_{b=1}^\infty D_b\right] = \mathbb{E}\left[2\sum_{b=2}^\infty \sum_{i=1}^m \frac{\eta_b}{m} \langle \mathbf{g}_{b,i}, \theta_b - (\theta^\star + \Delta_{b,i}) \rangle\right],$$

so the finiteness of this expectation also implies (2). This argument holds for any $\theta^\star \in \Theta^\star$ so by (1) for all $\theta^\star \in \Theta^\star$ [with probability 1 $\lim_{b\to\infty} \|\theta_b - \theta^\star\|$ exists], and since $\Theta^\star \subset \mathbb{R}^d$ by Proposition 1 in Chapter 5 of Ryu & Yin (2022) this implies (i) as with probability 1 [for all $\theta^\star \in \Theta^\star$, $\lim_{b\to\infty} \|\theta_b - \theta^\star\|$ exists]. For (ii), since $\sum_{b=1}^\infty \eta_b = \infty$, by our positivity constraint (2) implies that for any $\theta^\star \in \Theta^\star$,

$$\liminf_{b\to\infty} \frac{1}{m} \sum_{i=1}^m \langle \mathbf{g}_{b,i}, \theta_b - (\theta^\star + \Delta_{b,i}) \rangle = 0.$$

**Demonstrating Claim (iii).** The fact that $\|\theta_b - \theta^\star\|$ is bounded by claim (i) implies that $\theta_b$ is almost surely bounded too, so by (ii) there exists a subsequence $b_j \to \infty$

$$\theta_{b_j} \to \theta_\infty \qquad\qquad \frac{1}{m} \sum_{i=1}^m \langle \mathbf{g}_{b_j,i}, \theta_{b_j} - (\theta^\star + \Delta_{b_j,i}) \rangle \to 0.$$

We will now show that $\theta_\infty \in \Theta^\star$ by contradiction, so we assume that $\theta_\infty \notin \Theta^\star$ and argue that if this is the case

$$\limsup_{j\to\infty} \frac{1}{m} \sum_{i=1}^m \langle \mathbf{g}_{b_j,i}, \theta_{b_j} - (\theta^\star + \Delta_{b_j,i}) \rangle > 0,$$

which contradicts the condition above. The fact that our conformal predictors are regular implies that there exist infinitely many $k \geq 1$ such that

$$|q_{b_k,1}(\theta_\infty) - q_{b_k,1}^\star| \geq \epsilon$$

for some $\epsilon > 0$. We can now find $N$ large enough so that any $j \geq N$ satisfies $\|\theta_{b_j} - \theta_\infty\| \leq \frac{\epsilon}{2G}$ and by our Lipschitz assumption

$$|q_{b_j,1}(\theta_{b_j}) - q_{b_j,1}(\theta_\infty)| \leq \frac{\epsilon}{2}.$$

Therefore, for any of the infinitely many $k \geq N$ where $|q_{b_k,1}(\theta_\infty) - q_{b_k,1}^\star| \geq \epsilon$ we also have

$$|q_{b_k,i}(\theta_{b_k}) - q_{b_k,i}^\star| \geq \frac{\epsilon}{2}.$$

We now apply Lemma E.3 and our uniform local lower bound on the conditional densities for any such $k \geq N$ to conclude that there exist constants $c_1, c_2$ such that

$$\frac{1}{m} \sum_{i=1}^{m} \langle \mathbf{g}_{b_k,i}, \theta_{b_k} - (\theta^\star + \Delta_{b_k,i}) \rangle \geq \frac{1}{m} \langle \mathbf{g}_{b_k,1}, \theta_{b_k} - (\theta^\star + \Delta_{b_k,1}) \rangle$$

$$\geq \frac{1}{m} \left( L_{b_k,1}(\theta_{b_k}) - L_{b_k,1}(\theta^\star + \Delta_{b_k,1}) \right)$$

$$\geq \frac{1}{m} (c_1 \epsilon \wedge c_2 \epsilon^2) > 0.$$

This result contradicts our assumption that $\lim_{j \to \infty} \frac{1}{m} \sum_{i=1}^{m} \langle g_{b_j,i}, \theta_{b_j} - (\theta^\star + \Delta_{b_j,i}) \rangle = 0$, so $\theta_\infty \in \Theta^\star$. Finally, by (i) we know that almost surely $\|\theta_b - \theta^\star\|$ exists for all $\theta^\star \in \Theta^\star$, which includes $\theta^\star = \theta_\infty$ so $\|\theta_b - \theta_\infty\| \to 0$ and the entire sequence converges to $\theta_\infty \in \Theta^\star$ almost surely.

### C.3. Proof of Lemma 5.2

Since $\Theta$ is well-specified, there exists $\theta^\star \in \Theta$ such that $\theta^{\star\top} Z_{b,i} = q_{b,i}^\star$ is the conditional quantile of $S_{b,i} \mid \mathcal{F}_{b,i-1}$. Noting that $Z_{b,i}$ is measurable with respect to $\mathcal{F}_{b,i-1}$ and $|\theta^\top Z_{b,i} - \theta^{\star\top} Z_{b,i}| \leq GD$ by our Lipschtiz and finite diameter assumptions, we can use Lemma E.3, the lower bounds in the standard scalar case, to write

$$\frac{1}{m} \sum_{i=1}^{m} L_{b,i}(\theta) - L_{b,i}(\theta^\star) = \frac{1}{m} \sum_{i=1}^{m} \mathbb{E}[\ell_{\text{quantile}}(S_{b,i} - \theta^\top Z_{b,i}) - \ell_{\text{quantile}}(S_{b,i} - \theta^{\star\top} Z_{b,i}) \mid \mathcal{F}_{b,i-1}]$$

$$\geq \frac{1}{m} \sum_{i=1}^{m} \frac{p}{2} \min\left\{ \frac{\varepsilon}{GD}, 1 \right\} |\theta^\top Z_{b,i} - \theta^{\star\top} Z_{b,i}|^2.$$

This can further be lower bounded as:

$$\frac{1}{m} \sum_{i=1}^{m} L_{b,i}(\theta) - L_{b,i}(\theta^\star) \geq \frac{p}{2m} \min\left\{ \frac{\varepsilon}{GD}, 1 \right\} \sum_{i=1}^{m} (\theta - \theta^\star)^\top Z_{b,i} Z_{b,i}^\top (\theta - \theta^\star)$$

$$= \frac{p}{2} \min\left\{ \frac{\varepsilon}{GD}, 1 \right\} (\theta - \theta^\star)^\top \left( \frac{1}{m} \sum_{i=1}^{m} Z_{b,i} Z_{b,i}^\top \right) (\theta - \theta^\star)$$

$$\geq \frac{p\lambda_b}{2} \min\left\{ \frac{\varepsilon}{GD}, 1 \right\} \|\theta - \theta^\star\|_2^2$$

$$= \mu_b \|\theta - \theta^\star\|_2^2.$$

The second inequality follows from the fact that $x^\top A x \geq \lambda_{\min}(A) \|x\|^2$ for any symmetric matrix $A$ and with minimum eigenvalue $\lambda_{\min}(A) \in \mathbb{R}$, and any $x$. Note that $\lambda_b$ here is non-negative since its corresponding matrix is the average of positive semidefinite matrices. This completes the proof.

### C.4. Proof of Lemma 5.3

For simplicity, we consider only a single batch of $m$ observed covariate vectors $Z_i = (S_i, 1)$ for $i = 1, \ldots, m$, as in the assumed AR(1) process we have $S_i = \phi S_{i-1} + \varepsilon_i$, where $\varepsilon_i$ are i.i.d. with $\mathbb{E}[\varepsilon_i] = 0$. This is sufficient to show Lemma 5.3 since the proof for an arbitrary batch $b$ is identical after appropriately offsetting the indices of the scores. Define the matrix

$$\Sigma := \frac{1}{m} \sum_{i=1}^{m} Z_i Z_i^\top = \frac{1}{m} \sum_{i=1}^{m} \begin{bmatrix} S_i^2 & S_i \\ S_i & 1 \end{bmatrix}.$$

Let $\overline{S_m} = \frac{1}{m} \sum_{i=1}^{m} S_i$ and $\overline{S_m^2} = \frac{1}{m} \sum_{i=1}^{m} S_i^2$ be, respectively, the sample mean and second moment of the scores. We can relate the minimal eigenvalues of $\Sigma$ to first and second moments of $S_1^m$ via the following lemma:

**Lemma C.4.** *For the matrix $\Sigma = \frac{1}{m} \sum_{i=1}^{m} Z_i Z_i^\top$, we have*

$$\lambda_{\min}(\Sigma) \geq \left( 1 - \frac{\overline{S_m}^2}{\overline{S_m^2}} \right) \frac{\overline{S_m^2}}{1 + \overline{S_m^2}}.$$

Deferring the proof of Lemma C.4 temporarily, we see that to show Lemma 5.3, it suffices to show that both

(i) $\overline{S_m^2} \geq c_0 \sigma^2$ for a constant $c_0 > 0$

(ii) $\overline{S_m}^2 \leq c_1 \cdot \overline{S_m^2}$ for a constant $c_1 < 1$

with constant probability, as when these both occur, we have

$$\lambda_{\min}(\Sigma) \geq (1 - c_1) \frac{c_0 \sigma^2}{1 + c_0 \sigma^2}$$

with constant probability. For the remainder of the proof, we thus seek to demonstrate both of these, and in both cases, we assume $S_0 = s$ for a fixed scalar value $s \in \mathbb{R}$, as the process is independent of $(S_{-i})_{i \in \mathbb{N}}$ given $S_0$, and we condition implicitly on $S_0$ throughout.

**Demonstrating claim (i).** Writing $S_i = \phi S_{i-1} + \varepsilon_i$, we have

$$\sum_{i=1}^m S_i^2 = \sum_{i=1}^m (\phi S_{i-1} + \varepsilon_i)^2 \geq \sum_{i=1}^m \varepsilon_i^2 1\{\text{sign}(\varepsilon_i) = \text{sign}(S_{i-1})\}.$$

But of course, by assumption on the innovation process we have $\mathbb{P}([\varepsilon]_+ \geq \sigma) \geq \frac{1}{4}$ and $\mathbb{P}([-\varepsilon]_+ \geq \sigma) \geq \frac{1}{4}$, so that

$$\sum_{i=1}^m \varepsilon_i^2 1\{\text{sign}(\varepsilon_i) = \text{sign}(S_{i-1})\} \geq \frac{m}{4}(1 - o(1))\sigma^2$$

with high probability.[1] This demonstrates point (i).

**Demonstrating claim (ii).** This requires a substantially more tedious argument. We first develop explicit formulae for $S_i$ and their sums. Define the sequence of vectors

$$v_i := \begin{bmatrix} \phi^{i-1} & \phi^{i-2} & \cdots & 1 & 0 & \cdots & 0 \end{bmatrix}^\top \in \mathbb{R}_+^m.$$

Then we have

$$S_i = \sum_{j=1}^i \phi^{i-j} \varepsilon_j + \phi^i S_0 = \phi^i S_0 + v_i^T \varepsilon,$$

so that

$$\sum_{i=1}^m S_i = \phi \frac{1 - \phi^{m+1}}{1 - \phi} S_0 + \sum_{i=1}^m \frac{1 - \phi^{m-i+1}}{1 - \phi} \varepsilon_i \tag{5a}$$

and

$$\sum_{i=1}^m S_i^2 = \sum_{i=1}^m (\phi^i S_0 + v_i^T \varepsilon)^2 = \phi^2 \frac{1 - \phi^{2(m+1)}}{1 - \phi^2} S_0^2 + \varepsilon^T \left( \sum_{i=1}^m v_i v_i^T \right) \varepsilon + 2 S_0 \sum_{i=1}^m \phi^i v_i^T \varepsilon. \tag{5b}$$

We will control the deviations of both the expansions (5). For the former (5a), a quick calculation with Chebyshev's inequality shows that with probability at least $1 - 1/t^2$,

$$\left| \sum_{i=1}^m \frac{1 - \phi^{m-i+1}}{1 - \phi} \varepsilon_i \right| \leq O(1) t \sqrt{m} \frac{\sigma_\varepsilon}{1 - \phi}.$$

For the second term (5b), we control both the quadratic $\sum_{i=1}^m (v_i^T \varepsilon)^2$ and the final linear term.

**Claim C.3.** *Let $M = \sum_{i=1}^m v_i v_i^T$. Then $\mathbb{E}[\varepsilon^T M \varepsilon] = \sigma_\varepsilon^2 \operatorname{tr}(M)$, and there is a numerical constant $p > 0$ such that for all $t \in [0, 1]$,*

$$\mathbb{P}\left(\varepsilon^T M \varepsilon \geq t \sigma_\varepsilon^2 \operatorname{tr}(M)\right) \geq (1 - t)^2 \cdot p.$$

---

[1] Making this rigorous would use Azuma's inequality.

*Proof.* The expectation calculation is trivial. To lower bound the probabilities, note that by the Paley-Zygmund inequality, for any $t \in [0, 1]$,

$$\mathbb{P}(\varepsilon^T M \varepsilon \geq t \sigma_\varepsilon^2 \operatorname{tr}(M)) \geq (1 - t)^2 \frac{\sigma_\varepsilon^4 \operatorname{tr}(M)^2}{\mathbb{E}[(\varepsilon^T M \varepsilon)^2]},$$

so that the claimed constant $p = \frac{\sigma_\varepsilon^4 \operatorname{tr}(M)^2}{\mathbb{E}[(\varepsilon^T M \varepsilon)^2]}$. We now demonstrate that $p$ is indeed a (numerical) constant under the assumptions of Lemma 5.3. To control the denominator, the expansion $\varepsilon^T M \varepsilon = \sum_{i,j} \varepsilon_i \varepsilon_j M_{ij}$ and that the $\varepsilon_i$ are are i.i.d. and mean zero together imply

$$\mathbb{E}[(\varepsilon^T M \varepsilon)^2] = \sum_{i=1}^m M_{ii}^2 \mathbb{E}[\varepsilon_i^4] + O(1) \sum_{i \neq j} M_{ii} M_{jj} \mathbb{E}[\varepsilon_i^2 \varepsilon_j^2] + O(1) \sum_{i \neq j} M_{ij}^2 \mathbb{E}[\varepsilon_i^2 \varepsilon_j^2]$$

$$= O(1) \sigma_\varepsilon^4 \left( \operatorname{tr}(M)^2 + \|M\|_{\mathrm{Fr}}^2 \right).$$

Because $v_t$ has entries $v_{t,i} = \phi^{t-i}$ for $i \leq t$ and $v_{t,i} = 0$ otherwise,

$$M_{ij} = \sum_{t=1}^m v_{t,i} v_{t,j} = \sum_{t=i \vee j}^m \phi^{2t-i-j} = \phi^{-i-j} \sum_{t=i \vee j}^m \phi^{2t} = \phi^{2(i \vee j)-i-j} \frac{1 - \phi^{2(i \vee j+1)}}{1 - \phi^2}.$$

So $\operatorname{tr}(M) = \sum_{i=1}^m \frac{1 - \phi^{2(i+1)}}{1 - \phi^2} \gtrsim \frac{m}{1 - \phi^2}$, while by considering the diagonal, one-off diagonal, two-off-diagonal, and so on, $\|M\|_{\mathrm{Fr}}^2 \lesssim \sum_{i=0}^{m-1} (m - i) \phi^{2i} \frac{1}{(1 - \phi^2)^2}$. The following observation allows us to control such sums:

**Observation C.1.** *Let $c \neq 1$. Then for any $m \in \mathbb{N}$,*

$$\sum_{i=0}^{m-1} (m - i) c^i = m \frac{1 - c^m}{1 - c} + \frac{(m+1)c^m - 1}{1 - c} - \frac{c - c^{m+1}}{(1 - c)^2} + \frac{1 - c^m}{1 - c}.$$

*Proof.* We observe that $\sum_{i=0}^{m-1} c^i = \frac{1 - c^m}{1 - c}$ whenever $c \neq 1$, and so

$$\sum_{i=0}^{m-1} (m - i) c^i = m \frac{1 - c^m}{1 - c} - \sum_{i=0}^{m-1} (i + 1) c^i + \sum_{i=0}^{m-1} c^i$$

$$= m \frac{1 - c^m}{1 - c} - \frac{\partial}{\partial c} \sum_{i=0}^{m-1} c^{i+1} + \frac{1 - c^m}{1 - c}$$

$$= m \frac{1 - c^m}{1 - c} - \frac{\partial}{\partial c} \frac{c - c^{m+1}}{1 - c} + \frac{1 - c^m}{1 - c}.$$

Computing derivatives gives the result. $\qquad \square$

Returning to our chain of inequalities and substituting $c = \phi^2$ in the observation, we thus obtain

$$\|M\|_{\mathrm{Fr}}^2 \lesssim \frac{m}{(1 - \phi^2)^3} \ll \operatorname{tr}(M)^2,$$

and so the constant $p = \frac{\operatorname{tr}(M)^2}{O(1)(\operatorname{tr}(M)^2 + \|M\|_{\mathrm{Fr}}^2)} \gtrsim 1$ as desired. $\qquad \square$

Finally, we control the final sum in the expansion (5b). Defining the scalars

$$u_i = \frac{1}{1 - \phi} \phi^i \left( 1 - \phi^{m-i+1} \right),$$

so for the vector $u = (u_1, \ldots, u_m)$ we observe that

$$\sum_{i=1}^m \phi^i v_i^T \varepsilon = \sum_{i=1}^m u_i \varepsilon_i = u^T \varepsilon,$$

and so

$$\mathbb{E}\left[\left(\sum_{i=1}^{m}\phi^i v_i^T \varepsilon\right)^4\right] = \mathbb{E}[(u^T\varepsilon)^4] = \sum_{i=1}^{m} u_i^4 \mathbb{E}[\varepsilon_i^4] + O(1)\sum_{i\neq j} u_i^2 u_j^2 \mathbb{E}[\varepsilon_i^2\varepsilon_j^2] = O(1)\sigma_\varepsilon^4\left(\|u\|_4^4 + \|u\|_2^4\right).$$

Because $\|u\|_4^4 \leq \frac{\phi^4}{(1-\phi)^4(1-\phi^4)}$ and $\|u\|_2^2 \leq \frac{\phi^2}{(1-\phi)^2(1-\phi^2)}$, we obtain that

$$\mathbb{P}\left(\left|\sum_{i=1}^{m}\phi^i v_i^T \varepsilon\right| \geq t\sigma_\varepsilon\right) \leq \frac{\phi^4}{(1-\phi)^4(1-\phi^2)^2}\cdot\frac{O(1)}{t^4}$$

for all $t \geq 0$. Summarizing, in pursuit of proving claim (ii), we have shown

**Claim C.4.** *There is a numerical constant $p > 0$ such that for any $t_a, t_b \geq 0$, with probability at least $p - 1/t_a^2 - 1/t_b^4$,*

$$\left|\sum_{i=1}^{m} S_i - \phi\frac{1-\phi^{m+1}}{1-\phi}S_0\right| \leq O(1)t_a\sqrt{m}\frac{\sigma_\varepsilon}{1-\phi}$$

*and*

$$\sum_{i=1}^{m} S_i^2 \geq \phi^2\frac{1-\phi^{2(m+1)}}{1-\phi^2}S_0^2 + \frac{\sigma_\varepsilon^2}{2}\operatorname{tr}\left(\sum_{i=1}^{m} v_i v_i^\top\right) - O(1)S_0 t_b\sigma_\varepsilon\frac{\phi}{(1-\phi)\sqrt{1-\phi^2}}.$$

By claim C.4, we may evidently choose $t_a, t_b$ to be appropriate constants and see that with (numerical) constant probability,

$$\overline{S_m}^2 \lesssim \frac{\phi^2}{m^2(1-\phi)^2}S_0^2 + \frac{\sigma_\varepsilon^2}{m(1-\phi)^2} \quad\text{and}\quad \overline{S_m^2} \gtrsim \frac{\phi^2}{m}\frac{1}{1-\phi^2}S_0^2 + \frac{\sigma_\varepsilon^2}{1-\phi^2} - \frac{S_0\sigma_\varepsilon\phi}{m(1-\phi)\sqrt{1-\phi^2}}.$$

We consider two cases: depending on whether $S_0^2 \geq m\sigma_\varepsilon^2/\phi^2$. In the first case (that it is larger), we see that $\overline{S_m}^2 \lesssim \frac{\phi^2}{m^2(1-\phi)^2}S_0^2$ while

$$\overline{S_m^2} \gtrsim \frac{\phi^2 S_0^2}{m(1-\phi^2)} - \frac{S_0^2\phi^2}{m^{3/2}(1-\phi)\sqrt{1-\phi^2}}.$$

That $\frac{1}{(1-\phi)^2} \leq \frac{1}{1-\phi^2}$ immediately shows $\overline{S_m^2} \gtrsim \overline{S_m}^2$. In the latter case that $S_0^2 < m\sigma_\varepsilon^2/\phi^2$, we have $\overline{S_m}^2 \lesssim \frac{\sigma_\varepsilon^2}{m(1-\phi)^2}$ while by the Fenchel-Young inequality $ab \leq \frac{a^2}{2} + \frac{b^2}{2}$,

$$\overline{S_m^2} \gtrsim \frac{\phi^2 S_0^2}{m(1-\phi^2)} + \frac{\sigma_\varepsilon^2}{1-\phi} - \frac{S_0^2\phi^2}{2m(1-\phi^2)} - \frac{\sigma_\varepsilon^2}{2m(1-\phi)^2} \overset{(\star)}{\geq} \frac{\phi^2 S_0^2}{2m(1-\phi^2)} + \frac{\sigma_\varepsilon^2}{2m(1-\phi)^2},$$

where inequality $(\star)$ used that $m \geq \frac{1}{1-\phi}$. In particular, we have $\overline{S_m^2} \gtrsim \overline{S_m}^2$, as desired.

Lastly, we return to prove Lemma C.4:

*Proof.* We explicitly find the eigenvalues of a two-by-two matrix. For scalars $a, b$ and matrix

$$A = \begin{bmatrix} a & b \\ b & 1 \end{bmatrix} \quad\text{we have}\quad \det(A - \lambda I) = (a-\lambda)(1-\lambda) - b^2 = \lambda^2 - (a+1)\lambda + a - b^2.$$

Solving for $\det(A - \lambda I) = 0$, we obtain eigenvalues

$$\lambda = \frac{a + 1 \pm \sqrt{(a-1)^2 + 4b^2}}{2}.$$

Now, let us assume that $b^2 = (1-\gamma)a$ for some $\gamma \in [0,1]$. Then using the first-order concavity of the square root (that is, that $\sqrt{x+\Delta} \leq \sqrt{x} + \frac{\Delta}{2\sqrt{x}}$), we obtain

$$2\lambda_{\min}(A) = a + 1 - \sqrt{(a-1)^2 + 4(1-\gamma)a} = a + 1 - \sqrt{(a+1)^2 - 4\gamma a} \geq \frac{2\gamma a}{a+1}.$$

Substituting $a = \overline{S_m^2}$ and $b = \overline{S_m}$, which satisfy $b^2 \leq a$ by Jensen's inequality, on the event that $(1-\gamma)\overline{S_m^2} \geq \overline{S_m}^2$ we have $\lambda_{\min}(\Sigma) \geq \frac{\gamma\overline{S_m^2}}{1+\overline{S_m^2}}$. Noting that $(1-\gamma)\overline{S_m^2} \geq \overline{S_m}^2$ if and only if $\gamma \leq 1 - \frac{\overline{S_m}^2}{\overline{S_m^2}}$ gives the lemma. $\qquad\square$

## C.5. Proof of Theorem 5.4

We first introduce some new notation regarding gradients for ease of presentation. We denote the conditional subgradient by $\mathbf{g}_{b,i}(\theta) \in \partial_\theta L_{b,i}(\theta)$ and the observed subgradient by $\hat{\mathbf{g}}_{b,i}(\theta) \in \partial_\theta \ell_{\text{quantile}}(S_{b,i} - \theta^\top Z_{b,i})$, dropping the argument $\theta$ when it is clear from context. We denote the batch expected subgradient *conditioned on previous batches*, as $\mathbf{g}_b = \mathbb{E}[\frac{1}{m}\sum_{i=1}^m \hat{\mathbf{g}}_{b,i} \mid \mathcal{F}_{b-1,m}]$, the batch empirical subgradient as $\hat{\mathbf{g}}_b := \frac{1}{m}\sum_{i=1}^m \hat{\mathbf{g}}_{b,i}$, and define the difference $\mathbf{z}_b = \mathbf{g}_b - \hat{\mathbf{g}}_b$.

As in the proof of asymptotic consistency of Theorem 5.1, we start by establishing the connection between the conditional coverage guarantee (2) and the conformal predictor's parameter $\theta \in \Theta$. The following proposition states that the conditional coverage error can be bounded by the distance of $\theta$ to the optimal $\theta^\star \in \Theta$ that represents all conditional quantiles under well-specification.

**Proposition C.5.** *Let $S_{b,i} \mid \mathcal{F}_{b,i-1}$ have density $f_{b,i}$ upper-bounded by $u'$ in a $GD$-neighborhood around its conditional quantile $q_{b,i}^\star$. Then,*

$$\left| \mathbb{P}(Y_{b,i} \in C_{b,i}(X_{b,i}) \mid \mathcal{F}_{b,i-1}) - (1-\alpha) \right| \leq u'G \left\| \theta_b - \theta^\star \right\|.$$

Deferring the proof of Proposition C.5 temporarily, we see that to prove Theorem 5.4 it only remains to show that Algorithm 1's parameters $\theta_b \in \Theta$ satisfy $\|\theta_b - \theta^\star\|^2 \in O(\log(B\log(B)/\delta)/b^c)$ with high probability. We do this through the following two claims:

(i) For any $\delta > 0$ there exist recursive upper bounds $\|\theta_{b+1} - \theta^\star\|^2 \leq \sqrt{\sum_{i=1}^b A_{b,i} \|\theta_i - \theta^\star\|^2} + R_b$ that hold simultaneously for all $b \leq B$ with probability at least $1 - \delta$.

(ii) Under the event in (i), $\sqrt{\sum_{i=1}^b A_{b,i} \|\theta_i - \theta^\star\|^2} + R_b \in O(\log(B\log(B)/\delta)/b^c)$ by strong induction.

Substituting our bound on $\|\theta_b - \theta^\star\|^2$, we get that for some constant $C_b$, which is a function of $G, \mu_{\min}, u'$,

$$\left| \mathbb{P}(Y_{b,i} \in C_{b,i}(X_{b,i}) \mid \mathcal{F}_{b,i-1}) - (1-\alpha) \right| \leq C_b \sqrt{\frac{(\log(B\log(B)/\delta))}{b^c}}.$$

Without batching, as in the statement of Theorem 5.4 for different constant $C$, a function of $m, G, \mu_{\min}, u'$, we can write

$$\left| \mathbb{P}(Y_t \in C_t(X_t) \mid \mathcal{F}_{t-1}) - (1-\alpha) \right| \leq C \sqrt{\frac{(\log(T\log(T)/\delta))}{t^c}}.$$

We now turn to the proof, which borrows several parts from Proposition 1 of Rakhlin et al. (2012). The novelty lies in the batch setting, the changing objective function, and the generalized learning rates.

**Demonstrating Claim (i).** We start by unrolling one iteration of Algorithm 1, using the fact that $\theta_{b+1} = \Pi_\Theta(\theta_b - \eta_b\hat{\mathbf{g}}_b)$:

$$\begin{aligned}
\|\theta_{b+1} - \theta^\star\|^2 &\leq \|\theta_b - \eta_b\hat{\mathbf{g}}_b - \theta^\star\|^2 \\
&= \|\theta_b - \theta^\star\|^2 - 2\eta_b \langle \hat{\mathbf{g}}_b, \theta_b - \theta^\star \rangle + \eta_b^2 \|\hat{\mathbf{g}}_b\|^2 \\
&\leq \|\theta_b - \theta^\star\|^2 - 2\eta_b \langle \hat{\mathbf{g}}_b, \theta_b - \theta^\star \rangle + 2\eta_b \langle \mathbf{g}_b, \theta_b - \theta^\star \rangle - 2\eta_b \langle \mathbf{g}_b, \theta_b - \theta^\star \rangle + \eta_b^2 G^2 \\
&= \|\theta_b - \theta^\star\|^2 - 2\eta_b \frac{1}{m}\sum_{i=1}^m \mathbb{E}[\langle \mathbf{g}_{b,i}, \theta_b - \theta^\star \rangle \mid \mathcal{F}_{b-1,m}] + 2\eta_b \langle \mathbf{z}_b, \theta_b - \theta^\star \rangle + \eta_b^2 G^2.
\end{aligned}$$

This expression can now be upper bounded using the subgradient inequality $\langle \mathbf{g}_{b,i}, \theta_b - \theta^\star \rangle \geq L_{b,i}(\theta_b) - L_{b,i}(\theta^\star)$ and Lemma 5.2 due to well-specification:

$$\begin{aligned}
\|\theta_{b+1} - \theta^\star\|^2 &\leq \|\theta_b - \theta^\star\|^2 - 2\eta_b \frac{1}{m}\sum_{i=1}^m \mathbb{E}[(L_{b,i}(\theta_b) - L_{b,i}(\theta^\star)) \mid \mathcal{F}_{b-1,m}] + 2\eta_b \langle \mathbf{z}_b, \theta_b - \theta^\star \rangle + \eta_b^2 G^2 \\
&\leq \|\theta_b - \theta^\star\|^2 - 2\eta_b \|\theta_b - \theta^\star\|^2 \mathbb{E}[\mu_b \mid \mathcal{F}_{b-1,m}] + 2\eta_b \langle \mathbf{z}_b, \theta_b - \theta^\star \rangle + \eta_b^2 G^2.
\end{aligned}$$

We now use the assumption that $\mathbb{E}\left[\mu_b \mid \mathcal{F}_{b-1,m}\right]$ is uniformly lower bounded by $\mu_{\min} > 0$ with probability 1 (as in Lemma 5.3), and our choice of step size $\eta_b = \frac{2^{c-1}}{b^c \mu_{\min}}$ to argue that

$$\|\theta_{b+1} - \theta^\star\|^2 \leq \|\theta_b - \theta^\star\|^2 - 2\eta_b \|\theta_b - \theta^\star\|^2 \mu_{\min} + 2\eta_b \langle \mathbf{z}_b, \theta_b - \theta^\star \rangle + \eta_b^2 G^2$$
$$= (1 - 2\eta_b \mu_{\min}) \|\theta_b - \theta^\star\|^2 + 2\eta_b \langle \mathbf{z}_b, \theta_b - \theta^\star \rangle + \eta_b^2 G^2$$
$$\leq \left(1 - \left(\frac{2}{b}\right)^c\right) \|\theta_b - \theta^\star\|^2 + 2\eta_b \langle \mathbf{z}_b, \theta_b - \theta^\star \rangle + \eta_b^2 G^2.$$

Unrolling this recursion until $b = 2$, we obtain:

$$\|\theta_{b+1} - \theta^\star\|^2 \leq \prod_{i=2}^{b} \left(1 - \left(\frac{2}{i}\right)^c\right) \|\theta_2 - \theta^\star\|^2 + \sum_{i=2}^{b} \prod_{j=i+1}^{b} \left(1 - \left(\frac{2}{j}\right)^c\right) 2\eta_i \langle \mathbf{z}_i, \theta_i - \theta^\star \rangle + \sum_{i=2}^{b} \prod_{j=i+1}^{b} \left(1 - \left(\frac{2}{j}\right)^c\right) \eta_i^2 G^2$$
$$\leq \frac{2^c}{\mu_{\min}} \sum_{i=2}^{b} \prod_{j=i+1}^{b} \left(1 - \left(\frac{2}{j}\right)^c\right) \frac{1}{i^c} \langle \mathbf{z}_i, \theta_i - \theta^\star \rangle + \frac{G^2}{\mu_{\min}^2} \sum_{i=2}^{b} \prod_{j=i+1}^{b} \left(1 - \left(\frac{2}{j}\right)^c\right) \frac{1}{i^{2c}},$$

where the first term vanishes since $\left(\frac{2}{i}\right)^c = 1$ at $i = 2$. Note that the sum in the second term can be bounded by $\frac{1}{b^c}$ using Lemma E.6 so that using the simplified notation $f_b(i) := \prod_{j=i+1}^{b} \left(1 - \left(\frac{2}{j}\right)^c\right) \frac{1}{i^c}$,

$$\|\theta_{b+1} - \theta^\star\|^2 \leq \frac{2^c}{\mu_{\min}} \underbrace{\sum_{i=2}^{b} f_b(i) \langle \mathbf{z}_i, \theta_i - \theta^\star \rangle}_{(\star)} + \frac{G^2}{\mu_{\min}^2 b^c}.$$

The upper bound in its current form does not match Claim (i) as it depends on $\langle \mathbf{z}_i, \theta_i - \theta^\star \rangle$ rather than the norms $\|\theta_i - \theta^\star\|$. We bridge this gap in the following proposition using the fact that $(\star)$ is a martingale that we can bound with high probability.

**Proposition C.6.** *Let $M_b(i) := f_b(i) \langle \mathbf{z}_i, \theta_i - \theta^\star \rangle$ and $\delta > 0$. Then $M_b(i)$ is a martingale difference sequence and with probability at least $1 - \delta$*

$$\sum_{i=2}^{b} M_b(i) \leq 8G \max \left\{ \sqrt{\sum_{i=2}^{b} f_b(i)^2 \|\theta_i - \theta^\star\|^2}, \frac{Gf_b(b)}{\mu_{\min}} \sqrt{\log\left(\frac{B\log(B)}{\delta}\right)} \right\} \sqrt{\log\left(\frac{B\log(B)}{\delta}\right)},$$

*for all $b \leq B$ simultaneously.*

*Proof.* Our sequence $M_b(i)$ forms a martingale difference sequence since

$$\mathbb{E}[\hat{\mathbf{g}}_i - \mathbf{g}_i \mid \mathcal{F}_{i-1,m}] = \mathbb{E}\left[\frac{1}{m} \sum_{j=1}^{m} \hat{\mathbf{g}}_{i,j} - \mathbb{E}\left[\frac{1}{m} \sum_{j=1}^{m} \hat{\mathbf{g}}_{i,j} \mid \mathcal{F}_{i-1,m}\right] \mid \mathcal{F}_{i-1,m}\right] = 0,$$

$f_b(i)$ is constant, and $\theta_i$ is measurable with respect to $\mathcal{F}_{i-1,m}$. The conditional variance, using the fact that $\|\mathbf{z}_i\| \leq 2G$, can be bounded by

$$\mathrm{Var}(M_b(i) \mid \mathcal{F}_{i-1,m}) \leq 4f_b(i)^2 G^2 \|\theta_i - \theta^\star\|^2,$$

so that the sum of the conditional variances from $M_b(2)$ to $M_b(b)$ satisfies

$$\sum_{i=2}^{b} \mathrm{Var}(M_b(i) \mid \mathcal{F}_{i-1,m}) \leq 4G^2 \sum_{i=2}^{b} f_b(i)^2 \|\theta_i - \theta^\star\|^2.$$

We also have the uniform bound (since $f_b$ is increasing by arguments in the inductive step of Lemma E.4)

$$|M_b(i)| \leq f_b(b) 2G \|\theta_i - \theta^\star\| \leq \frac{2G^2 f_b(b)}{\mu_{\min}},$$

where the upper-bound $\|\theta_i - \theta^\star\| \leq \frac{G}{\mu_{\min}}$ follows from the fact that

$$G \|\theta_i - \theta^\star\| \geq \|\mathbf{g}_i\| \|\theta_i - \theta^\star\| \geq \langle \mathbf{g}_i, \theta_i - \theta^\star \rangle \geq \mathbb{E}\left[\mu_i \mid \mathcal{F}_{i-1,m}\right] \|\theta_i - \theta^\star\|^2 \geq \mu_{\min} \|\theta_i - \theta^\star\|^2. \tag{6}$$

Now, we can apply Lemma E.2 from Bartlett et al. (2008) to bound the sum of martingale differences $\sum_{i=2}^{b} M_b(i)$. As $b$ increases from 2 to $B$, we have $(B-1)$ martingales, which we will bound together with a union bound. As long as $B \geq 4$ and $\delta' \in (0, 1/e)$, with probability at least $1 - B\delta'$, for all $b \leq B$,

$$\sum_{i=2}^{b} M_b(i) \leq 2 \max \left\{ 2\sqrt{4G^2 \sum_{i=2}^{b} f_b(i)^2 \|\theta_i - \theta^\star\|^2}, \frac{2G^2 f_b(b)}{\mu_{\min}} \sqrt{\log\left(\frac{\log(B)}{\delta'}\right)} \right\} \sqrt{\log\left(\frac{\log(B)}{\delta'}\right)}$$

$$\leq 8G \max \left\{ \sqrt{\sum_{i=2}^{b} f_b(i)^2 \|\theta_i - \theta^\star\|^2}, \frac{G f_b(b)}{\mu_{\min}} \sqrt{\log\left(\frac{\log(B)}{\delta'}\right)} \right\} \sqrt{\log\left(\frac{\log(B)}{\delta'}\right)}.$$

Substituting $\delta' = \frac{\delta}{B}$ yields the stated result. $\qquad\square$

Summarizing, with the goal of establishing a $O(\log(B \log(B)/\delta)/b^c)$ rate for $\|\theta_b - \theta^\star\|^2$, we have shown that for all $b \leq B$, with probability at least $1 - \delta$,

$$\|\theta_{b+1} - \theta^\star\|^2$$

$$\leq \frac{2^c}{\mu_{\min}} 8G \max \left\{ \sqrt{\sum_{i=2}^{b} f_b(i)^2 \|\theta_i - \theta^\star\|^2}, \frac{G f_b(b)}{\mu_{\min}} \sqrt{\log\left(\frac{B \log(B)}{\delta}\right)} \right\} \sqrt{\log\left(\frac{B \log(B)}{\delta}\right)} + \frac{G^2}{\mu_{\min}^2 b^c}$$

$$\leq \frac{16G\sqrt{\log(B \log(B)/\delta)}}{\mu_{\min}} \sqrt{\sum_{i=2}^{b} f_b(i)^2 \|\theta_i - \theta^\star\|^2} + \frac{16G^2 \log(B \log(B)/\delta)}{\mu_{\min}^2 b^c} + \frac{G^2}{\mu_{\min}^2 b^c}$$

$$= \frac{16G\sqrt{\log(B \log(B)/\delta)}}{\mu_{\min}} \sqrt{\sum_{i=2}^{b} f_b(i)^2 \|\theta_i - \theta^\star\|^2} + \frac{G^2(16 \log(B \log(B)/\delta) + 1)}{\mu_{\min}^2 b^c}.$$

This is equivalent to Claim (i).

**Demonstrating Claim (ii).** We now prove by strong induction that $\|\theta_b - \theta^\star\|^2 \leq \frac{a}{b^c}$ for suitably chosen $a$. This holds for $b = 1, 2$ if $a \geq \frac{2G^2}{\mu_{\min}^2}$, since that implies $\|\theta_b - \theta^\star\|^2 \leq \frac{G^2}{\mu_{\min}^2}$, which holds by inequality (6). To show the inductive step, we write the above inequality as

$$\|\theta_{b+1} - \theta^\star\|^2 \leq x \sqrt{\sum_{i=2}^{b} f_b(i)^2 \|\theta_i - \theta^\star\|^2} + \frac{y}{b^c},$$

for $x = \frac{16G\sqrt{\log(B \log(B)/\delta)}}{\mu_{\min}}$ and $y = \frac{G^2(16 \log(B \log(B)/\delta) + 1)}{\mu_{\min}^2}$. By the inductive hypothesis, $\|\theta_i - \theta^\star\|^2 \leq \frac{a}{i^c}$ for $i = 1, \ldots b$. To show that $\|\theta_{b+1} - \theta^\star\|^2 \leq \frac{a}{(b+1)^c}$, it suffices to find $a$ sufficiently large so that

$$x \sqrt{\sum_{i=2}^{b} f_b(i)^2 \frac{a}{i^c}} + \frac{y}{b^c} \leq \frac{a}{(b+1)^c}.$$

By Lemma E.7, which bounds $\sum_{i=2}^{b} f_b(i)^2 \frac{1}{i^c}$

$$x\sqrt{a} \cdot \frac{1}{b^c} + \frac{y}{b^c} \leq \frac{a}{(b+1)^c} \Leftrightarrow (x\sqrt{a} + y)\left(\frac{b+1}{b}\right)^c \leq a.$$

Using the fact that $\left(\frac{b+1}{b}\right)^c \leq \frac{b+1}{b} \leq \frac{3}{2}$ for $b \geq 2$ and $c \in [0,1]$, then

$$a \geq \frac{3}{2}(x\sqrt{a} + y) \Leftrightarrow a - \frac{3x}{2}\sqrt{a} - \frac{3y}{2} \geq 0.$$

Solving the quadratic inequality, we get that

$$\sqrt{a} \geq \frac{1}{2}\left(\frac{3x}{2} + \sqrt{\frac{9x^2}{4} + 6y}\right) \Longleftarrow a \geq \frac{1}{2}\left(\frac{9x^2}{4} + \frac{9x^2}{4} + 6y\right) = \frac{9x^2}{4} + 3y.$$

Substituting in the values of $x$ and $y$, we get that

$$a \geq \frac{576G^2 \log(B\log(B)/\delta)}{\mu_{\min}^2} + \frac{3G^2(16\log(B\log(B)/\delta) + 1)}{\mu_{\min}^2} \Longleftarrow a \geq \frac{G^2(624\log(B\log(B)/\delta) + 3)}{\mu_{\min}^2}.$$

Note that this satisfies the condition on $a$ in the base case, that is $a \geq \frac{2G^2}{\mu_{\min}^2}$, since $\frac{3G^2}{\mu_{\min}^2} \geq \frac{2G^2}{\mu_{\min}^2}$. This shows Claim (ii) and the first statement of the theorem.

Lastly, we return to prove Proposition C.5:

*Proof.* We use our assumption that $S_t \mid \mathcal{F}_{t-1}$ has density $f_t$ upper-bounded by $u'$ in a $GD$ neighborhood around the conditional quantile $q_t^\star$, so that

$$
\begin{aligned}
\left|\mathbb{P}(Y_{b,i} \in C_{b,i}(X_{b,i}) \mid \mathcal{F}_{b,i-1}) - (1-\alpha)\right| &= \left|\mathbb{P}(q_{b,i}(\theta_b) \geq S_{b,i} \mid \mathcal{F}_{b,i-1}) - \mathbb{P}(q_{b,i}(\theta^\star) \geq S_{b,i} \mid \mathcal{F}_{b,i-1})\right| \\
&= \max\left\{\int_{q_{b,i}(\theta_b)}^{q_{b,i}(\theta^\star)} f_{b,i}(x)dx, \int_{q_{b,i}(\theta^\star)}^{q_{b,i}(\theta_b)} f_{b,i}(x)dx\right\} \\
&\leq u'|q_{b,i}(\theta_b) - q_{b,i}(\theta^\star)| \\
&\leq u'G\|\theta_b - \theta^\star\|.
\end{aligned}
$$

$\square$

## C.6. Proof of Theorem 6.1

We first provide a roadmap for the proof, and then the details for each step.

For our iterates $\theta_t$ to converge to $\theta^\star$ which defines the best quantile predictor, we must first establish the existence and uniqueness of the minimizer $\theta^\star$. We will argue that under our assumptions, this is a consequence of the strong convexity of $L_\Pi(\theta)$ around its minimizer.

**Claim C.5.** *Let the assumptions of Theorem 6.1 hold. Then:*

(i) *There exists $\theta^\star \in \Theta$ such that $L_\Pi(\theta^\star) \leq L_\Pi(\theta)$ for any other $\theta \in \Theta$.*

(ii) *For any $\theta \in \Theta$ and $\mu = p_\Pi \lambda \min\left\{1, \frac{\varepsilon}{GD}\right\}$*

$$L_\Pi(\theta) - L_\Pi(\theta^\star) \geq \frac{\mu}{2}\|\theta - \theta^\star\|^2,$$

*so $\theta^\star$ is unique.*

Claim C.5 not only states the existence and uniqueness of $\theta^\star$, but also the fact that $L_\Pi$ behaves like a strongly convex function with respect to its minimizer. This allows us to bound the squared distance $\|\theta_t - \theta^\star\|^2$ using a recursion argument based on Proposition 1 of Rakhlin et al. (2012):

**Claim C.6.** *The iterates $\theta_t$ of Algorithm 1 with $m = 1$ and $\eta_t = \frac{2^c}{\mu t^c}$ satisfy*

$$\|\theta_{t+1} - \theta^\star\|^2 \leq \sum_{i=2}^{t} \prod_{j=i+1}^{t} \left(1 - \left(\frac{2}{j}\right)^c\right) 2\eta_i (L_\Pi(\theta_i) - L_\Pi(\theta^\star) - \ell_i(\theta_i) + \ell_i(\theta^\star))$$

$$+ \sum_{i=2}^{t} \prod_{j=i+1}^{t} \left(1 - \left(\frac{2}{j}\right)^c\right) \eta_i^2 \|\hat{\mathbf{g}}_i\|^2.$$

To prove the first part of Theorem 6.1 it now suffices to control the expected value of the bound in Claim C.6. Deriving a bound for the rightmost term is straightforward from our Lipschitz assumption and the learning rate inequality in Lemma E.6. To control the remaining term we rely on the following lemma.

**Lemma C.7.** *Let $\{\nu_2, \cdots, \nu_t\}$ be any sequence of positive scalars and*

$$A = \sum_{i=2}^{t} \nu_i (L_\Pi(\theta_i) - L_\Pi(\theta^\star) - \ell_i(\theta_i) + \ell_i(\theta^\star)),$$

*then for $0 < \tau < t$*

$$\mathbb{E}[A] \leq GD \left(\sum_{i=2}^{t-\tau} \left[\nu_i \cdot \frac{\beta(\tau)}{2} + |\nu_i - \nu_{i+\tau}| + \frac{\nu_i \tau \eta_i G}{D}\right] + \sum_{i=2}^{\tau} \nu_i + \sum_{i=t-\tau+1}^{t} 2\nu_i\right).$$

*The expectation is taken over the samples $\{(S_i, Z_i)\}_{i=1}^{t}$.*

Combining the bounds for both terms now yields the first part of Theorem 6.1, which we summarize below.

**Proposition C.8.** *Based on the bound in Claim C.6, the iterates $\theta_t$ of Algorithm 1 with $m = 1$ and $\eta_t = \frac{2^c}{\mu t^c}$ satisfy*

$$\mathbb{E}\left[\|\theta_{t+1} - \theta^\star\|^2\right] \leq C \left(\frac{G^2}{\mu^2} + \frac{GD}{\mu}\right) \frac{\tau}{(t+1)^c},$$

*for $t \in [T+1]$ and a universal constant $C$.*

The final step is to link this upper bound to the expected coverage gap via the following lemma.

**Lemma C.9.** *Let $S_t \mid \mathcal{F}_{t-1}$ have density $f_t$ upper-bounded by $u$ in a $\varepsilon$-neighborhood around $Z_t^\top \theta^\star$. Then,*

$$\mathbb{E}\left[|\mathbb{P}\left(Y_t \notin C_t(X_t) \mid \mathcal{F}_{t-1}\right) - \alpha_t|\right] \leq \left(\frac{G}{\varepsilon} + u_\Pi G\right) \sqrt{\mathbb{E}\left[\|\theta_{t+1} - \theta^\star\|^2\right]}.$$

We now provide proofs for each of these results, with the proof of Lemma C.7 deferred to the end of the section.

**Demonstrating Claim C.5.** Part (i) is a straightforward consequence of the extreme value theorem since $\Theta$ is closed and bounded and $L_\Pi(\theta)$ is continuous, so there exists $\theta^\star \in \Theta$ such that $L_\Pi(\theta^\star) \leq L_\Pi(\theta)$ for any other $\theta \in \Theta$. Part (ii) requires a more detailed analysis that we split into 3 sub-parts:

(ii)(a) The Hessian $\nabla^2 L_\Pi(\theta)$ exists for all $\theta \in \mathbb{B}_2(\theta^\star, \frac{\varepsilon}{G})$.

(ii)(b) $L_\Pi$ is $p_\Pi \lambda$-strongly convex in $\mathbb{B}_2(\theta^\star, \frac{\varepsilon}{G})$ since for any $\theta$ in this set the Hessian satisfies

$$\nabla^2 L_\Pi(\theta) \succeq p_\Pi \lambda I.$$

(ii)(c) Part (b) and the finite diameter of $\Theta$ imply part (ii) of Claim C.5.

**Demonstrating (ii)(a).** Recall that by the chain rule for subdifferentials

$$\nabla L_\Pi(\theta) = \mathbb{E}_\Pi \left[ Z(\mathbb{P}\left(S \le Z^\top \theta \mid Z\right) - (1 - \alpha)) \right].$$

For any $z$ and $i \in [d]$ we now define the auxiliary function

$$g_i(\theta; z) = z_i(\mathbb{P}\left(S \le z^\top \theta \mid Z = z\right) - (1 - \alpha)),$$

so that for any $j \in [d]$ if we now let $\pi_z$ be the continuous conditional density of $S \mid Z = z$, then the derivative

$$\frac{\partial g_i(\theta; z)}{\partial \theta_j} = z_i \left( \frac{\partial}{\partial \theta_j} \int_{-\infty}^{z^\top \theta} \pi_z(s) ds \right) = \pi_z(z^\top \theta) z_i z_j,$$

exists by the chain rule and the fundamental theorem of calculus. We now note that for any $\theta \in \mathbb{B}_2(\theta^\star, \frac{\varepsilon}{G})$

1. $|z_i| \le G < \infty$ so $g_i(\theta; z)$ is an integrable function of $z$.

2. $|z_i z_j| \le G^2 < \infty$ and $\pi_z$ is upper bounded by $u_\Pi$ over $\{s = z^\top \theta : \theta \in \mathbb{B}_2(\theta^\star, \frac{\varepsilon}{G}), \|z\| \le G\}$

$$\left| \frac{\partial g_i(\theta; z)}{\partial \theta_j} \right| \le G^2 u_\Pi$$

with probability 1 and $G^2 u_\Pi < \infty$ is an integrable function of $z$.

These results allow us to swap expectation and differentiation to conclude that for any $\theta \in \mathbb{B}_2(\theta^\star, \frac{\varepsilon}{G})$

$$\nabla^2 L_\Pi(\theta) = \mathbb{E}_\Pi \left[ \pi_Z(Z^\top \theta) Z Z^T \right].$$

**Demonstrating (ii)(b).** Armed with our proof of (ii)(a) and an explicit expression for $\nabla^2 L_\Pi(\theta)$, the result (ii)(b) now follows from the fact that for any $\theta$ in this set $\pi_Z(Z^\top \theta) \ge p_\Pi$ and

$$\nabla^2 L_\Pi(\theta) \succeq p_\Pi \lambda I.$$

We have thus shown that $L_\Pi(\theta)$ is $\mu$-strongly convex for $\mu = p_\Pi \lambda$ on $\mathbb{B}_2(\theta^\star, \frac{\varepsilon}{G})$.

**Demonstrating (ii)(c).** If $\varepsilon \ge GD$ then $L_\Pi$ is $\mu$-strongly convex for all $\Theta$ so the claim immediately holds. If $\varepsilon < GD$ for any $\theta \in \Theta$ and $\beta \in [0, 1]$ we can define $\tilde{\theta} = \beta \theta + (1 - \beta)\theta^\star \in \Theta$ so that

$$\left\| \tilde{\theta} - \theta^\star \right\| = \beta \left\| \tilde{\theta} - \theta^\star \right\| \le \beta D,$$

and choosing $\beta = \frac{\varepsilon}{GD} \in (0, 1)$ now ensures that $\tilde{\theta} \in \mathbb{B}_2(\theta^\star, \frac{\varepsilon}{G})$. We now know that for any other $\theta \ne \theta^\star$ the convex optimality conditions indicate that $\langle \nabla L_\Pi(\theta^\star), \theta - \theta^\star \rangle \ge 0$ and by the convexity of $L_\Pi(\theta)$

$$
\begin{aligned}
L_\Pi(\theta) - L_\Pi(\theta^\star) &\ge \frac{1}{\beta} \left( L_\Pi(\tilde{\theta}) - L_\Pi(\theta^\star) \right) \\
&\ge \langle \nabla L_\Pi(\theta^\star), \theta - \theta^\star \rangle + \frac{p_\Pi \lambda}{2\beta} \left\| \tilde{\theta} - \theta^\star \right\|^2 \\
&\ge \frac{p_\Pi \lambda}{2} \beta \left\| \theta - \theta^\star \right\|^2.
\end{aligned}
$$

This implies that $\theta^\star$ is unique since $L_\Pi(\theta) > L_\Pi(\theta^\star)$ for $\theta \ne \theta^\star$.

**Demonstrating Claim C.6.** To obtain our upper bound we define $\hat{\mathbf{g}}_t \in \partial l_t(\theta_t)$ as our empirical subgradient for time $t$ and turn to the standard recursive relationship

$$\|\theta_{t+1} - \theta^\star\|^2 \leq \|\theta_t - \theta^\star\|^2 - 2\eta_t \langle \hat{\mathbf{g}}_t, \theta_t - \theta^\star \rangle + \eta_t^2 \|\hat{\mathbf{g}}_t\|_2^2 \,.$$

We can now use the convexity of $\ell_t$ to conclude that

$$\|\theta_{t+1} - \theta^\star\|^2 \leq \|\theta_t - \theta^\star\|^2 - 2\eta_t(\ell_t(\theta_t) - \ell_t(\theta^\star)) + \eta_t^2 \|\hat{\mathbf{g}}_t\|_2^2$$
$$= \|\theta_t - \theta^\star\|^2 - 2\eta_t(L_\Pi(\theta_t) - L_\Pi(\theta^\star)) + 2\eta_t(L_\Pi(\theta_t) - L_\Pi(\theta^\star) - \ell_t(\theta_t) + \ell_t(\theta^\star)) + \eta_t^2 \|\hat{\mathbf{g}}_t\|_2^2 \,.$$

Recall that by Claim C.5 if $\mu = p_\Pi \lambda \min\left\{\frac{\varepsilon}{GD}, 1\right\}$ then

$$L_\Pi(\theta) - L_\Pi(\theta^\star) \geq \frac{\mu}{2} \|\theta - \theta^\star\|^2 \,,$$

so that

$$\|\theta_{t+1} - \theta^\star\|^2 \leq (1 - \mu\eta_t) \|\theta_t - \theta^\star\|^2 + 2\eta_t(L_\Pi(\theta_t) - L_\Pi(\theta^\star) - \ell_t(\theta_t) + \ell_t(\theta^\star)) + \eta_t^2 \|\hat{\mathbf{g}}_t\|_2^2 \,.$$

For arbitrary $t \geq 2$ we now set $\eta_t = \frac{2^c}{\mu t^c}$ and unroll the inequality to obtain

$$\|\theta_{t+1} - \theta^\star\|^2 \leq \prod_{i=2}^t \left(1 - \left(\frac{2}{i}\right)^c\right) \|\theta_2 - \theta^\star\|^2$$
$$+ \sum_{i=2}^t \prod_{j=i+1}^t \left(1 - \left(\frac{2}{j}\right)^c\right) 2\eta_i(L_\Pi(\theta_i) - L_\Pi(\theta^\star) - \ell_i(\theta_i) + \ell_i(\theta^\star))$$
$$+ \sum_{i=2}^t \prod_{j=i+1}^t \left(1 - \left(\frac{2}{j}\right)^c\right) \eta_i^2 \|\hat{\mathbf{g}}_i\|^2 \,,$$

where the first term vanishes since $(1 - 2^c/i^c) = 0$ at $i = 2$.

**Demonstrating Proposition C.8.** We can obtain a bound for the leftmost term in Claim C.6 by applying Lemma C.7 with

$$\nu_i = \prod_{j=i+1}^t \left(1 - \left(\frac{2}{j}\right)^c\right) \frac{2^{1+c}}{\mu i^c} \,,$$

and using the learning rate inequalities in Lemmas E.4, E.5, E.6, and E.8 to bound each individual term. After some algebra, this yields the upper bound

$$\mathbb{E}[A] \leq \frac{GD\beta(\tau)}{\mu} + \frac{28\tau GD}{\mu t^c} + \frac{8\tau G^2}{\mu^2 t^c} \,,$$

and choosing $\tau = \tau_\beta(P, (t+1)^{-c})$

$$\mathbb{E}[A] \leq \frac{GD}{\mu(t+1)^c} + \frac{28\tau GD}{\mu t^c} + \frac{8\tau G^2}{\mu^2 t^c} \,.$$

To handle the remaining term, we see that by the learning rate inequality in Lemma E.6

$$\sum_{i=2}^t \prod_{j=i+1}^t \left(1 - \left(\frac{2}{j}\right)^c\right) \frac{2^{2c}}{\mu^2 i^{2c}} \mathbb{E}\left[\|\hat{\mathbf{g}}_i\|^2\right] \leq \frac{4G^2}{\mu^2 t^c} \leq \frac{4\tau G^2}{\mu^2 t^c} \,,$$

and thus since $\left(\frac{t+1}{t}\right)^c \leq \frac{3}{2}$ for $t \geq 2$,

$$\mathbb{E}\left[\|\theta_{t+1} - \theta^\star\|^2\right] \leq \frac{GD}{\mu(t+1)^c} + \frac{56\tau GD}{\mu(t+1)^c} + \frac{18\tau G^2}{\mu^2(t+1)^c}$$
$$\leq 58 \left(\frac{G^2}{\mu^2} + \frac{GD}{\mu}\right) \frac{\tau}{(t+1)^c} \,.$$

The value $t \geq 2$ was chosen arbitrarily so the result holds for any $3 \leq t + 1 \leq T + 1$. Moreover, the inequality (6) in the proof of Theorem 5.4 also holds in this case, with the corresponding $\mu$, so $\|\theta_i - \theta^\star\|^2 \leq GD/\mu$ and the bound is also valid for $t + 1 = 1, 2$.

**Demonstrating Lemma C.9.** We start by using our local upper bound on the conditional density to upper bound the coverage gap

$$
\begin{aligned}
R_t &= |\mathbb{P}\left(Y_t \in C_t(X_t) \mid \mathcal{F}_{t-1}\right) - (1 - \alpha_t)| \\
&= |\mathbb{P}\left(S_t \leq q_t(\theta_t) \mid \mathcal{F}_{t-1}\right) - \mathbb{P}\left(S_t \leq q_t(\theta^\star) \mid \mathcal{F}_{t-1}\right)| \\
&= \left|\int (\mathbf{1}_{S_t \in [q_t(\theta^\star), q_t(\theta_t)]} - \mathbf{1}_{S_t \in [q_t(\theta_t), q_t(\theta^\star)]})dP_{S_t|\mathcal{F}_{t-1}}\right| \\
&\leq \mathbf{1}_{\|\theta_t - \theta^\star\| > \frac{\varepsilon}{G}} + \left|\int \mathbf{1}_{\|\theta_t - \theta^\star\| \leq \frac{\varepsilon}{G}}(\mathbf{1}_{S_t \in [q_t(\theta^\star), q_t(\theta_t)]} - \mathbf{1}_{S_t \in [q_t(\theta_t), q_t(\theta^\star)]})dP_{S_t|\mathcal{F}_{t-1}}\right| \\
&\leq \mathbf{1}_{\|\theta_t - \theta^\star\| > \frac{\varepsilon}{G}} + u\,|q_t(\theta_t) - q_t(\theta^\star)| \\
&\leq \mathbf{1}_{\|\theta_t - \theta^\star\| > \frac{\varepsilon}{G}} + uG\,\|\theta_t - \theta^\star\|.
\end{aligned}
$$

We now take the expectation of $R_t$ to conclude that

$$
\mathbb{E}\left[R_t\right] \leq \mathbb{P}\left(\|\theta_t - \theta^\star\| > \frac{\varepsilon}{G}\right) + uG\mathbb{E}\left[\|\theta_t - \theta^\star\|\right],
$$

and by Markov's inequality,

$$
\mathbb{E}\left[R_t\right] \leq \left(\frac{G}{\varepsilon} + uG\right)\mathbb{E}\left[\|\theta_t - \theta^\star\|\right].
$$

Finally, applying Jensen's inequality

$$
\mathbb{E}\left[R_t\right] \leq \left(\frac{G}{\varepsilon} + uG\right)\sqrt{\mathbb{E}\left[\|\theta_t - \theta^\star\|^2\right]},
$$

which proves the desired result.

**Demonstrating Lemma C.7.** The proof of this lemma follows from a similar decomposition as in the proof of Theorem 3.1 in Duchi et al. (2012), so we first introduce the following two lemmas adapted from their analysis.

**Lemma C.10.** *[Adapted from Lemma 6.3 in Duchi et al. (2012)] Let our ergodicity assumption hold, let our conformal models be $G$-Lipschitz, and let $\Theta$ have finite diameter $D$. Then for $\theta \in m(\mathcal{F}_i)$, $t > \tau \geq 0$, $i \leq t - \tau$, and any $\theta^\star \in \Theta$*

$$
\mathbb{E}\left[L_\Pi(\theta) - L_\Pi(\theta^\star) - \ell_{i+\tau}(\theta) + \ell_{i+\tau}(\theta^\star) \mid \mathcal{F}_i\right] \leq GD\left\|P_{[i]}^{i+\tau} - \Pi\right\|_{\mathrm{TV}}
$$

*Proof.* Note that

$$
\begin{aligned}
\mathbb{E}\left[L_\Pi(\theta) - L_\Pi(\theta^\star) - \ell_{i+\tau}(\theta) + \ell_{i+\tau}(\theta^\star) \mid \mathcal{F}_i\right] = &\int (\ell(s - z^\top\theta) - \ell(s - z^\top\theta^\star))d\Pi(z, s) \\
&- \int (\ell(s - z^\top\theta) - \ell(s - z^\top\theta^\star))dP_{[i]}^{i+\tau}(z, s),
\end{aligned}
$$

and since we assume that $\Pi$ and $P_{[i]}^{i+\tau}$ have densities with respect to the Lebesgue measure $\lambda$ on $\mathbb{R}^{d+1}$

$$
\begin{aligned}
|\mathbb{E}\left[L_\Pi(\theta) - L_\Pi(\theta^\star) - \ell_{t+\tau}(\theta) + \ell_{t+\tau}(\theta^\star) \mid \mathcal{F}_t\right]| &= \left|\int (\ell(s - z^\top\theta) - \ell(s - z^\top\theta^\star))(\pi(z, s) - p_{[i]}^{i+\tau}(z, s))d\lambda(z, s)\right| \\
&\leq GD\int |\pi(z, s) - p_{[i]}^{i+\tau}(z, s)|d\lambda(z, s) \\
&= GD\left\|P_{[i]}^{i+\tau} - \Pi\right\|_{\mathrm{TV}}.
\end{aligned}
$$

$\square$

**Lemma C.11.** *[Adapted from Lemma 6.4 in [Duchi et al. (2012)]] Let our conformal models be G-Lipschitz and let $\{\eta_t\}_t$ be a non-increasing sequence of step sizes. For $t > \tau \geq 0$ and $i \leq t - \tau$ the iterates produced by Algorithm 1 with batch size 1 satisfy*

$$|\ell_{i+\tau}(\theta_i) - \ell_{i+\tau}(\theta_{i+\tau})| \leq \tau \eta_i G^2.$$

*Proof.* Recall that

$$\theta_{i+\tau} = \theta_i - \sum_{r=i}^{i+\tau-1} \eta_r \partial \ell_r(\theta_r),$$

and since the learning rate schedule $\{\eta_t\}$ is non-increasing

$$\|\theta_{i+\tau} - \theta_i\| = \left\| \sum_{r=i}^{i+\tau-1} \eta_r \partial \ell_r(\theta_r) \right\|$$

$$\leq \sum_{r=i}^{i+\tau-1} \eta_r \|\partial \ell_r(\theta_r)\|$$

$$\leq \tau \eta_i G.$$

Finally, using our Lipschitz assumption once again

$$|\ell_{i+\tau}(\theta_i) - \ell_{i+\tau}(\theta_{i+\tau})| \leq \tau \eta_i G^2.$$

$\square$

We now note that we can decompose $A$ into the following 5 terms

$$A_1 = \sum_{i=2}^{t-\tau} \nu_i \left( L_\Pi(\theta_i) - L_\Pi(\theta^\star) - \ell_{i+\tau}(\theta_i) + \ell_{i+\tau}(\theta^\star) \right)$$

$$A_2 = \sum_{i=2}^{t-\tau} \nu_{i+\tau} (\ell_{i+\tau}(\theta_i) - \ell_{i+\tau}(\theta_{i+\tau}))$$

$$A_3 = \sum_{i=2}^{t-\tau} (\nu_i - \nu_{i+\tau})(\ell_{i+\tau}(\theta_i) - \ell_{i+\tau}(\theta^\star))$$

$$A_4 = \sum_{i=2}^{\tau+1} \nu_i \left( \ell_i(\theta^\star) - \ell_i(\theta_i) \right)$$

$$A_5 = \sum_{i=t-\tau+1}^{t} \nu_i (L_\Pi(\theta_i) - L_\Pi(\theta^\star) - \ell_i(\theta_i) + \ell_i(\theta^\star)),$$

so that $A = \sum_{k=1}^{5} A_k$. We will now bound the expectation of each of these terms individually. Firstly, we observe that

$$\mathbb{E}[A_1] = \sum_{i=2}^{t-\tau} \nu_i \mathbb{E}\left[ (L_\Pi(\theta_i) - L_\Pi(\theta^\star) - \ell_{i+\tau}(\theta_i) + \ell_{i+\tau}(\theta^\star)) \right]$$

$$= \sum_{i=2}^{t-\tau} \nu_i \mathbb{E}\left[ \mathbb{E}\left[ (L_\Pi(\theta_i) - L_\Pi(\theta^\star) - \ell_{i+\tau}(\theta_i) + \ell_{i+\tau}(\theta^\star)) \mid \mathcal{F}_i \right] \right],$$

and apply Lemma C.10 to obtain

$$\mathbb{E}[A_1] \leq \sum_{i=2}^{t-\tau} \nu_i GD \frac{\beta(\tau)}{2}.$$

For $A_2$ we note that by Lemma C.11

$$A_2 \leq \sum_{i=2}^{t-\tau} \nu_{i+\tau}\tau\eta_i G^2,$$

so $\mathbb{E}[A_2]$ must also be upper bounded by this quantity. The upper bounds for the remaining 3 terms follow from the triangle inequality, our Lipschitz assumption, and our diameter bound

$$\mathbb{E}[A_3] \leq \sum_{i=2}^{t-\tau} GD|\nu_i - \nu_{i+\tau}|$$

$$\mathbb{E}[A_4] \leq \sum_{i=2}^{\tau} \nu_i GD$$

$$\mathbb{E}[A_5] \leq \sum_{i=t-\tau+1}^{t} 2\nu_i GD.$$

Summing up all the stated upper bounds yields the result.

## D. Justifying the quantile loss

Here we provide some intuition to justify why the quantile loss is an effective metric for evaluating and comparing online conformal algorithms. We argue that the minimizers of the quantile loss exhibit desirable properties in both mis-specified ergodic and adversarial settings, as discussed in Sections D.1 and D.2, respectively.

In addition to these results, we also point out that the quantile loss is often quite natural for the online conformal task. Specifically, it grows linearly as the predicted quantile deviates further from the observed score $S_t$, and its asymmetric nature penalizes mis-coverage, where an actual error occurs, more heavily than over-coverage, where the conformal set is simply less conservative than necessary, as long as $\alpha < 0.5$.

### D.1. Properties of the quantile loss in the mis-specified ergodic setting

In Section 6, we argued that if our quantile models are linear and the covariate and score pairs $\{(Z_t, S_t)\}$ are produced by an ergodic process $P$ converging to a stationary distribution $\Pi$, then the iterates of Algorithm 1 with batch size $m = 1$ converge to $\theta^\star$ minimizing $L_\Pi$, the expected quantile loss with respect to the stationary distribution. This implies that the coverage properties of our algorithm are determined by

$$\alpha_t = \mathbb{P}\left(S_t > Z_t^\top \theta^\star \mid \mathcal{F}_{t-1}\right).$$

To argue that these $\alpha_t$ often provide good coverage in the mis-specified setting, we consider the case where $\mathbb{P}\left(S_t > Z_t^\top \theta \mid \mathcal{F}_{t-1}\right) = \mathbb{P}\left(S_t > Z_t^\top \theta \mid Z_t\right)$. This holds, for example, when the distribution of $S_t$ only depends on the previous $q$ scores and we use an LQT model of order $p \geq q$. In this case we can provide an alternative characterization of the stronger guarantee (2), since we assume that $\Pi$ is the stationary distribution of our process

$$\alpha_t(z) := \mathbb{P}\left(S_t > z^\top \theta^\star \mid Z_t = z\right) = \mathbb{E}_\Pi\left[\mathbf{1}_{\{S > z^\top \theta^\star\}} \mid Z = z\right],$$

so that $\alpha_t = \alpha \in \sigma(Z)$ if and only if

$$\sup_{V \in L^2(\Omega, \sigma(Z), \Pi_Z)} \mathbb{E}_\Pi\left[V(\alpha - \mathbf{1}_{\{S > Z^\top \theta^\star\}})\right] = 0.$$

In the mis-specified setting, our predictor will not be able to attain conditional coverage, but may be able to attain the weaker condition of coverage with respect to a family of witnesses $\mathcal{V} \subset L^2(\Omega, \sigma(Z), \Pi_Z)$:

$$\sup_{V \in \mathcal{V}} \mathbb{E}_\Pi\left[V(\alpha - \mathbf{1}_{\{S > Z^\top \theta^\star\}})\right] = 0.$$

We will now argue that these $\alpha_t$ provide good coverage within our class of conformal predictors since for any other $\theta$ we can find a witness function $V$ defined by our features $Z$ such that the sets obtained from $q_t(\theta)$ have strictly worse coverage than $q_t(\theta^\star)$ relative to $V$.

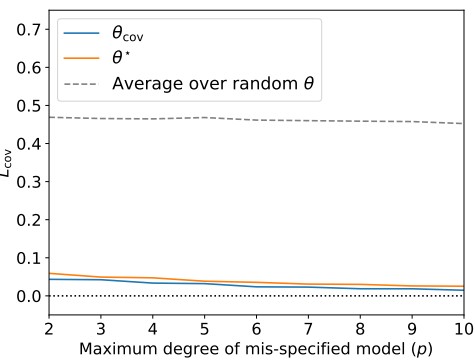

Figure 4. Comparison of attained $L_{\mathrm{cov}}$ for $\theta_{\mathrm{cov}}$, $\theta^\star$, and the average value of $L_{\mathrm{cov}}$ over 10,000 randomly chosen $\theta$.

**Theorem D.1.** *Let the assumptions in the previous paragraph and those of Theorem 6.1 hold. Then, for any $\theta \in \Theta \setminus \{\theta^\star\}$ there exists $\beta(\theta) \in \mathbb{R}^d$ such that the witness $V = \beta(\theta)^\top Z$ satisfies*

$$\mathbb{E}_\Pi \left[ V(\alpha - \mathbf{1}_{\{S > Z^\top \theta\}}) \right] > \mathbb{E}_\Pi \left[ V(\alpha - \mathbf{1}_{\{S > Z^\top \theta^\star\}}) \right] \geq 0.$$

*Proof.* Recall from our proof of Theorem 6.1 that $L_\Pi(\theta) = \mathbb{E}_\Pi \left[ \ell_{\mathrm{quantile}}(S - Z^\top \theta) \right]$ has gradient

$$\nabla L(\theta) = \mathbb{E}_\Pi \left[ Z(\alpha - \mathbf{1}_{\{S > Z^\top \theta\}}) \right],$$

and that by the convex optimality conditions

$$\langle \nabla L_\Pi(\theta), \theta - \theta^\star \rangle \geq L_\Pi(\theta) - L_\Pi(\theta^\star) \geq \langle \nabla L_\Pi(\theta^\star), \theta - \theta^\star \rangle + \frac{\mu}{2} \|\theta - \theta^\star\|_2^2 > \langle \nabla L_\Pi(\theta^\star), \theta - \theta^\star \rangle \geq 0.$$

This immediately implies that for $\beta(\theta) = (\theta - \theta^\star)$ and $V = \beta(\theta)^\top Z$

$$\mathbb{E}_\Pi \left[ V(\alpha - \mathbf{1}_{\{S > Z^\top \theta\}}) \right] > \mathbb{E}_\Pi \left[ V(\alpha - \mathbf{1}_{\{S > Z^\top \theta^\star\}}) \right] \geq 0.$$

$\square$

Note that the proof of Theorem D.1 also shows that when $\theta^\star \in \mathrm{Int}(\Theta)$, the model defined by $\theta^\star$ is the only one to achieve coverage relative to our features $Z_i$ since $\nabla L(\theta^\star) = \mathbb{E}_\Pi \left[ Z(\alpha - \mathbf{1}_{\{S > Z^\top \theta^\star\}}) \right] = 0$. This result illustrates why minimizing the quantile loss is desirable, especially when $\theta^\star \in \mathrm{Int}(\Theta)$, as the minimizer obtains the best coverage relative to our features.

We can also verify that $\alpha_t$ provide good coverage in synthetic experiments where the stationary distribution $\Pi$ is known. For our experiment we consider the standard AR(1) model with standard Gaussian i.i.d. innovations and parameter 0.5, and use a mis-specified linear model with feature vector $\Phi_t(S_1^{t-1})^\top = [1, S_1^2, \cdots, S_1^p]$ for $p \geq 2$. In this case, a natural way to measure coverage of our model is through

$$L_{\mathrm{cov}}(\theta) = \mathbb{E}_\Pi \left[ \left| \mathbb{P} \left( S > Z^\top \theta \mid Z \right) - \alpha \right| \right],$$

the expected absolute difference between the conditional mis-coverage of our model and $\alpha$. We now show the values of $L_{\mathrm{cov}}$ attained by the minimizer $\theta^\star$ of $L_\Pi$, the minimizer $\theta_{\mathrm{cov}}$ of $L_{\mathrm{cov}}$, and the average value of $L_{\mathrm{cov}}$ for 10,000 randomly chosen $\theta$, in Figure 4. Our results suggest that even under the criterion $L_{\mathrm{cov}}$ the $\theta^\star$ minimizing the quantile loss provides good coverage comparable to $\theta_{\mathrm{cov}}$, which is significantly better than the coverage attained by most other parameter values.

### D.2. Properties of the quantile loss in the adversarial setting

In many cases, we can interpret online conformal prediction algorithms from online convex optimization perspective, where we analyze the performance of an algorithm $\mathcal{A}$ producing iterates $\theta_t$ optimizing a sequence of convex functions $f_t : \Theta \to \mathbb{R}$ in terms of *static regret* with respect to some set $\bar{\Theta} \subseteq \Theta$

$$R_T(\mathcal{A}, \{f_t\}, \bar{\Theta}) = \sum_{t=1}^T f_t(\theta_t) - \inf_{\theta \in \bar{\Theta}} \sum_{t=1}^T f_t(\theta).$$

In this context, by the argument in Theorem 3.1 of (Hazan, 2016) Algorithm 1 ($\mathcal{A}_1$) also satisfies the following regret bound.

**Theorem D.2.** *Let $\ell_t$ be convex and $G$-Lipschtiz for every $t \in \mathbb{N}$, and let $\Theta$ have finite diameter $D$. Then the iterates $\{\theta_t\}_t$ produced by Algorithm 1 with non-increasing learning rates $\{\eta_t\}_t$ and $m = 1$ satisfy*

$$R_T(\mathcal{A}_1, \{\ell_t\}, \Theta) \leq \frac{D^2}{\eta_T} + G^2 \sum_{t=1}^{T} \eta_t.$$

We now note that by Theorem 4.1, when scores are bounded Algorithm 1 also achieves small constraint violation with respect to the constraint functions $g_t(\theta) = \mathbb{1}_{q_t(\theta) < S_t} - \alpha$. Therefore, if we define the set of parameters that satisfy the cumulative constraints in hindsight

$$\bar{\Theta}_T = \left\{ \theta \in \Theta : \sum_{t=1}^{T} g_t(\theta) \leq 0 \right\},$$

Algorithm 1 with learning rates $\eta_t = ct^{-1/2-\epsilon}$ for some $c, \epsilon > 0$ and $m = 1$ provides a solution to the constrained non-convex online optimization problem

$$\min_{\theta_1, \cdots, \theta_T \in \Theta} \sum_{t=1}^{T} \ell_t(\theta_t) - \min_{\theta \in \bar{\Theta}_T} \sum_{t=1}^{T} \ell_t(\theta)$$

$$\text{s.t.} \sum_{t=1}^{T} g_t(\theta_t) \leq 0,$$

having sublinear regret $\mathcal{R}_T(\mathcal{A}_1, \{\ell_t\}, \bar{\Theta}_T) = O(T^{1/2+\epsilon})$ and constraint violations $\sum_{t=1}^{T} g_t(\theta_t) \leq O(T^{1/2+\epsilon})$, simultaneously.

Note that any algorithm with sublinear constraint violations will satisfy

$$\frac{1}{T} \sum_{t=1}^{T} \text{err}_t - \alpha \leq o(1),$$

which is equivalent to providing at least $(1 - \alpha)$ adversarial marginal coverage in the long-run. This naturally prompts the question of whether we can design algorithms with at least $(1 - \alpha)$ adversarial marginal coverage and similarly small regret $\mathcal{R}_T(\mathcal{A}_1, \{f_t\}, \bar{\Theta}_T)$ for an arbitrary set of convex functions $\{f_t\}$ chosen by the user. The following Lemma, which follows from a simple adaptation of the argument in Proposition 4 of Mannor et al. (2009), points out that this is impossible in general even when objective and constraint functions are linear. We provide a proof of this fact in Section D.3 for convenience of the reader.

**Theorem D.3.** *Let $\Theta$ have finite diameter $D$. Then there exist sequences of linear functions $\{f_t\}, \{g_t\}$ such that any algorithm $\mathcal{A}$ attaining*

$$\limsup_{T \to \infty} \frac{1}{T} \sum_{t=1}^{T} g_t(\theta_t) \leq 0,$$

*also suffers*

$$\limsup_{T \to \infty} \frac{\mathcal{R}_T(\mathcal{A}, \{f_t\}, \bar{\Theta}_T)}{T} > 0,$$

*and it is thus impossible to attain sublinear regret and constraint violations simultaneously.*

However, we believe the design of online conformal algorithms that allow the user to trade off between coverage and regret with respect to some arbitrary sequence of convex loss functions $\{f_t\}$ is a promising area of future research.

### D.3. Proof of Theorem D.3

This proof closely follows the proof of Proposition 4 of Mannor et al. (2009); we only adapt it to our online convex optimization setup for the convenience of the reader. To prove this result, we first define the behavior of our adversary and then show how this choice makes it impossible to attain sublinear regret and constraint violations simultaneously.

**Defining the adversary.** Let $(v, w) \in \operatorname{argmax}_{v,w \in \Theta} \|v - w\|$ be the two furthest points in our set so that $\|v - w\| = D < \infty$, and consider a setting where the behavior of our adversary is dictated by a sequence of indices $j_t$ so that

$$g_t(\theta) = \begin{cases} -1 & j_t = 1 \\ \frac{1}{D^2}(w - v)^\top(\theta - v) & j_t = 2, \end{cases}$$

and $f_t(\theta) = 1 - \frac{1}{D^2}(w - v)^\top(\theta - v)$. For any $\theta \in \Theta$

$$f_t(\theta) \geq 1 - \frac{1}{D^2}\|w - v\| \|\theta - v\| \geq 0,$$

and $f_t(w) = 0$, so that $w$ minimizes $f_t$ for any $t$. Let $\hat{q}_T = \frac{1}{T}\sum_{t=1}^{T} \mathbb{1}_{j_t=2}$ then if $\hat{q}_T \leq 1/2$ we could choose $\theta = w$ (in hindsight) to minimize all $f_t(\theta)$ while satisfying the average constraint. We now let our adversary initialize a counter $k = 0$ and pick the indices $j_t$ using the following two steps:

1. While $k = 0$ or $\frac{1}{D^2(t-1)}\sum_{r=1}^{t-1}(w - v)^\top(\theta_r - v) > \frac{1}{2}$ pick $j_t = 2$ and increment $k$ by 1.

2. For the next $k$ time indices $j_t = 1$. Then, reset counter $k$ to 0 and go back to step 1.

**Proving the impossibility result.** We will now show that any algorithm that has sublinear constraint violations against this adversary cannot attain sublinear regret. To do so, we show that:

(i) The adversary executes both steps infinitely often.

(ii) Any algorithm attains at least linear regret when the adversary finishes executing step 2.

**Demonstrating claim (i).** We will prove this result by contradiction, so we first assume that step 2 occurs only a finite number of times. This implies that after a finite number of time steps, the adversary only chooses $j_t = 2$ so that $\hat{q}_T \to 1$. However, our sublinear constraint

$$\limsup_{T \to \infty} \frac{1}{T}\sum_{t=1}^{T} g_t(\theta_t) \leq 0$$

implies that the condition $\frac{1}{D^2(t-1)}\sum_{r=1}^{t-1}(w - v)^\top(\theta_r - v) > \frac{1}{2}$ must be violated at some point, triggering step 2. This result implies that both steps occur infinitely often, so we can find infinitely many $t_i < t'_i < t_{i+1}$ such that the adversary plays step 1 for $t \in (t_i, t'_i]$ and step 2 for $t \in (t'_i, t_{i+1}]$.

**Demonstrating claim (ii).** Since both steps are equally long we know that $\hat{q}_{t_i} = \frac{1}{2}$ and

$$\inf_{\theta \in \Theta_{t_i}} \sum_{r=1}^{t_i} f_r(\theta) = 0.$$

Observe that by construction $t_{i+1} - t'_i \leq t'_i$ so that $t'_i \geq \frac{t_{i+1}}{2}$ and thus

$$\frac{1}{D^2 t_{i+1}}\sum_{r=1}^{t_{i+1}}(w - v)^\top(\theta_r - v) \leq \frac{1}{t_{i+1}}\left(\frac{t'_i}{2} + (t_{i+1} - t'_i)\right)$$

$$= \frac{1}{t_{i+1}}\left(t_{i+1} - \frac{t'_i}{2}\right)$$

$$\leq \frac{1}{t_{i+1}}\left(t_{i+1} - \frac{t_{i+1}}{4}\right) = \frac{3}{4}.$$

This implies that

$$\frac{\mathcal{R}_{t_{i+1}}(\mathcal{A}, \{f_t\}, \bar{\Theta}_{t_{i+1}})}{t_{i+1}} = 1 - \frac{1}{D^2 t_{i+1}}\sum_{r=1}^{t_{i+1}}(w - v)^\top(\theta_r - v) \geq \frac{1}{4},$$

and also the desired result

$$\limsup_{T \to \infty} \frac{\mathcal{R}_T(\mathcal{A}, \{f_t\}, \bar{\Theta}_T)}{T} \geq \frac{1}{4} > 0.$$

## E. Auxiliary results

### E.1. Existing Theorems and Lemmas

In this section we compile several known results that we use in our proofs for the convenience of the reader. The first result is a useful tool for asymptotic convergence proofs commonly known as the Robbins–Siegmund quasimartingale convergence Theorem. We apply this result in our proof of Theorem 5.1.

**Theorem E.1.** *[Adapted from Robbins & Siegmund (1971)] Let $(\Omega, \mathcal{F}, P)$ be a probability space and $\mathcal{F}_1 \subset \mathcal{F}_2 \subset \cdots$ be a sequence of sub-$\sigma$-algebras of $\mathcal{F}$. For each $n = 1, 2, \cdots$ let $V_n, D_n, U_n,$ and $\beta_n$ be non-negative $\mathcal{F}_n$-measurable random variables such that*

$$\mathbb{E}\left[V_{n+1} \mid \mathcal{F}_n\right] \leq V_n(1 + \beta_n) - D_n + U_n,$$

*then $\lim_{n \to \infty} V_n$ exists and is finite and $\sum_{n=1}^{\infty} D_n < \infty$ almost surely on*

$$\left\{\sum_{n=1}^{\infty} \beta_n < \infty, \sum_{n=1}^{\infty} U_n < \infty\right\}.$$

The second result is a concentration inequality for martingale difference sequences from Bartlett et al. (2008) and is used to derive the high-probability bounds in Theorem 5.4.

**Lemma E.2.** *[Copied from Bartlett et al. (2008)] Let $d_1, \ldots, d_T$ be a martingale difference sequence with a uniform bound $|d_i| \leq b$ for all $i$. Let $V = \sum_{t=1}^{T} \text{Var}_{t-1}(d_t)$ be the sum of conditional variances of $d_t$'s. Further, let $\sigma = \sqrt{V}$. Then we have, for any $\delta < 1/e$ and $T \geq 4$,*

$$\mathbb{P}\left(\sum_{t=1}^{T} d_t > 2 \max\left\{2\sigma, b\sqrt{\log(1/\delta)}\right\} \sqrt{\log(1/\delta)}\right) \leq \log(T)\delta.$$

The final result provides some lower bounds on the expected quantile loss gap between some $q \in \mathbb{R}$ and the true quantile $q^\star$, in terms of the distance between $q$ and $q^\star$. We use this lemma for our proofs of the results in Section 5 and 6.

**Lemma E.3.** *Let $S$ have a positive continuous density $f$, lower bounded by $p > 0$ in an $\varepsilon$-neighborhood around its unique $(1 - \alpha)$ quantile $q^\star$. Then,*

(i) *For $q$ such that $|q - q^\star| \leq \varepsilon$,*

$$\mathbb{E}[\ell_{quantile}(S - q) - \ell_{quantile}(S - q^*)] \geq \frac{p}{2}|q - q^\star|^2.$$

(ii) *For $q$ such that $|q - q^\star| > \varepsilon$,*

$$\mathbb{E}[\ell_{quantile}(S - q) - \ell_{quantile}(S - q^*)] \geq \frac{p\varepsilon}{2}|q - q^\star|.$$

(iii) *For $q$ such that $|q - q^\star| \leq K$ with $K \in (\varepsilon, \infty)$,*

$$\mathbb{E}[\ell_{quantile}(S - q) - \ell_{quantile}(S - q^*)] \geq \frac{p\varepsilon}{2K}|q - q^\star|^2.$$

*Proof.* We dedicate a section to the proof of each lower bound.

**Demonstrating (i).** We only prove (i) for $q \geq q^\star$ as the other direction follows by an identical argument. We temporarily adopt the notation $L(q) := \mathbb{E}[\ell_{\text{quantile}}(S - q)]$ and recall that

$$\partial_q \ell_{\text{quantile}}(S - q) = \begin{cases} \alpha - \mathbb{1}_{S-q>0} & S \neq q \\ [\alpha - 1, \alpha] & S = q. \end{cases}$$

The subgradient of the expected quantile loss $L$ is thus

$$\partial_q L(q) = \partial_q \mathbb{E}[\ell_{\text{quantile}}(S - q)] = \mathbb{E}[\partial_q \ell_{\text{quantile}}(S - q)] = \mathbb{E}[\alpha - \mathbb{1}_{S-q>0}] = \mathbb{P}(S \leq q) - (1 - \alpha),$$

since we assume that $S$ has a density. This is single valued so we can conclude that $L(q)$ is differentiable and $L'(q) = \mathbb{P}(S \leq q) - (1 - \alpha)$. Note that $\mathbb{P}(S \leq q^\star) = (1 - \alpha)$ since $f$ is continuous, and for $q \in [q^\star, q^\star + \varepsilon]$,

$$L'(q) = L'(q) - L'(q^\star) = \mathbb{P}(S \leq q) - \mathbb{P}(S \leq q^\star) = \int_{q^\star}^{q} f(s)ds \geq p(q - q^\star).$$

This now implies that

$$L(q) - L(q^\star) = \int_{q^\star}^{q} L'(z)dz \geq p\left(\frac{q^2}{2} - qq^\star - \frac{q^{\star 2}}{2} + q^{\star 2}\right) = \frac{p}{2}(q - q^\star)^2.$$

**Demonstrating (ii).** We once again only prove (ii) for $q \geq q^\star$ as the other direction follows by an identical argument. In this case, $q > q' := q^\star + \varepsilon$ so that

$$
\begin{aligned}
L(q) - L(q^\star) &= L(q) - L(q') + L(q') - L(q^\star) \\
&\geq L'(q')(q - q') + \frac{p}{2}(q' - q^\star)^2 && \text{(convexity of } L; \text{ lower bound for } q' \in [q^\star, q^\star + \varepsilon]) \\
&\geq p\varepsilon(q - q') + \frac{p\varepsilon}{2}(q' - q^\star) && (q' - q^\star = \varepsilon) \\
&\geq \frac{p\varepsilon}{2}(q - q^\star),
\end{aligned}
$$

as desired.

**Demonstrating (iii).** This follows immediately from the previous bound by noting that if $|q - q^\star| \leq \varepsilon$ then

$$L(q) - L(q^\star) \geq \frac{p}{2}|q - q^\star|^2 \geq \frac{p\varepsilon}{2K}|q - q^\star|^2,$$

and if $|q - q^\star| > \varepsilon$

$$
\begin{aligned}
L(q) - L(q^\star) &\geq \frac{p\varepsilon}{2}(q - q^\star) \\
&= \frac{p\varepsilon}{2}\frac{(q - q^\star)^2}{(q - q^\star)} \\
&\geq \frac{p\varepsilon}{2K}(q - q^\star)^2.
\end{aligned}
$$

$\square$

### E.2. Learning rate inequalities

The proofs of Theorems 5.4 and 6.1 rely on multiple inequalities bounding functions of the products $\prod_{j=i+1}^{t}\left(1 - \left(\frac{2}{j}\right)^c\right)$ for $2 \leq i \leq t$, so we present and prove them in this section. In standard analyses of strongly convex stochastic gradient descent, the learning rates are of the form $\Theta(1/t^c)$ for $c = 1$, which simplifies the analysis of these products.

**Lemma E.4.** *For any $t \geq 2$, $i \in [2, t]$ and $c \in [0, 1]$,*

$$\prod_{j=i+1}^{t}\left(1 - \left(\frac{2}{j}\right)^c\right)\frac{1}{i^c} \leq \frac{1}{t^c}.$$

*Proof.* We prove the statement by induction on $i = t, \ldots, 2$.

**Setting up the induction.** Let

$$f_t(i) = \prod_{j=i+1}^{t} \left(1 - \left(\frac{2}{j}\right)^c\right) \frac{1}{i^c},$$

and note the recursive relationship

$$f_t(i-1) = \left(1 - \left(\frac{2}{i}\right)^c\right) \frac{i^c}{(i-1)^c} \cdot f_t(i). \tag{7}$$

Our goal is now to show that $f_t(t) \leq \frac{1}{t^c}$ for the base case, and given $f_t(i) \leq \frac{1}{t^c}$ then $f_t(i-1) \leq \frac{1}{t^c}$ for the inductive step.

**Base case.** Observe that $f_t(t) = \frac{1}{t^c}$, so the inequality immediately holds.

**Inductive step.** We now move to the inductive step, where by our previous relationship (7) and the inductive hypothesis

$$
\begin{aligned}
f_t(i-1) &= \left(1 - \left(\frac{2}{i}\right)^c\right) \frac{i^c}{(i-1)^c} \cdot f_t(i) \\
&\leq \left(\frac{i^c - 2^c}{(i-1)^c}\right) \frac{1}{t^c} \\
&\leq \frac{1}{t^c},
\end{aligned}
$$

proving the desired result. $\qquad\square$

**Lemma E.5.** *For $t \geq 2$ and $c \in [0, 1]$,*

$$\sum_{i=2}^{t} \prod_{j=i+1}^{t} \left(1 - \left(\frac{2}{j}\right)^c\right) \frac{1}{i^c} \leq \frac{1}{2^c}.$$

*Proof.* We prove the statement by induction on $t \geq 2$.

**Setting up the induction.** Let

$$X_t := \sum_{i=2}^{t} \prod_{j=i+1}^{t} \left(1 - \left(\frac{2}{j}\right)^c\right) \frac{1}{i^c},$$

and note the recursive relationship

$$X_t = \left(1 - \left(\frac{2}{t}\right)^c\right) X_{t-1} + \frac{1}{t^c}. \tag{8}$$

We already know that $X_2 = 2^{-c}$ for the base case, so it remains to show that if $X_t \leq 2^{-c}$, then $X_{t+1} \leq 2^{-c}$.

**Inductive step.** We now move to the inductive step where, by our previous relationship (8) and the inductive hypothesis

$$
\begin{aligned}
X_{t+1} &= \left(1 - \left(\frac{2}{t+1}\right)^c\right) X_t + \frac{1}{(t+1)^c} \\
&\leq \left(1 - \left(\frac{2}{t+1}\right)^c\right) \frac{1}{2^c} + \frac{1}{(t+1)^c} \\
&\leq \frac{1}{2^c},
\end{aligned}
$$

proving the desired result.

$\qquad\square$

**Lemma E.6.** *For $t \geq 1$ and $c \in [0, 1]$,*

$$\sum_{i=2}^{t} \prod_{j=i+1}^{t} \left(1 - \left(\frac{2}{j}\right)^c\right) \frac{1}{i^{2c}} \leq \frac{1}{t^c}.$$

*Proof.* We prove the statement by induction on $t \geq 1$.

**Setting up the induction.** Let

$$X_t := \sum_{i=2}^{t} \prod_{j=i+1}^{t} \left(1 - \left(\frac{2}{j}\right)^c\right) \frac{1}{i^{2c}},$$

and note the recursive relationship

$$X_t = \left(1 - \left(\frac{2}{t}\right)^c\right) X_{t-1} + \frac{1}{t^{2c}}. \tag{9}$$

We already know that $X_1 = 0 \leq 1$ for the base case, so our goal is to show that if $X_{t-1} \leq (t-1)^{-c}$ then $X_t \leq t^{-c}$.

**Inductive step.** For $t \geq 2$, using the recursive relationship (9) and the inductive hypothesis,

$$X_t \leq \left(1 - \left(\frac{2}{t}\right)^c\right) \frac{1}{(t-1)^c} + \frac{1}{t^{2c}} \leq \left(\frac{t^c - 2^c + 1}{(t-1)^c}\right) \frac{1}{t^c}.$$

We now define $g(t) := t^c - (t-1)^c$ and note that $g'(t) = c(t^{c-1} - (t-1)^{c-1}) \leq 0$ so $g(t)$ is decreasing with $g(t) \leq g(2) = 2^c - 1$, and

$$X_t \leq \left(\frac{t^c - (t-1)^c - 2^c + 1}{(t-1)^c} + 1\right) \frac{1}{t^c} = \left(\frac{g(t) - 2^c + 1}{(t-1)^c} + 1\right) \frac{1}{t^c} \leq \frac{1}{t^c},$$

proving the desired result. $\square$

**Lemma E.7.** *For $t \geq 1$ and $c \in [0, 1]$,*

$$\sum_{i=2}^{t} \prod_{j=i+1}^{t} \left(1 - \left(\frac{2}{j}\right)^c\right)^2 \frac{1}{i^{3c}} \leq \frac{1}{t^{2c}}.$$

*Proof.* We prove the statement by induction on $t \geq 1$.

**Setting up the induction.** Let

$$X_t := \sum_{i=2}^{t} \prod_{j=i+1}^{t} \left(1 - \left(\frac{2}{j}\right)^c\right)^2 \frac{1}{i^{3c}},$$

and note the recursive relationship

$$X_t = \left(1 - \left(\frac{2}{t}\right)^c\right)^2 X_{t-1} + \frac{1}{t^{3c}}. \tag{10}$$

We already know that $X_1 = 0 \leq 1$ for the base case so our goal is to show that if $X_t \leq (t-1)^{-2c}$ then $X_t \leq t^{-2c}$.

**Inductive step.** For $t \geq 2$, using the recursive relationship (10) and the inductive hypothesis,

$$X_t \leq \left(1 - \left(\frac{2}{t}\right)^c\right)^2 \frac{1}{(t-1)^{2c}} + \frac{1}{t^{3c}} = \frac{1}{t^{2c}} \left(\left(\frac{t^c - 2^c}{(t-1)^c}\right)^2 + \frac{1}{t^c}\right).$$

We now recall from the last step of the proof of Lemma E.6 that

$$0 \le \frac{t^c - 2^c}{(t-1)^c} \le 1 - \frac{1}{(t-1)^c} \le 1,$$

so that $\left(\frac{t^c - 2^c}{(t-1)^c}\right)^2 \le \frac{t^c - 2^c}{(t-1)^c}$ and

$$X_t \le \frac{1}{t^{2c}} \left(1 - \frac{1}{(t-1)^c} + \frac{1}{t^c}\right) \le \frac{1}{t^{2c}},$$

proving the desired result.

$\square$

**Lemma E.8.** *For any $t \ge 3$, $\tau \in [t-2]$ and $c \in [0,1]$*

$$Q_{t,\tau} = \sum_{i=2}^{t-\tau} \left| \prod_{j=i+1}^{t} \left(1 - \left(\frac{2}{j}\right)^c\right) \frac{1}{i^c} - \prod_{j=i+\tau+1}^{t} \left(1 - \left(\frac{2}{j}\right)^c\right) \frac{1}{(i+\tau)^c} \right| \le \frac{3\tau}{t^c}$$

*Proof.* Note that

$$Q_{t,\tau} = \sum_{i=2}^{t-\tau} \prod_{j=i+\tau+1}^{t} \left(1 - \frac{2^c}{j^c}\right) \frac{1}{(i+\tau)^c} \left| 1 - \prod_{j=i+1}^{i+\tau} \left(1 - \frac{2^c}{j^c}\right) \frac{(i+\tau)^c}{i^c} \right|,$$

so applying Lemma E.9,

$$Q_{t,\tau} \le \sum_{i=2}^{t-\tau} \prod_{j=i+\tau+1}^{t} \left(1 - \frac{2^c}{j^c}\right) \frac{3\tau}{(i+\tau)^{2c}}.$$

We finally note that

$$Q_{t,\tau} \le \sum_{i=2+\tau}^{t} \prod_{j=i+1}^{t} \left(1 - \frac{2^c}{j^c}\right) \frac{3\tau}{i^{2c}}$$

$$\le \sum_{i=2}^{t} \prod_{j=i+1}^{t} \left(1 - \frac{2^c}{j^c}\right) \frac{3\tau}{i^{2c}}.$$

and applying the bound in Lemma E.6,

$$Q_{t,\tau} \le \frac{3\tau}{t^c}.$$

$\square$

**Lemma E.9.** *For any $t \ge 2$ and $\tau \ge 1$,*

$$\left| 1 - \prod_{i=t+1}^{t+\tau} \left(1 - \frac{2^c}{i^c}\right) \frac{(t+\tau)^c}{t^c} \right| \le \frac{3\tau}{(t+\tau)^c}.$$

*Proof.* For every $t \ge 2$ we will use induction on $\tau \ge 1$ to show that the stated bound holds.

**Setting up the induction.** Note that we can express

$$R_{t,\tau} = \prod_{i=t+1}^{t+\tau} \left(1 - \frac{2^c}{i^c}\right) \frac{(t+\tau)^c}{t^c},$$

using the telescoping product

$$\frac{(t+\tau)^c}{t^c} = \prod_{i=t+1}^{t+\tau} \frac{i^c}{(i-1)^c},$$

so that

$$R_{t,\tau} = \prod_{i=t+1}^{t+\tau} \left(1 - \frac{2^c}{i^c}\right) \left(1 + \frac{1}{i-1}\right)^c.$$

We will now show that for any $t \geq 2$, then $|1 - R_{t,1}| \leq \frac{3}{(t+1)^c}$ for the base case, and if $|1 - R_{t,\tau}| \leq \frac{3\tau}{(t+\tau)^c}$ then $|1 - R_{t,\tau+1}| \leq \frac{3(\tau+1)}{(t+\tau+1)^c}$ for the inductive step.

**Base case.** For the base case with $\tau = 1$, observe that

$$1 - R_{t,1} = 1 - \left(1 + \frac{1}{t}\right)^c + \frac{2^c}{t^c},$$

so we can argue that

$$1 - R_{t,1} \leq \frac{2^c}{t^c} = \frac{(t+1)^c}{t^c} \cdot \frac{2^c}{(t+1)^c} \leq \frac{3}{(t+1)^c},$$

and

$$R_{t,1} - 1 \leq \left(1 + \frac{1}{t}\right)^c - 1 \leq \frac{c}{t} \leq \frac{2^c}{t^c} \leq \frac{3}{(t+1)^c}.$$

This implies that the condition holds for the base case since

$$|1 - R_{t,1}| \leq \frac{3}{(t+1)^c}.$$

**Inductive step.** We now move to the inductive step and note that

$$1 - R_{t,\tau+1} = \left[1 - \frac{(t+\tau+1)^c - 2^c}{(t+\tau)^c}\right] + \frac{(t+\tau+1)^c - 2^c}{(t+\tau)^c}\left[1 - R_{t,\tau}\right],$$

and by the triangle inequality

$$|1 - R_{t,\tau+1}| \leq \left|1 - \frac{(t+\tau+1)^c - 2^c}{(t+\tau)^c}\right| + \frac{(t+\tau+1)^c - 2^c}{(t+\tau)^c}|1 - R_{t,\tau}|.$$

Focusing on the first term we can apply the same logic as before to conclude that

$$1 - \frac{(t+\tau+1)^c - 2^c}{(t+\tau)^c} \leq \frac{2^c}{(t+\tau)^c},$$

and

$$\frac{(t+\tau+1)^c - 2^c}{(t+\tau)^c} - 1 \leq \frac{c}{t+\tau} \leq \frac{2^c}{(t+\tau)^c}$$

or equivalently

$$\left|1 - \frac{(t+\tau+1)^c - 2^c}{(t+\tau)^c}\right| \leq \frac{2^c}{(t+\tau)^c}.$$

We now let $g_{t+\tau}(c)$ be as defined in Lemma E.10 and use the inductive hypothesis on the second term to conclude that

$$
\begin{aligned}
|1 - R_{t,\tau+1}| &\leq \frac{2^c}{(t+\tau)^c} + \left(\frac{(t+\tau+1)^c - 2^c}{(t+\tau)^c}\right)\frac{3\tau}{(t+\tau)^c}\\
&= \left(\frac{t+\tau+1}{t+\tau}\right)^c \frac{2^c}{(t+\tau+1)^c} + g_{t+\tau}(c)\cdot \frac{3\tau}{(t+\tau+1)^c}.
\end{aligned}
$$

We now apply Lemma E.10 to conclude that

$$
\begin{aligned}
|1 - R_{t,\tau+1}| &\leq \frac{3}{(t+\tau+1)^c} + \frac{3\tau}{(t+\tau+1)^c}\\
&\leq \frac{3(\tau+1)}{(t+\tau+1)^c},
\end{aligned}
$$

proving the desired result. $\qquad\square$

**Lemma E.10.** *For $u \geq 3$ and $c \in [1/2, 1]$ the function*

$$g_u(c) := \left(\frac{(u+1)^c - 2^c}{u^c}\right)\left(\frac{u+1}{u}\right)^c,$$

*satisfies $g_u(c) \leq g_u(1) \leq 1$.*

*Proof.* By computation we find that the derivative of our function at $c \in [1/2, 1]$ is

$$g'_u(c) = \frac{(u+1)^c}{u^{2c}}\left(2(u+1)^c \log\left[\frac{u+1}{u}\right] - 2^c \log\left[\frac{2(u+1)}{u^2}\right]\right).$$

Note that since $u \geq 3$ we have $\frac{u+1}{u} > 1$ and $\frac{2(u+1)}{u^2} \leq \frac{8}{9} < 1$, so that for any $u$, $g'_u(c) > 0$ and the function $g_u(c)$ is increasing. Therefore, for any $u$,

$$g_u(c) \leq g_u(1) = \frac{(u+1)^2 - 2(u+1)}{u^2} = 1 - \frac{1}{u^2} \leq 1.$$

$\qquad\square$

