# OpenReview forum: "Online Conformal Prediction via Online Optimization"
_ICML.cc/2025/Conference — ICML 2025 poster_

### Official Review · Reviewer_KPdH · 2025-03-08

**Overall Recommendation:** 4

**Summary:**

The authors propose an algorithm for online conformal prediction that achieve both long-run deterministic coverage as well as conditional coverage under stochastic data which is well-behaved. The idea is to train a predictive model of the quantile on the sequence of data. Different results are assumed under different assumption one model well-specification and data generating distributions. The approach is shown to reduce the set prediction set size and cumulative loss.

**Claims And Evidence:**

Yes, the developed theory is convincing and the results are clearly presented.

**Essential References Not Discussed:**

N/A

**Experimental Designs Or Analyses:**

I’ve reviewed the experiments in the main text, and they are sound; however, I could not find any results regarding adversarial coverage

**Methods And Evaluation Criteria:**

Yes, the proposed algorithm based on the estimation of the NC scores is thoroughly evaluated. In particular the algorithm’s benefit is shown to have better prediction set size and cumulative loss. However, I couldn’t find any long-run coverage results in the tables in the main text.

**Other Comments Or Suggestions:**

N/A

**Other Strengths And Weaknesses:**

The paper is original in that it proposes a new algorithm for online conformal prediction, achieving both long-run deterministic coverage and conditional coverage under well-behaved stochastic data. The core idea of the algorithm is to train a predictive model of the quantile based on the data sequence. This approach is novel, and, to the best of my knowledge, the asymptotics of online coverage have only been explored in i.i.d. settings for standard ACI [Online Conformal Prediction with Decaying Step Sizes] or localized ACI [Localized Adaptive Risk Control]. I don’t see any weaknesses in the existing approach, other than the fact that the current experiments lack results comparing long-run coverage.


## update after rebuttal: My score remains positive and unchanged after the rebuttal.

**Questions For Authors:**

N/A

**Relation To Broader Scientific Literature:**

The results are well-connected to existing literature and the Appendix A does a good job at providing an overview of existing results in online calibration and stochastic optimization.

**Theoretical Claims:**

I did not have checked the proof of the theorems.

---

> ### Author Rebuttal · Authors · 2025-04-01
>
> We thank the reviewer for their appreciation of our work.
>
> **However, I couldn’t find any long-run coverage results in the tables in the main text.**
>
> The long-run coverage for all the methods in Table 1 is at least 0.87 - we will make this clearer in the final version. We have also attached plots showing the long-run coverage rates on the preregistered ERCOT data at this link [https://drive.google.com/file/d/1AT1hbP3ohYowIt0_Hy209yadYvZjl9cJ/view?usp=sharing], where we see that all methods converge rather quickly. We have also attached plots showing the long-run coverage tradeoff for LQT in terms of the learning rate, for ERCOT data.
>
> **I don’t see any weaknesses in the existing approach, other than the fact that the current experiments lack results comparing long-run coverage.**
>
> Please see the comment above.

---

> > ### Comment · Reviewer_KPdH · 2025-04-03
> >
> > Thanks a lot for the additional experiments! I think plots like the one attached should be included, perhaps in the appendix. Unfortunately, tables don’t effectively showcase long-term coverage. I confirm that this looks like a good paper to me!

---

### Official Review · Reviewer_Ve1r · 2025-03-09

**Overall Recommendation:** 2

**Summary:**

The paper proposes an online conformal prediction method that integrates stochastic optimization to improve coverage guarantees. It claims existing methods use bang-bang control and lack time-conditional guarantees. The proposed method supposedly achieves better theoretical guarantees and outperforms baselines in 15 datasets, including a pre-registered electricity demand forecasting task.

**Claims And Evidence:**

The paper may inflate its novelty, rely on strong assumptions, and adopt a relaxed **batch** online setting. Moreover, it misuses a lot of basic conformal inference terminology and misinterprets existing works.

1. The claim that existing methods resemble bang-bang control is misleading.

    Actually, existing online conformal inference methods are variants of online gradient descent (OGD)[1], instead of a "2 step on–off controller that switches abruptly between two states". This is a fundamental misinterpretation of current online conformal inference methods.

2. Online setting vs batched online setting.

  From the abstract, the paper focuses on modeling data dependency in online conformal prediction. However, in the methodology, it shifts to a batched online setting, which is an obvious relaxation of the pure online setting.

3. In Eq. (1), the authors claim the ACI guarantee

    $\left|\frac{1}{T} \sum_{t=1}^{T} \text{err}_t - \alpha\right| = o(1)$, can be obtained by trivially setting $-\infty, \infty$ threshold. This is unclear, as the target error rate is $\alpha$, not $0$.

4. intertwined theoretical contribution

    While the title of the paper is 'Online Conformal Prediction,' the proposed methods are an intertwined combination of conformal inference and asymptotic methods with strong model assumptions. Notably, conformal prediction is unique in its finite-sample guarantee and distribution-free nature. It difficult to discern the main purpose of this paper.

5. Impractical assumption of well-specified models and conformal predictor is linear in its parameter.

    The theoretical guarantees assume well-specified models and, moreover, a linear conformal predictor. This is highly impractical and further proves the paper's overstatement of its contributions.


[1] Ramalingam, Ramya, Shayan Kiyani, and Aaron Roth. "The Relationship between No-Regret Learning and Online Conformal Prediction." arXiv preprint arXiv:2502.10947 (2025).

**Essential References Not Discussed:**

Most of the related works are mentioned, but the discussion lacks depth and contains fundamental misunderstandings.

**Experimental Designs Or Analyses:**

1. Since the main novelty of the paper is claimed upon the modeling of the data dependency and auto-regressive cases, the authors should compare with baseline in [2]. [2] handles time series with general dependency and auto-regressive case.

    However, the model does not include it in the baselines.

2. Lack of essential 'sets vs time step' figure.
    The paper contains no figures of prediction sets visualization for either its method or the baselines, which are essential in an online setting.

[2] Zaffran, Margaux, et al. "Adaptive conformal predictions for time series." International Conference on Machine Learning. PMLR, 2022.

**Methods And Evaluation Criteria:**

See above.

**Other Comments Or Suggestions:**

There are lots of misinterpretation of conformal methods and ambiguity in the presentation. For example,

1. "The first algorithm to propose this approach, ACI, constructs confidence sets using a single parameter that governs their level of conservativeness.''

    In fact, in ACI, the choice of the parameter $\gamma$ creates a trade-off between adaptability and stability. A larger $\gamma$ induces greater volatility in the realized coverage. However, "conservativeness" in conformal prediction refers to the size of the prediction intervals, which is a distinct concept.

2. Despite the technical errors through the whole paper, the grammar is also cumbersome with ambiguous statements. For example,

    "so that we do not rely on (likely invalid) modeling assumptions, but **which can in fact do the “right thing,”** adapting to underlying structure when it exists."

    {$(X_t, Y_t)$}$_t$

3. The paper overuses $q_t$ to denote both the threshold and the conformal predictor, which may cause confusion.

**Other Strengths And Weaknesses:**

See above.

**Questions For Authors:**

Since the main claim of the paper has a major flaw, along with an inaccurate theoretical framing, I don't think requesting additional experiment results would be helpful.

**Relation To Broader Scientific Literature:**

N/A

**Theoretical Claims:**

I did not check the correctness of the proofs because of the above issues.

---

> ### Author Rebuttal · Authors · 2025-04-01
>
> We thank the reviewer for their detailed comments. We first provide a few general remarks.
>
> Our work provides stronger guarantees than standard online conformal works, by offering not just the standard adversarial marginal guarantee, but also a stronger conditional one when the data is non-adversarial. Crucially, our algorithm is adaptive when the process is non-adversarial. Adaptive and optimistic algorithms have been effective solutions in several areas of machine learning, such as online convex optimization and reinforcement learning.
>
>
> **Claims & Evidence**
>
> 1. This is not a misinterpretation of online conformal methods when we take scalar quantile tracker (SQT, Example 3.1) updates to be controller actions. The SQT system, with $\eta=1$ and $S_t = 0.1$ for all $t$ for simplicity, behaves precisely like a bang-bang controller with on signal = +0.9, off signal = -0.1, and set point = 0.1. Namely, if the state ($q_t$) is below the set point (0.1), the controller will always output the on signal (+0.9), and it will output the off signal (-0.1) otherwise.
>
> 2. The batch version of our algorithm is fully compatible with the standard online conformal prediction setting. In the batched algorithm, the conformal predictor’s parameter is updated after every m (the batch size) steps, but *the prediction set at each time t is always generated immediately after the covariate X_t is observed.*
>
> 3. We state that one can trivially obtain the guarantee “by *alternating* between $q_t = \infty$ and $q_t = -\infty$.” The thresholds should not always be $\infty$ or $-\infty$ as this would indeed lead to error rates of 0 or 1, respectively.
>
> 4. The main purpose of this paper is to show that we can develop online conformal algorithms that enjoy strong guarantees for stochastic processes *while maintaining adversarial guarantees*. The only asymptotic part of the paper is Theorem 5.1, and this is in the analysis of the algorithm. Theorems 4.1, 5.4, and 6.1 are finite-sample guarantees for our algorithm.
>
> 5. Our theoretical guarantees extend beyond well-specified models – we devote Section 6 to the mis-specified setting and preserve the adversarial coverage guarantees with no such assumptions. Our linear model assumption represents a generalization over methods such as ACI, which only considers scalars, and our experimental results across 17 datasets, along with our pre-registered experiment, suggest this generalization leads to significant performance improvements.
>
> **Experimental Designs Or Analyses**
>
> 1. We have adapted the code provided by the authors of [2] to compare with our method and baselines on the preregistered ERCOT data. The algorithm provided in [2], AgACI, comes with several aggregation strategies, and we provide results at this link [https://drive.google.com/file/d/1AT1hbP3ohYowIt0_Hy209yadYvZjl9cJ/view?usp=sharing].
> These all achieve marginal coverage, but the set sizes and quantile losses are all larger than every method appearing in Table 1. We also point out that AgACI is not proven to achieve long-run coverage and does not provide theoretical guarantees.
>
> 2. We have attached these figures for our main pre-registered dataset and several methods at the link above (Figure 1 contains essentially this figure, we also attached analogous figures for the ACI and PID(theta) baselines for comparison).
>
> **Due to the lack of essential figures ... but did not find them.**
>
> Please see the comment above.
>
> **Additionally, I read through the related work section ... auto-regressive case.**
>
> Thank you for the suggestion, we will add a discussion of [2] to the final version.
>
> **Other Comments Or Suggestions**
>
> 1. The parameter that we refer to here is the threshold, not the learning rate, which is instead a hyperparameter.
>
> 2. In spite of the reviewer’s comments, we have not identified any technical errors. The reviewer, as far as we can tell, also does not seem to identify technical errors in our theoretical or experimental results. While we are sure there are typos that we have missed, it seems unreasonable to ask for more (on the technical side) than correct proofs and pre-registered real-world experiments.
>
>     In Section 2, we define “right thing” formally– always achieving the adversarial guarantee (1) while also achieving (2) when possible. We will clarify this statement in the final version.
>
>      We would be happy to clarify if anything about the notation $\\{(X_t,Y_t)\\}_t$ is unclear.
>
> 3. We have been careful about dropping the argument to the conformal predictor when clear from context. If there is any instance that caused confusion, we would be happy to clarify.
>
> **Since the main claim of the paper has a major flaw, along with an inaccurate theoretical framing, I don't think requesting additional experiment results would be helpful.**
>
> Despite the reviewer’s comments, we have not identified any “major flaw” or “inaccurate theoretical framing”. Perhaps the reviewer can elaborate on what these refer to.

---

> > ### Comment · Reviewer_Ve1r · 2025-04-05
> >
> > I thank the authors for addressing my questions regarding the addition of necessary result figures and baselines. After these adaptations, the paper is more persuasive in terms of empirical validity and efficiency.
> >
> > On the other hand, I believe the presentation still requires careful revision, particularly in the discussion of current online conformal prediction methods:
> >
> > - A more respectful opinion on current online conformal prediction methods, especially acknowledge that they are online optimization methods (e.g. online gradient descent). This perspective is supported by several recent conformal prediction papers (e.g., [1]–[2]).
> >
> > Regarding the presentation of the theoretical results, the logical flow between the various theorems is unclear. Different theorems rely on different assumptions (e.g., correct model specification, linear parameters) and offer different types of guarantees (e.g., asymptotic vs. finite-sample). It would greatly benefit the paper if the authors could clearly highlight the most general assumptions under which they obtain theoretical guarantees—whether asymptotic or finite-sample.
> >
> > [1] Ramalingam, Ramya, Shayan Kiyani, and Aaron Roth. "The Relationship between No-Regret Learning and Online Conformal Prediction." arXiv preprint arXiv:2502.10947 (2025).
> >
> > [2] Angelopoulos, Anastasios N., Rina Foygel Barber, and Stephen Bates. "Online conformal prediction with decaying step sizes." arXiv preprint arXiv:2402.01139 (2024).

---

### Official Review · Reviewer_cZQF · 2025-03-16

**Overall Recommendation:** 4

**Summary:**

This paper introduces a conformal inference algorithm in the online setting that achieves both long-term coverage guarantees in adversarial settings, as well as expected coverage (conditioned on the past sequence) that converges to the desired coverage rate in certain stochastic settings. Many existing online conformal prediction algorithms apply some form of online gradient descent on the $(1-\alpha)$-quantile loss, on either a (single-dimensional) prediction of the quantile or non-conformity threshold for the current time-step. The algorithm presented here generalizes this approach by instead performing OGD on a richer parameter class $\Theta$, and then computing the non-conformity threshold $q_t$ as a deterministic function (called the conformal predictor) of the current iterate $\theta_t$.

This algorithm inherits the long-term coverage guarantees of prior work by bounding the outcomes of the conformal predictors (analogous to directly bounding the predicted quantiles / thresholds). In stochastic settings, they define the notion of a process being well-specified if a single parameter in $\Theta$ can correctly predict the quantile that minimizes expected quantile loss across all rounds. In such (or similar) settings, with appropriately chosen learning rates the algorithm converges to this parameter, thus recovering an expected coverage that converges to $1 - \alpha$, with finite-sample guarantees if a batched version of the algorithm is run. When $\Theta$ and the set of conformal predictors are chosen appropriately, well-specified processes include more than just IID settings (such as autoregressive processes).

The authors run experiments with several datasets, showing that their algorithm performs on par with existing methods in the adversarial setting, while achieving better performance in terms of quantile loss and set size when there is more of a linear correlation between the predicted parameter and the appropriate quantile.

**Claims And Evidence:**

The claims are all proved and there is sufficient evidence (in particular the experiment registered by the authors themselves) that their algorithm shows improved performance over prior ones.

**Essential References Not Discussed:**

Not to my knowledge.

**Experimental Designs Or Analyses:**

The experiments are fairly extensive. However, I couldn’t find the long-term coverage rate reported for any of the experiments. A plot showing the convergence rate to the desired coverage across the different methods would have been interesting. Also since LQT with a decaying learning rate is optimized for the stochastic setting, I would have liked to see the trade-off it has when it comes to long-term coverage.

**Methods And Evaluation Criteria:**

Yes, it seems fine.

**Other Comments Or Suggestions:**

As the authors mention, to achieve optimal convergence rates for coverage in the stochastic setting, we set the learning rate in a way that achieves a vacuous bound on long-term coverage in the adversarial sense (and vice versa). Did you think about the settings where you achieve these guarantees at the same time (though not at the optimal rates)?

**Other Strengths And Weaknesses:**

Strengths:

The most interesting aspect of this paper for me is the idea of using higher-dimensional iterates to map to non-conformity thresholds, and performing gradient descent on these iterates rather than on thresholds directly. Turning the online conformal-prediction problem from one where we predict a one-dimensional threshold into one where we predict a higher-dimensional vector that may encode richer information but has a direct connection to the thresholds + coverage, through the conformal predictors $q_t$ at each time-step is clever. It generalizes the prior algorithms in this area in a meaningful way, and opens the possibility of using other kinds of conformal predictors to expand the collection of settings where this well-specification property holds.

Weaknesses:

There seems to be a tradeoff in terms of the stochastic and adversarial coverage guarantees when it comes to setting both the batch size and the set of learning rates, so the optimal rates of both cannot be achieved simultaneously. This is understandable and seems like a necessary tradeoff but in terms of application what approach would be taken to decide which algorithm to use? In unknown settings / sequences, it seems like one would want to set the parameters so that we achieve non-trivial long-term coverage (assuming adversarial), but this would have worse convergence rates than the other methods if you also wanted non-trivial guarantees of the stochastic variety.

**Questions For Authors:**

1.	Section 5 mentions that the finite sample rates are achieved with a particular batch size m and I didn’t get into the details of this proof – could the authors clarify what the relationship between m and the constant C’ in Theorem 5.4 is? Is m a function of T?
2.	How was the learning rate set for the decaying version of LQT in the experiments? Did it decrease at a O(1/t) rate or was it slower to also achieve a non-trivial long-term coverage convergence rate?

**Relation To Broader Scientific Literature:**

This paper has close connections with much of the work using OGD with pinball loss to obtain conformal-type coverage guarantees in adversarial settings, starting with the paper that introduced ACI (Gibbs et al 2021) and pointed out the connection (though not directly through no-regret guarantee), and then subsequent work that built on it with slight modifications (Bhatnagar et al. 2023, Gibbs & Candes 2022). This work follows a very similar idea, the main departure as I’ve understood is noting that in stochastic settings where there is a single parameter that optimizes expected coverage (well-specification), convergence to this parameter is possible using an appropriately chosen sequence of learning rates, and when a richer class of parameters than just the set of possible non-conformity thresholds is used, there are more settings in which the above well-specification property becomes true.

**Theoretical Claims:**

I only briefly looked at most of the proofs, apart from Theorem 4.1 and Lemma 5.2 I looked at completely which both look sound.

---

> ### Author Rebuttal · Authors · 2025-04-01
>
> **I couldn’t find the long-term coverage rate reported for any of the experiments...I would have liked to see the trade-off it has when it comes to long-term coverage.**
>
> At this link [https://drive.google.com/file/d/1AT1hbP3ohYowIt0_Hy209yadYvZjl9cJ/view?usp=sharing], we have attached additional experiments to describe both (1) long-run coverage across several methods, and (2) the trade-off in terms of the learning rate for LQT. For (1), we observe that all methods converge rather quickly, and for (2), we observe that indeed there is a slower convergence for more quickly decaying learning rates.
>
> In general, we do not report the long-run coverage because all methods achieve it. We also point out that Table 6 in the appendix shows that in all but 1 instance, the proposed algorithms (and baselines) achieve at least 0.87 long-run coverage when the target is set to 0.9. The baselines achieved tight marginal coverage since we tuned their hyperparameters on the test set.
>
> **There seems to be a tradeoff in terms of the stochastic and adversarial coverage guarantees when it comes to setting both the batch size and the set of learning rates, so the optimal rates of both cannot be achieved simultaneously. This is understandable and seems like a necessary tradeoff but in terms of application what approach would be taken to decide which algorithm to use? In unknown settings / sequences, it seems like one would want to set the parameters so that we achieve non-trivial long-term coverage (assuming adversarial), but this would have worse convergence rates than the other methods if you also wanted non-trivial guarantees of the stochastic variety.**
>
> We thank the reviewer for their appreciation of our work.
> We offer three answers to the tradeoff question:
> - The batch size and learning rate decay are hyperparameters that can be tuned on validation data, as we have done with the magnitude of the learning rate for our experiments. We sought the best performance on the quantile loss, given a constraint on the long-run coverage, but these heuristics can be changed to fit the practitioner’s goals.
> - Domain-specific knowledge, if available, can help guide this choice. If the well-specified assumption is reasonable, or the errors happen to be linearly correlated often, then faster rates of decay are preferable to speed up convergence, while constant learning rates will provide the tightest coverage when the data is fully adversarial.
> - Choosing intermediate rates of decay (i.e. $\Theta(t^{-0.6})$ as in the paper) allows us to attain both long-run coverage and consistency under well-specification, albeit at slower rates. Across our experiments, we have found this (and small batch sizes) to be a safe choice. We think these hyperparameter questions are important, so we have been careful to empirically validate this intuition through the preregistration.
>
> **As the authors mention, to achieve optimal convergence rates for coverage in the stochastic setting, we set the learning rate in a way that achieves a vacuous bound on long-term coverage in the adversarial sense (and vice versa). Did you think about the settings where you achieve these guarantees at the same time (though not at the optimal rates)?**
>
> Our results apply to a range of learning rates $\Theta(t^{-c})$ for $c \in [0,1]$, not just the $\Theta(1/t)$ learning rates attaining optimal convergence rates in the stochastic setting and $\Theta(1)$ attaining optimal long-run coverage rates in the adversarial setting. Choosing $c \in (0,1)$ thus allows us to attain both long-run coverage and consistency under well-specification asymptotically, albeit at non-optimal rates.
>
> **Section 5 mentions that the finite sample rates are achieved with a particular batch size m and I didn’t get into the details of this proof – could the authors clarify what the relationship between m and the constant C’ in Theorem 5.4 is? Is m a function of T?**
>
> The batch size m is not a function of T. It should be chosen so that the corresponding minimum eigenvalue in each sample covariance matrix has a sufficiently large conditional expectation. In Lemma 5.3, this just leads to a batch size of at least 2. The batch size m is constant, and the constant C’ increases with m.
>
> **How was the learning rate set for the decaying version of LQT in the experiments? Did it decrease at a $O(1/t)$ rate or was it slower to also achieve a non-trivial long-term coverage convergence rate?**
>
> The rate of decay for our experiments is $\Theta(t^{-0.6})$ following Angelopoulos et al. (2024). This is indeed to ensure non-trivial long-run coverage. The only exception is the synthetic experiment in Appendix B3 where we use the optimal $\Theta(1/t)$.

---

### Official Review · Reviewer_oGmv · 2025-03-18

**Overall Recommendation:** 3

**Summary:**

This paper is looking at the problem of online conformal prediction, focusing on deriving stronger (conditional) guarantees than long run marginal coverage. They propose to look at asymptotic absolute value coverage deviation from the nominal value at each time step, conditioned on all the past observations. They then design and algorithm based on quantile tracking through pinball loss minimization. Relying on standard proof techniques through the connection of pinball loss derivatives and coverage, they both and adversarial long run coverage validity and stronger conditional coverage guarantees in the stochastic setup.



I thank the authors for their response. I agree with them. I strongly encourage the authors to incorporate the comments on improving their presentations. In particular, it is good that the list of contributions would be stated in the introduction with pointers to the associated parts of the paper.

The paper [1] that the authors mentioned is indeed what i referred to.

I keep my score and vote for acceptance.

**Claims And Evidence:**

All the claims are backed by meaningful theorems and experiments.

**Essential References Not Discussed:**

A very similar notion of conditional coverage has been previously discussed by [1], in a stochastic iid batch setting, named mean squared conditional error.

**Experimental Designs Or Analyses:**

I have not checked all the details of the experiments, but from a high level perspective, everything makes sense.

**Methods And Evaluation Criteria:**

The evaluations make sense to me.

**Other Comments Or Suggestions:**

In the current version, the organization of the assumptions needed for the theoretical argument is hard to navigate. It's good to state the assumptions once and assign numbers (or letters to them), assumption 1, assumption 2, ..., and then reference to them in the theorem statements. This will improve the clarity of the theorems. That being said, it is also good, to discuss whether such assumptions are conventional and/or how restricting they are. (Though I appreciate the three examples given, but it is also good to point out some examples, which might be of importance in practice, but your assumptions don't hold then.)

**Other Strengths And Weaknesses:**

The paper needs improvement in the presentation, particularly when presenting the algorithm. The algorithm suddenly appears without motivating or hinting toward what are the algorithmic principles behind it. To that end, it is not obvious what is the novel/new tweak to the algorithm beyond just running online gradient descent on pinball loss, which is the core algorithm in the literature.

**Questions For Authors:**

What are the list of contributions this paper is claiming? is there also any algorithmic contributions? or it is mainly analyzing the online gradient descent on pinball loss under different assumptions and different notions of coverage?

**Relation To Broader Scientific Literature:**

The online conformal prediction is a very important and active area of research, as often time in practice there is a need to do uncertainty quantification in an online fashion. Even though conformal prediction can be extended fairly easily to the adversarial setup, but the theoretical guarantees of the form of long run marginal coverage are very weak and could be somewhat misleading. This work make progress toward offering stronger guarantees, which could be of interest in real world applications, providing a more robust and interpretable uncertainty quantification.

**Theoretical Claims:**

The theoretical claims and proofs sound reasonable to me.

---

> ### Author Rebuttal · Authors · 2025-04-01
>
> **A very similar notion of conditional coverage has been previously discussed by [1], in a stochastic iid batch setting, named mean squared conditional error**
>
> We thank the reviewer for pointing us to this reference. We would like to verify that it refers to [1], and we would be happy to incorporate it into our discussion of conditional coverage in the i.i.d. offline setting.
> [1] Kiyani, S., Pappas, G. and Hassani, H. Conformal prediction with learned features. arXiv:2404.17487 [cs.LG]. 2024.
>
> **The paper needs improvement in the presentation...beyond just running online gradient descent on pinball loss, which is the core algorithm in the literature.**
>
> Our main contribution is a new algorithm that both satisfies adversarial guarantees (long-run coverage) and does the “right thing” (achieves time-conditional coverage) if the data is not adversarial. Most online conformal work focuses solely on adversarial guarantees, but we provide a novel approach that can adapt to both settings.
>
> In terms of the algorithm itself, the main novelty is that we run online gradient descent on the quantile loss with more complex models for the conformal predictor. Past work (e.g. ACI) only considers scalar models, while our algorithm applies to arbitrary parametric models. For our stochastic guarantees, we simply require that the conformal predictor q_t(\theta) is linear in \theta. We will preface the algorithm with more motivation and discuss the novelties in the final version.
>
> **In the current version, the organization of the assumptions...but your assumptions don't hold then.)**
>
> We will make the assumptions for each guarantee clearer, and expand on how conventional and restrictive they are in the final version. At a high level, our assumptions fall into two categories: model assumptions and distributional assumptions. Our model assumptions determine the structure of the conformal predictor; most of our analysis requires that the conformal predictor $q_t(\theta)$ is linear in its parameter $\theta \in \mathbb{R}^d$ (though this can in fact be dropped for the adversarial guarantee in Theorem 4.1). These are less restrictive than conventional model assumptions that only allow for scalar models (as discussed in our response to the previous point). Our distributional assumptions hold for many data-generating processes. For example, in Theorem 5.4 we only require that the conditional quantiles are a linear function of the past, which is true for AR(p) processes and i.i.d. data. This is less restrictive than (Angelopoulos et al., 2024), which only considers the i.i.d. setting.
>
>
> Examples 3.1, 3.2, and 3.3 all satisfy the linear model assumption, and each has a description of when it is well-specified (for the guarantees of Section 5 to apply). Our experiments focus on these three examples, where we find that 3.2 and 3.3 perform the best. Algorithm 1 does allow for models that do not satisfy our assumptions (e.g. nonlinear), but these do not enjoy the same stochastic theoretical guarantees, which are a crucial part of our contribution. We believe that extending our guarantees to more complex model classes that could provide further practical benefits is a promising area of future research, and we will make this point clearer in the final version.
>
> **What are the list of contributions this paper is claiming? is there also any algorithmic contributions? or it is mainly analyzing the online gradient descent on pinball loss under different assumptions and different notions of coverage?**
>
> Concretely, our contributions are:
> - A new algorithm for online conformal prediction that allows for conformal predictors beyond scalars as in ACI.
> - Theoretical guarantees showing that our algorithm always achieves
>   -  marginal coverage,
>
>     and under appropriate stochastic assumptions:
>   - conditional coverage when well-specified
>   - relaxed conditional coverage when mis-specified.
> - Experiments spanning 17 datasets and a pre-registration to compare our method with existing alternatives and provide evidence for the empirical benefits of our algorithm.
>
> As previously discussed, the algorithmic contribution is to extend the model for the conformal predictor beyond a single scalar parameter (as in ACI) to parametric functions, and specifically for our theoretical results: linear functions of feature maps. This new algorithm can adapt to stochastic data, where it enjoys strong convergence guarantees both when well-specified and mis-specified. While our algorithm can replicate SQT as a special case, it encompasses a larger family of procedures and is thus not equivalent to exploring the behavior of existing algorithms under different assumptions. Moreover, our notions of coverage are not just different, but more general and stronger: we satisfy the standard marginal coverage property in the adversarial setting, but also a notion of conditional coverage when the data is stochastic. This has not been explored in past work beyond the i.i.d. setting.

---

### Decision · Program_Chairs · 2025-05-01

**Decision:**

Accept (poster)

**Comment:**

This paper proposes a family of algorithms for online conformal prediction that provide long-run coverage guarantees in the adversarial data setting, as well as consistency guarantees in the stochastic setting. While existing work, such as that of Angelopoulos et al. (2024), has also proposed online conformal prediction algorithms with guarantees under both adversarial and stochastic setups, the framework developed in this paper allows for handling potentially non-i.i.d. stochastic data. The core idea is to consider a parametric model for the conditional quantiles. Both the well-specified and misspecified cases are analyzed.

Such an explicit parametric modeling approach, though classical in many other areas in statistics and learning, actually leads to solid contributions to the field of conformal prediction. All reviewers are positive about this work, except Reviewer Ve1r who is concerned about the presentation. Hence, I recommend acceptance of this submission.

It would be beneficial to discuss whether the best-in-class-type guarantee in the misspecified case is an unavoidable feature of the parametric modeling approach, given that guarantees of this type do not appear in the work of Angelopoulos et al. (2024) for i.i.d. data.